# Quantifying the controllability of coarsely characterized networked dynamical systems

## Abstract

We study the controllability of large-scale networked dynamical systems when complete knowledge of network structure is unavailable. In particular, we establish the power of learning community-based representations to understand the ability of a group of control nodes to steer the network to a target state. We are motivated by abundant real-world examples, ranging from power and water systems to brain networks, in which practitioners do not have access to fine-scale knowledge of the network. Rather, knowledge is limited to coarse summaries of network structure. Existing work on "model order reduction" starts with full knowledge of fine-scale structure and derives a coarse-scale (lower-dimensional) model that well-approximates the fine-scale system. In contrast, in this paper the controllability aspects of the coarse system are derived from coarse summaries *without* knowledge of the fine-scale structure. We study under what conditions measures of controllability for the (unobserved) fine-scale system can be well approximated by measures of controllability derived from the (observed) coarse-scale system. To accomplish this, we require knowledge of some inherent parametric structure of the fine-scale system that makes this type of inverse problem feasible. To this end, we assume that the underlying fine-scale network is generated by the stochastic block model (SBM) often studied in community detection. We quantify controllability using the "average controllability" metric and bound the difference between the controllability of the fine-scale system and that of the coarse-scale system. Our analysis indicates the necessity of underlying structure to make possible the learning of community-based representations, and to be able to quantify accurately the controllability of coarsely characterized networked dynamical systems.

## 1 Introduction

In this paper we study controllability for networked dynamical systems when our knowledge of system structure is limited to coarse summaries. We are motivated by myriad real-world settings where system identification must be performed based upon measurements taken by low-resolution instruments unable to probe fine-scale structure. Our motivating example is the human brain. While efforts are under way to produce a canonical human brain map, our knowledge of the brain as an interconnected, network system is not yet to the level of the whole-brain individual neuron (Betzel and Bassett, 2017). And yet, motivated by emerging medical technologies, there are important control tasks we would like to tackle. For example, novel brain implants designed for epilepsy patients aim to "steer" the brain away from states that correspond to seizures (Heck et al., 2014; Muldoon et al., 2016). Our goal is to quantify the controllability of a fine-scale networked dynamical system given access only to coarse knowledge of network structure. Generally, without parametric structure, this is impossible. But real networks do have structure and so in our model we assume the fine-scale network has a connectivity induced by an underlying stochastic block model (SBM).

Approximation of high-dimensional (fine-scale) dynamical systems by lower-dimensional (coarse-scale) ones is known as "model order reduction" (MOR) in the controls literature. There is a key difference in assumptions that differentiate our setting from that literature. In MOR the starting point is a complete description of the high-dimensional system. The task is to formulate a lower-dimension system, the dynamics of which well-approximate those of the full system. In contrast, we start from coarse summaries of the fine-scale system. We do not have access to the fine-scale dynamics and must exploit parametric knowledge (via the assumption of a generative SBM). One might think of the

distinction as akin to "active" versus "passive" MOR. Traditional MOR is active in that it actively decides how to coarsen the system to yield the best reduction. But for us, our knowledge is limited by the precision of our instrumental observations, so passively collected data is our starting point.

Controllability is a function both of system dynamics *and* how we actuate the system (Pasqualetti et al., 2014; Yuan et al., 2013). Herein we assume that we both measure and actuate a system only coarsely. A control question we study is which coarse-level actuations are most "influential" in controlling the underlying fine-scale system. Such knowledge can assist with actuation selection; e.g., in our motivating epilepsy application, where best to position devices to be able to collapse the unstable brain-state oscillations that lead to seizures. (O'Leary et al., 2018; Pazhouhandeh et al., 2019; Kassiri et al., 2017; Shulyzki et al., 2015) To accomplish our goal we characterize the average controllability of a vector of systems, each corresponding to a different coarse-scale actuation input. By comparing these vectors, and because these vectors well approximate the corresponding vectors for the fine-scale system, we aim (in the long term) to produce clinically-usable information for the neurologist.

**Contribution**: Our work is the first of its kind that proposes a learning-based framework for inferring the controllability of fine networks from coarse measurements, and characterizes the mismatch between the controllability of the coarse and fine-scale networks. We study two approaches.

1. In Section 5, we build from MOR. We define an auxiliary, fictitious, reduced-order system based on the coarse data, and use the average controllability vector of this system to approximate that of the fine-scale system. We derive a tight upper bound on the "approximation-error" which is the sum of two terms. One term goes to zero as the coarse network size increases and the network becomes dense. The second term is a function of the synchronization between the coarse summary data and the underlying community structure. If synchronization is not sufficiently high, this term may not approach zero even as the network size increases.

2. In Section 6 we learn the fine-scale system's average controllability vector directly from the coarse data. This learning-based algorithm builds on the mixed-membership algorithm of Mao et al. (2017) for unsupervised learning of the parameters and the community structure of a SBM. We derive a tight upper bound on estimation error and characterize its convergence. Although the error bound implicitly depends on synchronization, unlike in the MOR-based approach, the error of this approach converges to zero as the coarse network size and its density increases.

## 2 BACKGROUND / RELATED WORK

*Coarsened SBM as a generative process*: The study of extracting community structure from coarse summaries is recent. The authors in (Ghoroghchian et al., 2021) used the stochastic block model (SBM), developed in the community detection literature (Abbe, 2017), to lay out a framework for a coarsened and weighted variant of the SBM. We build off those results in this paper. The structure of many real-world networks, including brain networks is, at least empirically, known to have community structure across various spatial scales (Sporns and Betzel, 2016; Pavlović et al., 2020). The SBM and its variants provide a powerful modeling framework to facilitate fundamental understanding of graph community organization and have found applications in many domains, including social and power networks. (Dulac et al., 2020; Funke and Becker, 2019; Abbe, 2017).

*Complex networks controllability*: The development of control methods for complex networks is a major effort in network science (Scheid et al., 2020). Coupling traditional notions of controllability with graph theory reveals several insights into the role of network structure (e.g., presence of communities, diameter, and sparsity), size, and edge weight strength in controlling large-scale networks (Wu-Yan et al., 2018; Kim et al., 2018; Constantino et al., 2019; Sun, 2015). Further one may want to understand which group of nodes, when actuated as inputs, can be used to steer the network to an arbitrary target state, and at what cost (Cortesi et al., 2014; Gu et al., 2015). Recent works in network neuroscience Gu et al. (2015) have popularized the notion of *average controllability*. This scalar metric associate a measure the relative control influence of a group of nodes. In this paper we consider a vector of such scalar measures to study the comparative influence of different sets of nodes. To the best of our knowledge ours is the first work that characterizes this type of error bounds for the controllability of coarse graphs.

## 3 PRELIMINARY NOTIONS

**Notation**: We denote vectors and matrices using bold faced small and upper case letters. The $n$ dimensional all-ones and -zero vectors are denoted by $\mathbf{1}_n$ and $\mathbf{0}_n$. For $\mathbf{M} = [\mathbf{M}_{uv}] \in \mathbb{R}^{n \times m}$, define $\|\mathbf{M}\|_\infty = \max_{1 \le u \le n} \sum_{v=1}^m |\mathbf{M}_{uv}|$; $\|\mathbf{M}\|_{\max} = \max_{u,v} |\mathbf{M}_{uv}|$; and $\|\mathbf{M}\|_2 = \sqrt{\lambda_{\max}(\mathbf{M}^\mathsf{T}\mathbf{M})}$. Let $m = n$, then define the spectral radius by $\rho(\mathbf{M}) = \max_i\{|\lambda_i|\}$; $\operatorname{diag}(\mathbf{M}) = [\mathbf{M}_{11}, \dots, \mathbf{M}_{nn}]^\mathsf{T} \in \mathbb{R}^n$; and $\operatorname{Diag}(\mathbf{M})$ sets the off-diagonal entries of $\mathbf{M}$ to zero. For matrices $\mathbf{M}'_i s$ with arbitrary dimensions, $\operatorname{BlkDiag}(\mathbf{M}_1, \dots, \mathbf{M}_d)$ denotes the block diagonal matrix. The inequality $\mathbf{M}_1 \le \mathbf{M}_2$ implies element wise inequality. We write $f(n) = \mathcal{O}(h(n))$ iff there exist positive reals $c_0$ and $n_0$ such that $|f(n)| \le c_0 h(n)$ for all $n \ge n_0$. The support of a vector, $\operatorname{supp}(\mathbf{m})$, is the set of indices $i$ such that $\mathbf{m}_i \ne 0$. The cardinality of a set $\mathcal{V}$ is denoted by $|\mathcal{V}|$. For a positive integer $m$, we denote $[m] \triangleq \{1, \dots, m\}$. $\mathbf{1}(\mathbf{m})$ returns a vector of same size with non-zero replaced by 1.

**Networks**: A network is defined by an un-directed graph $\mathcal{G} \triangleq (\mathcal{V}, \mathcal{E})$, where the node set $\mathcal{V} \triangleq \{1, \dots, n\}$ and edge set $\mathcal{E} \subseteq \mathcal{V} \times \mathcal{V}$. For an edge $(u, v) \in \mathcal{E}$, assign the weight $\mathbf{A}_{uv} = \mathbf{A}_{vu} \in \mathbb{R}$, and define the *weighted symmetric adjacency matrix* of $\mathcal{G}$ as $\mathbf{A} \triangleq [\mathbf{A}_{uv}]$, where $\mathbf{A}_{uv} = \mathbf{A}_{uv} = 0$ whenever $\mathbf{A}_{uv} \notin \mathcal{E}$. A random network is an un-directed graph with a random adjacency matrix.

### 3.1 LINEAR DYNAMICAL SYSTEM ON RANDOM NETWORK

For a network $\mathcal{G}$ with $n$ nodes and the symmetric adjacency matrix $\mathbf{A}$, associate a state $x_i[k] \in \mathbb{R}$ to the $i$-th node, and let the nodes evolve with the linear and time-invariant (LTI) dynamics [1]:

$$\mathbf{x}[t+1] = \frac{1}{c \cdot \operatorname{tr}(\mathbf{A})} \mathbf{A} \mathbf{x}[t] + \mathbf{B}\mathbf{u}[t], \qquad \forall \ t = 0, 1, \dots. \tag{1}$$

The state $\mathbf{x}[t] = [x_1[t], \dots, x_n[t]]^\mathsf{T}$ is steered to an arbitrary value by an input $\mathbf{u}[t] \in \mathbb{R}^n$. Here, the input matrix $\mathbf{B} = \operatorname{Diag}(\mathbf{b}) \in \mathbb{R}^{n \times n}$, where $\mathbf{b} \in \{0, 1\}^n$ determines which components of $\mathbf{u}[t]$ enters the network[2]. For e.g., for $\mathbf{B} = \operatorname{Diag}(\mathbf{1}_{n_1}, \mathbf{0}_{n-n_1})$, the input enters the network through control nodes set $\mathcal{K} = \{1, \dots, n_1\}$. The normalization $c \cdot \operatorname{tr}(\mathbf{A})$ factor, with appropriately chosen constant $c > 0$, ensures that system in Eq. 1 is asymptotically stable. Finally, we define $\mathbf{A}_{\text{nom}} \triangleq \frac{1}{c \cdot \operatorname{tr}(\mathbf{A})} \mathbf{A}$ for the normalized matrix, and use this convention throughout the paper.

For fixed system matrix $\mathbf{A}_{\text{nom}}$, a necessary and sufficient condition for the asymptotic stability[3] of Eq. 1 is that $\rho(\mathbf{A}_{\text{nom}}) \le 1$. For random $\mathbf{A}_{\text{nom}}$, we consider the probabilistic stability: $\mathbb{P}[\rho(\mathbf{A}_{\text{nom}}) \le 1]$—the greater the value, the greater the chance that $\mathbf{A}_{\text{nom}}$ is stable. For SBM generated random symmetric matrices, we provide sharp non-asymptotic lower bounds on $\mathbb{P}[\rho(\mathbf{A}_{\text{nom}}) \le 1]$.

The networked LTI system in Eq. 1 is *T-step controllable* if $\mathbf{x}[0] = \mathbf{0}$ can be steered to any target state $\mathbf{x} \in \mathbb{R}^n$ for some inputs: $\mathbf{u}[0], \dots, \mathbf{u}[T-1]$. The $T$-step controllability Gramian of Eq. 1 given below, among other things, allows us to study if Eq. 1 is controllable or not.

$$\mathcal{C}_T(\mathbf{A}_{\text{nom}}, \mathbf{B}) = \sum_{t=0}^{T-1} (\mathbf{A}_{\text{nom}})^t \mathbf{B}\mathbf{B}^\mathsf{T} (\mathbf{A}_{\text{nom}})^t. \tag{2}$$

By definition $\mathcal{C}_T(\mathbf{A}_{\text{nom}}, \mathbf{B}) \succeq 0$, and it is well known that $\mathcal{G}$ with $n$ nodes is $T$-step controllable if $n$-step controllable; or equivalently, $\mathcal{C}_T(\mathbf{A}_{\text{nom}}, \mathbf{B}) \succ 0$. For other interesting properties of Eq. 2 we refer to (Chen, 1999). For the simplicity of exposition, we let $T \to \infty$ and consider the infinite time horizon Gramian: $\mathcal{C}(\mathbf{A}_{\text{nom}}, \mathbf{B}) = \lim_{T \to \infty} \mathcal{C}_T(\mathbf{A}_{\text{nom}}, \mathbf{B})$, which exists with $1 - \mathbb{P}[\rho(\mathbf{A}_{\text{nom}}) \ge 1]$; see also Pasqualetti et al. (2014). We drop the notation $(\mathbf{A}_{\text{nom}}, \mathbf{B})$ in $\mathcal{C}$ when the context is clear.

*Average energy*: A widely used metric to measure how hard or easy it is to control the network is average energy: $\int_{\|\mathbf{x}\|_2=1} \mathbf{x}^\mathsf{T} \mathcal{C}^\dagger \mathbf{x} \, d\mathbf{x} / \int_{\|\mathbf{x}\|_2=1} d\mathbf{x}$, which evaluates to $n^{-1}\operatorname{tr}(\mathcal{C}^\dagger)$ (Cortesi et al., 2014). Here, where $\mathcal{C}^\dagger$ is the pseudo inverse, and $\mathbf{x}^\mathsf{T} \mathcal{C}^\dagger \mathbf{x}$ is the minimum control energy needed to steer $\mathbf{x}[0] = \mathbf{0}$ to an arbitrary target state $\mathbf{x} \in \mathbb{R}^n$. Thus, average energy measures the minimum control energy required to steer $\mathbf{x}[0] = \mathbf{0}$ to an arbitrary state uniformly distributed over the unit sphere.

---

[1] One may think our LTI model as the linearized system of an underlying non-linear system. Controllability of non-linear systems require a case by case analysis and we leave this topic for future research.

[2] Alternatively, $\mathbf{B}\mathbf{u}[t] = \mathbf{B}_\mathcal{K}\mathbf{u}_\mathcal{K}[t]$, where $\mathbf{B}_\mathcal{K}$ is the sub-matrix of $\mathbf{B}$ whose columns are indexed by $\mathcal{K} \subset [n]$. However, we stick with notation in Eq. 1 to make our analysis less cumbersome.

[3] The LTI system Eq. 1 is asymptotically stable if $\|\mathbf{x}[t]\|^2 \to 0$ as $t \to \infty$, for $\mathbf{u}[t] = \mathbf{0}$ and $\mathbf{x}[0] \ne \mathbf{0}$.

*Average controllability*: Numerical computation of $\mathcal{C}^\dagger$ for large-scale networks is demanding. Owing to the fact that $\mathrm{tr}(\mathcal{C}^\dagger) \geq 1/\mathrm{tr}(\mathcal{C})$, one uses $\mathrm{tr}(\mathcal{C})$—called the *average controllability*—as a proxy for average energy Gu et al. (2015). The higher the average controllability is for a given set of control nodes defined by $\mathbf{B}$, the smaller their average energy, thus higher their influence on the network.

### 3.2 STOCHASTIC BLOCK MODELS

Stochastic block models (SBMs) are probabilistic models that produce random graphs with planted communities. Formally, let $\mathcal{G}_{\text{fine}} \triangleq (\mathcal{V}, \mathcal{E})$ be the un-directed graph (also referred as fine graph) with $n$ nodes and random edge weights generated according to the general SBM$(n, \mathbf{Q}, \mathbf{p})$:

**Definition 1.** **(General SBM)** In the general SBM$(n, \mathbf{Q}, \mathbf{p})$, the graph $\mathcal{G}_{\text{fine}}$ is partitioned to $K$ disjoint sub-graphs (or communities) of relative sizes $\mathbf{p} = [p_1, \ldots, p_K]$ such that $\mathcal{V} = \cup_{k=1}^K \mathcal{V}_k$. Two nodes $u \in \mathcal{V}_k$ and $v \in \mathcal{V}_{k'}$ are joined by an edge with the weight $\mathbf{A}_{uv} \in \{0, 1\}$, which is drawn with probability $\mathbf{Q}_{kk'}$ independently from other edges, for all $k, k' \in [K]$. □

In General SBM, the probability distribution of weights $\mathbf{A}_{uv}$ is common for all $u \in \mathcal{V}_k$ and $u \in \mathcal{V}_{k'}$. The general SBM$(n, \mathbf{Q}, \mathbf{p})$ thus generates a weighted symmetric graph with $K$ communities with non-identical in- and cross-edge connection probabilities given by $\mathbf{Q} \in [0, 1]^{K \times K}$. Alternatively,

$$\mathbf{A}_{uv} \sim \text{Bernoulli}(\mathbf{Q}_{k,k'}) \qquad \text{if } k, k' \in [K] : \mathbf{P}_{ku} > 0, \mathbf{P}_{k'v} > 0, \tag{3}$$

where the *community membership matrix* $\mathbf{P} = [\mathbf{P}_{kv}] \in \mathbb{R}^{K \times n}$ is given by

$$\mathbf{P}\mathbf{P}^\mathsf{T} = \text{Diag}\left(|\mathcal{V}_1|, \ldots, |\mathcal{V}_K|\right) \text{ with } \mathbf{P}_{kv} = \begin{cases} 1 & \text{if } v \in \mathcal{V}_k, \\ 0 & \text{otherwise.} \end{cases} \tag{4}$$

We define $\mathbf{D} \triangleq \frac{1}{n}\mathbf{P}\mathbf{P}^\mathsf{T}$ which is a diagonal matrix of *relative* community sizes.

**Definition 2.** **(Coarse SBM** Ghoroghchian et al. (2021)) Define a coarse-scale summary to $\mathbf{A}$ as

$$\widetilde{\mathbf{A}} \triangleq \mathbf{W}\mathbf{A}\mathbf{W}^\mathsf{T} \quad \in \mathbb{R}^{m \times m}, \tag{5}$$

where the *coarsening matrix* $\mathbf{W} \in \mathbb{R}^{m,n}$ is (a) $r$-*homogeneous*, for all $i \in [m]$; that is, each $i$-th row of $\mathbf{W}$ (say $\mathbf{w}_i$) has $r$ non-zero terms and all rows have constant row sum and (b) ($\mathbf{W}\mathbf{W}^\mathsf{T} = \frac{1}{r}\mathbf{I}_m$). □

Here, $\widetilde{\mathbf{A}}$ can be interpreted as the symmetric adjacency matrix of an un-directed graph $\mathcal{G}_{\text{coarse}}$ with $m$ nodes— referred to as coarse graph. This interpretation is helpful when we discuss LTI system associated with $\widetilde{\mathbf{A}}$ in Section 5. We refer the nodes in $\mathcal{G}_{\text{coarse}}$ to as *c-nodes*[4] as opposed to the fine nodes in $\mathcal{G}_{\text{fine}}$. Note that $r \leq \frac{n}{m}$, and $r \ll n$ indicating that c-nodes can cover the fine graph only sparsely. In other words, there may exist (several) fine nodes that do not contribute to $\mathbf{A}$ (see Fig. 1). The main goal of our paper is to quantify controllability of $\mathcal{G}_{\text{fine}}$, with community structure, using the coarsely inferred network $\mathcal{G}_{\text{coarse}}$. Importantly, we do not have access to the way the coarse graph is acquired at the time of decision making though the results depend on them.

From Eq. 8 and Eq. 3, the expected quantities of $\bar{\mathbf{A}} \triangleq \mathbb{E}[\mathbf{A}]$ and $\bar{\widetilde{\mathbf{A}}} \triangleq \mathbb{E}[\widetilde{\mathbf{A}}]$ can be computed as

$$\bar{\mathbf{A}} = \mathbf{P}^\mathsf{T}\mathbf{Q}\mathbf{P} \quad \text{and} \quad \bar{\widetilde{\mathbf{A}}} = \underbrace{(\mathbf{W}\mathbf{P}^\mathsf{T})}_{\Phi}\mathbf{Q}(\mathbf{W}\mathbf{P}^\mathsf{T})^\mathsf{T}, \tag{6}$$

where $\Phi \in \mathbb{R}^{m \times K}$ is the *coarse community membership matrix*, and $\Phi_{ik}$ captures the extent to which the $i$-th c-node overlaps with the $k$-th community. Let us also define the resolution parameter.

$$\nu \triangleq \min_{i \in [m], k \in [K] : \Phi_{ik} > 0} \Phi_{ik}. \tag{7}$$

By definition $1/r \leq \nu \leq 1$. In what follows, we assume that a c-node has a constant minimum overlap (i.e. $\nu$) with each community that independent of other system parameters.

The example below will highlight the structural differences among matrices $\mathbf{P}$, $\Phi$, and $\mathbf{W}$.

**Example 1.** *For fine network shown in Fig. 1, the following hold*

---

[4]"c-" stands for compound or coarse.

1. $\mathbf{P} = \mathrm{BlkDiag}(\mathbf{1}_{12}^{\mathsf{T}}, \mathbf{1}_{18}^{\mathsf{T}}, \mathbf{1}_{6}^{\mathsf{T}})$. *Here, $\mathbf{1}_d$ is the d-dimensional all-ones column vector.*

2. $\mathbf{\Phi} = [\mathbf{\Phi}_1^{\mathsf{T}}, \mathbf{\Phi}_2^{\mathsf{T}}, \cdots, \mathbf{\Phi}_6^{\mathsf{T}}]^{\mathsf{T}}$, *where* $\mathbf{\Phi}_1 = [1, 0, 0]$; $\mathbf{\Phi}_2 = [\frac{2}{3}, \frac{1}{3}, 0]$; $\mathbf{\Phi}_3 = \mathbf{\Phi}_4 = [0, 1, 0]$; $\mathbf{\Phi}_5 = [0, \frac{1}{3}, \frac{2}{3}]$; *and* $\mathbf{\Phi}_6 = [0, 0, 1]$.

3. *Finally, each row of the coarsening matrix $W$ has three non-zero entries, all equal to $1/3$.*

*Each c-node in $\{1, 3, 4, 6\}$ covers one community; each c-nodes in $\{2, 5\}$ overlap with two.* ☐

**Assumption 1.** *(SBM scaling Abbe (2017)) For all $k \in [K]$, we have $0 < c_{min} \leq \frac{|\mathcal{V}_k|}{n} \leq c_{max} < 1$, where $c_{min}, c_{max}$ are constants. There exists a $\rho_n \in (0, 1)$ and a non-negative matrix $\mathbf{Q}^{(c)}$, such that*

$$\mathbf{Q} = \rho_n \mathbf{Q}^{(c)}, \tag{8}$$

**Assumption 2.** *(Fully-Synchronized c-node): The coarse graph has at least one pure node per community $k \in [K]$. Formally, for all $k \in [K]$, there exist a coarse node $i \in [m]$ such that $\mathbf{\Phi}_{ik} = 1$.*

**Assumption 3.** *(Uniform Coarsening): There exist $\tilde{c}_{min}$ and $\tilde{c}_{max}$ independent of $m$ such that $\tilde{c}_{min}\mathbf{1}_K \leq \mathbf{1}_m^{\mathsf{T}}\mathbf{\Phi}/m \leq \tilde{c}_{max}\mathbf{1}_K$.*

We also assume that communities have self-connections, that is, $\mathrm{tr}(\mathbf{Q}^{(c)}) > 0$, and $\mathbf{Q}^c \leq \mathbf{1}_{K \times K}$. Assumption 1 helps us uniformly control the sparsity of connection in and cross communities. For several real-world networks, $\rho_n$ typically decreases with $n$ Abbe (2017). Assumption 2 states that for each community there exist at least one c-node that is fully inside one community. We call such c-node fully synchronized (see Remark 1). Finally, Assumption 3 ensures that the relative coverage of each community measured by coarse nodes scales linearly with respect to the graph size.

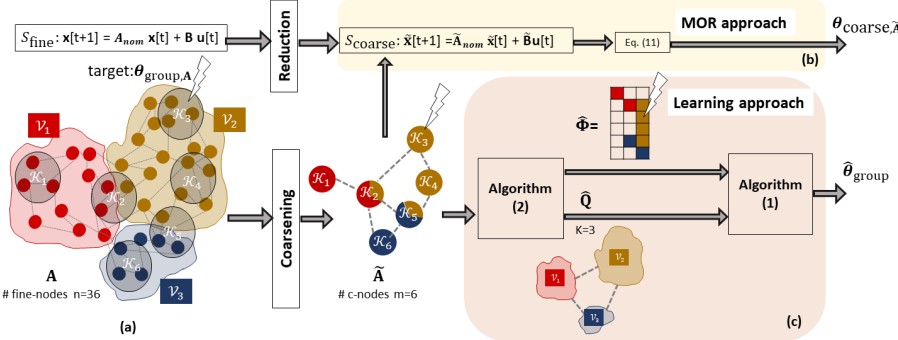

Figure 1: Schematic of MOR and learning-based approaches for estimating $\boldsymbol{\theta}_{\mathrm{group},\mathbf{A}}$. (a) For $\mathcal{G}_{\mathrm{fine}}$ consisting of $n = 36$ nodes, we have $K = 3$ communities ($\mathcal{V}_1, \mathcal{V}_2, \mathcal{V}_3$) with $m = 6$ coarse (c)-nodes ($\mathcal{K}_1, \ldots, \mathcal{K}_6$), each having a coverage size $r = 3$. The control node set is $\mathcal{K}_3$. (b) In MOR approach, we infer $\boldsymbol{\theta}_{\mathrm{group},\mathbf{A}}$ via reduced order dynamical system $\mathcal{S}_{\mathrm{coarse}}$. (c) Learning based approach capitalizes on mixed membership (MM) Algorithm 2 to estimate $\boldsymbol{\theta}_{\mathrm{group},\mathbf{A}}$ directly from $\widetilde{\mathbf{A}}$, thereby avoiding the need to do consider $\mathcal{S}_{\mathrm{coarse}}$.

**Remark 1.** *(Synchronization of community and coarsening): The coarsening operation is oblivious to the community structure in $\mathcal{G}_{\mathrm{fine}}$. Thus, the $i$-th c-node contains information about multiple communities if $\mathrm{supp}(\mathbf{W}_{i,:}) \cap \mathrm{supp}(\mathbf{P}_{i,:}) \neq \emptyset$ (the subscript denotes the $i$-th row), for $i \neq j$. Perfect synchronization: A special case where the intersection is non-empty only when $i = j$. This happens when all the communities have the same size and each c-node covers only one whole community.*

## 4 PROBLEM STATEMENT

Consider the LTI system on the network $\mathcal{G}_{\mathrm{fine}}$, defined by the symmetric adjacency matrix in $\mathbf{A}$ Eq. 3:

$$\mathcal{S}_{\mathrm{fine}} : \quad \mathbf{x}[t+1] = \mathbf{A}_{\mathrm{nom}}\mathbf{x}[t] + \mathbf{B}_{\mathcal{K}}\mathbf{u}[t], \tag{9}$$

where $\mathbf{A}_{\mathrm{nom}} \triangleq \frac{1}{c \cdot \mathrm{tr}(\mathbf{A})}\mathbf{A}$, $\mathbf{A} \in \mathbb{R}^{n \times n}$, and $\mathbf{B}_{\mathcal{K}} \in \mathbb{R}^{n \times n}$ selects a set of control nodes $\mathcal{K} \subset \mathcal{G}_{\mathrm{fine}}$. Depending on the controllability properties of $\mathcal{G}_{\mathrm{fine}}$ (see, Section 3.1), the inputs at these $|\mathcal{K}|$ control nodes may or may not effect all the states in $\mathbf{x}[k]$. Let $\mathcal{K}_i = \mathrm{supp}(\mathbf{w}_i)$, be the set (hereafter, group) of $r$ nodes coarsened by $\mathbf{w}_i$. The following assumption states that $\mathcal{G}_{\mathrm{fine}}$ is controllable from all $\mathcal{K}_i \subset \mathcal{G}_{\mathrm{fine}}$; see Remark 2.

**Assumption 4.** *Let* $\mathbf{B}_{\mathcal{K}_i} = \text{Diag}(\mathbf{w}_i^{\mathsf{T}})$. *For any* $i \in [m]$, *the Gramian* $\mathcal{C}(\mathbf{A}_{nom}, \mathbf{B}_{\mathcal{K}_i})$ *is full rank.*

As $\rho(\mathbf{A}_{\text{nom}}) < 1$ holds with high probability (see Lemma 1), it follows that $\mathcal{C}(\mathbf{A}_{\text{nom}}, \mathbf{B}_{\mathcal{K}_i})$ exits, and hence, the infinite time Gramian $\mathcal{C}(\mathbf{A}_{\text{nom}}, \mathbf{B}_{\mathcal{K}_i})$ exists. Assumption 4 ensures that average energy (see Section 3.1) is finite; however, average controllability need not be finite.

For $\mathcal{K}_i$-th control node set with input matrix $\mathbf{B}_{\mathcal{K}_i} = \text{Diag}(\mathbf{w}_i^{\mathsf{T}})$, associate the average controllabity measure: $\boldsymbol{\theta}_{\text{group},\mathbf{A}}^{(i)} \triangleq \text{tr}[\mathcal{C}(\mathbf{A}_{\text{nom}}, \mathbf{B}_{\mathcal{K}_i})]$, for all $i \in [m]$. Since $\mathcal{C}(\cdot)$ is a p.s.d matrix, it follows that $\boldsymbol{\theta}_{\text{group},\mathbf{A}}^{(i)} \geq 0$. Accordingly, define the group average controllability vector for $\mathcal{S}_{\text{fine}}$:

$$\boldsymbol{\theta}_{\text{group},\mathbf{A}} \triangleq \left( \boldsymbol{\theta}_{\text{group},\mathbf{A}}^{(1)}, \ldots, \boldsymbol{\theta}_{\text{group},\mathbf{A}}^{(m)} \right) \in \mathbb{R}^m, \tag{10}$$

which summarizes the average controllability measure (see Sec 3.1) for all control nodes sets $\{\mathcal{K}_1, \ldots, \mathcal{K}_m\}$. Thus, $\boldsymbol{\theta}_{\text{group},\mathbf{A}}$ helps infer (i) $\mathcal{K}_i s$ that drive the network to the desired target state with least control effort, and (ii) if such control node sets should have a community structure.

In this paper, using only the knowledge of $\widetilde{\mathbf{A}}$ in Eq. 5, we want to estimate the random vector $\boldsymbol{\theta}_{\text{group},\mathbf{A}}$ in Eq. 10. We consider two contrasting approaches: (i) the traditional model order reduction (MOR), where we rely upon the reduced order auxiliary system $\mathcal{S}_{\text{coarse}}$ (see Eq. 11) governed by $\widetilde{\mathbf{A}}$ to infer $\boldsymbol{\theta}_{\text{group},\mathbf{A}}$; and (ii) the learning based approach, where we directly estimate $\boldsymbol{\theta}_{\text{group},\mathbf{A}}$ using a clustering based mixed membership community learning algorithm; see Section 6. Fig. 1 provides a nice graphical illustration of our approaches. Broadly, our analysis highlights the role of community structure, coarsening process, and graph sparsity conditions on the performance of these both approaches. Our numerical simulations show that both approaches can outperform each other; however, learning based approach outperforms its counterpart in several parametric regions Finally, our main results in Sections 5 and 6 are probabilistic in nature because $\mathbf{A}$ is a random matrix.

**Remark 2.** *(Group nodes controllability) Assumption 4 demands that a group of nodes should be able to control $\mathcal{G}_{fine}$, which holds true for brain networks (Pasqualetti et al., 2019). This assumption is a very weaker condition than asking $\mathcal{G}_{fine}$ to be controllable from every single node. Moreover, if $\mathcal{G}_{fine}$ is not controllable from the control nodes, we can decompose the state-space of $\mathcal{G}_{fine}$ into controllable and uncontrollable sub-spaces (Chen, 1999), and adapt our analysis to the controllable sub-space.*

## 5 MOR APPROACH FOR GROUP AVERAGE CONTROLLABILITY

We provide a tight upper bound on the element wise error between $\boldsymbol{\theta}_{\text{group},\mathbf{A}}$ in Eq. 10 and $\boldsymbol{\theta}_{\text{coarse},\widetilde{\mathbf{A}}}$ in Eq. 13. The latter quantity is the group average controllability vector of the reduced order system:

$$\mathcal{S}_{\text{coarse}} : \ \widetilde{\mathbf{x}}[t+1] = \widetilde{\mathbf{A}}_{\text{nom}}\widetilde{\mathbf{x}}[t] + \widetilde{\mathbf{B}}\mathbf{u}[t] , \tag{11}$$

where $\widetilde{\mathbf{A}}_{\text{nom}} \triangleq \frac{1}{\tilde{c} \cdot \text{tr}(\widetilde{\mathbf{A}})} \widetilde{\mathbf{A}}$, $\widetilde{\mathbf{A}} = \mathbf{WAW}^{\mathsf{T}}$ is given by Eq. 5 and the normalization factor $\tilde{c} \cdot \text{tr}(\widetilde{\mathbf{A}})$ is used for stability purposes, where $\tilde{c} > 0$. Importantly, the state $\widetilde{\mathbf{x}}[t] \in \mathbb{R}^m$ is not a compression of the true state $\mathbf{x}[t]$ in $\mathcal{S}_{\text{fine}}$ and $m \ll n$ (hence the name MOR). Rather, $\widetilde{\mathbf{x}}[t]$ is a fictitious state that describes the dynamics of the (scaled) matrix $\widetilde{\mathbf{A}}$. This fictitious state is controlled by the input $\widetilde{\mathbf{B}}\mathbf{u}[t] \in \mathbb{R}^m$.

The following lemma states that $\mathcal{S}_{\text{fine}}$ and $\mathcal{S}_{\text{coarse}}$ are asymptotically stable with very high probability. Thus, $\boldsymbol{\theta}_{\text{group},\mathbf{A}}$ in 10 and $\boldsymbol{\theta}_{\text{coarse},\widetilde{\mathbf{A}}}$ in 13 are well defined. Let $c, \tilde{c}, \beta \geq 0$, and define the stability indices: $\kappa_{\text{nom}} := (c\beta \, \text{tr}(\bar{\mathbf{A}}) - \|\bar{\mathbf{A}}\|_\infty)/(n(1-c\beta)^2)$ and $\widetilde{\kappa}_{\text{nom}} := (\tilde{c}\beta \, \text{tr}(\bar{\widetilde{\mathbf{A}}}) - \|\bar{\widetilde{\mathbf{A}}}\|_\infty)/(m(1-r\tilde{c}\beta)^2)$, where $\bar{\mathbf{A}} \in \mathbb{R}^{n \times n}$ and $\bar{\widetilde{\mathbf{A}}} \in \mathbb{R}^{m \times m}$ are given in Eq. 6 and $r$ is the coverage size.

**Lemma 1.** *(Probabilistic stability of $\mathcal{S}_{fine}$ and $\mathcal{S}_{\text{coarse}}$): Suppose that the stability indices $\kappa_{nom}$ and $\widetilde{\kappa}_{nom}$ are strictly positive for some constants $c, \tilde{c} > 0$, and $0 \leq \beta < 1$. Then,*

$$\mathbb{P}\left[\rho(\mathbf{A}_{nom}) \geq \beta\right] \leq n \exp\left(-2\kappa_{nom}\right) \quad and \quad \mathbb{P}[\rho(\widetilde{\mathbf{A}}_{nom}) \geq \beta] \leq m \exp\left(-2\widetilde{\kappa}_{nom}\right). \tag{12}$$

For $m = n$, we have $r = 1$, and hence, the probabilistic inequalities in Eq. 12 coincide with each other. Further, the higher $\kappa_{nom}$ and $\widetilde{\kappa}_{nom}$, the higher the chance that $\mathcal{S}_{\text{fine}}$ and $\mathcal{S}_{\text{coarse}}$ are stable. Interestingly, $\kappa_{nom}$ and $\widetilde{\kappa}_{nom}$ do not explicitly scale with $n$ and $m$. In fact, $\kappa_{nom} \geq \kappa_{nom,lb}$, where

$\kappa_{nom,\,lb} := (c\beta\, c_{\min}\,\text{tr}(\mathbf{Q}) - c_{\max}\|\mathbf{Q}\|_\infty)/(1 - c\beta)^2$ is independent of $n$. Thus, stability is not guaranteed for larger networks modeled using $\mathcal{S}_{\text{fine}}$ with $c \cdot \text{tr}\mathbf{A}$ normalization.

Instead, $\kappa_{nom,\,lb} \to \infty$ as $c\beta \to 1$, thereby $\mathbb{P}\left[\rho(\mathbf{A}_{nom}) \geq \beta\right] \leq n\exp(-2\kappa_{nom,\,lb}) \to 0$. So, under what conditions $\kappa_{\text{nom}} > 0$ for $c\beta = 1$? One such condition is $c\beta \geq c_{\max}\|\mathbf{Q}\|_\infty/(c_{\min}\,\text{tr}(\mathbf{Q}))$. Let $c_{\max} = c_{\min}$. For diagonally dominant probability matrix $\mathbf{Q}$ (i.e., the community structure is assortative—more in-community edges than across-community edgess), $\text{tr}(\mathbf{Q}) \geq \|\mathbf{Q}\|_\infty$, and hence, we can choose $c$ and $\beta$ such that $c\beta = 1$. However, non diagonally dominant matrices can satisfy $\text{tr}(\mathbf{Q}) \geq \|\mathbf{Q}\|_\infty$. For example, a symmetric matrix $\mathbf{Q} \in \mathbb{R}^{3\times3}$, with $\mathbf{Q}_{ii} = 0.2$, $\mathbf{Q}_{12} = 0.25$, and $\mathbf{Q}_{13} = 0.01$ is not diagonally dominant because $\mathbf{Q}_{ii} \leq \sum_{j\neq i}\mathbf{Q}_{ij}$, for all $i$, but $\text{tr}(\mathbf{Q}) \geq \|\mathbf{Q}\|_\infty$.

We now bound the difference between $\boldsymbol{\theta}_{\text{group},\mathbf{A}}$ in 10 and $\boldsymbol{\theta}_{\text{coarse},\widetilde{\mathbf{A}}}$ in 13. Let $\widetilde{\mathbf{A}}_{\text{nom}} \triangleq \widetilde{\mathbf{A}}/(\tilde{c}\cdot\text{tr}(\widetilde{\mathbf{A}}))$ and $\widetilde{\mathbf{B}}_i = r\mathbf{W}\text{diag}(\mathbf{w}_i^\mathsf{T})$—the coarsened input matrix. Let the average controllability for the $i$-th c-node be $\boldsymbol{\theta}^{(i)}_{\text{group},\widetilde{\mathbf{A}}} \triangleq \text{tr}[\mathcal{C}(\widetilde{\mathbf{A}}_{\text{nom}}, \widetilde{\mathbf{B}}_i)] \geq 0$. Define the average controllability vector for $\mathcal{S}_{\text{coarse}}$:

$$\boldsymbol{\theta}_{\text{coarse},\widetilde{\mathbf{A}}} \triangleq \left(\boldsymbol{\theta}^{(1)}_{\text{coarse},\widetilde{\mathbf{A}}}, \dots, \boldsymbol{\theta}^{(m)}_{\text{coarse},\widetilde{\mathbf{A}}}\right) \in \mathbb{R}^m. \tag{13}$$

In what follows, when the synchronization holds, we show that $\boldsymbol{\theta}_{\text{coarse},\widetilde{\mathbf{A}}}$ associated with $\mathcal{S}_{\text{coarse}}$ can well approximate $\boldsymbol{\theta}_{\text{group},\mathbf{A}}$ associated with $\mathcal{S}_{\text{fine}}$. Define the error metric:

$$\Delta_i(\mathbf{A}, \widetilde{\mathbf{A}}) \triangleq \left| \frac{r\boldsymbol{\theta}^{(i)}_{\text{group},\mathbf{A}} - 1}{\sum_{i=1}^m [r\boldsymbol{\theta}^{(i)}_{\text{group},\mathbf{A}} - 1]} - \frac{\boldsymbol{\theta}^{(i)}_{\text{coarse},\widetilde{\mathbf{A}}} - 1}{\sum_{i=1}^m [\boldsymbol{\theta}^{(i)}_{\text{coarse},\widetilde{\mathbf{A}}} - 1]} \right|, \quad \text{for all } i \in [m]. \tag{14}$$

The proposed error metric $\Delta_i(\mathbf{A}, \widetilde{\mathbf{A}})$ allows us to do a fair shifting- and scaling-free comparison between the vectors $\boldsymbol{\theta}_{\text{group},\mathbf{A}}$ and $\boldsymbol{\theta}_{\text{coarse},\widetilde{\mathbf{A}}}$. The shift factor "-1" accounts for the inherent "+1" shift in the average controllability definition and the scale factor $r$ discounts the $1/r$ factor in $\boldsymbol{\theta}_{\text{group},\mathbf{A}}$. Akin to $\Delta_i(\mathbf{A}, \widetilde{\mathbf{A}})$ in Eq. 14, define $\Delta_i(\bar{\mathbf{A}}, \widetilde{\bar{\mathbf{A}}})$ associated with the expected quantities $\bar{\mathbf{A}}$ and $\widetilde{\bar{\mathbf{A}}}$.

**Theorem 1.** *(Element wise bound on $\boldsymbol{\theta}_{\text{group},\mathbf{A}} - \boldsymbol{\theta}_{\text{coarse},\widetilde{\mathbf{A}}}$): Let $\Delta_i(\mathbf{A}, \widetilde{\mathbf{A}})$ and $\Delta_i(\bar{\mathbf{A}}, \widetilde{\bar{\mathbf{A}}})$ be defined as above. Then, under the assumptions in Lemma 1, $\Delta_i(\mathbf{A}, \widetilde{\mathbf{A}}) \leq \Delta_i(\bar{\mathbf{A}}, \widetilde{\bar{\mathbf{A}}}) + \mathcal{O}\left(\frac{1}{\rho_n m} + \frac{r^2}{\sqrt{m}\rho_n^3}\right)$ with probability at least $1 - 6\exp(-2\kappa(\mathbf{Q}^{(c)}, \mathbf{D}, \nu, m, n, \rho_n))$, where, for constants $0 < \delta, \zeta < 1$,*

$$\kappa(\mathbf{Q}^{(c)}, \mathbf{D}, \nu, m, n, \rho_n) = \min\left\{ n\rho_n^2(\text{tr}(\mathbf{D}\mathbf{Q}^{(c)})\zeta)^2, \ m(\rho_n\nu^2\text{tr}(\mathbf{Q}^{(c)})\delta)^2 \right.$$

$$\left. \rho_n^2 mn(\tilde{c}_{min}c_{min}\|\mathbf{Q}^{(c)}\|_{1,1}\zeta)^2, \ \rho_n^2 m^2(\tilde{c}_{min}^2\|\mathbf{Q}^{(c)}\|_{1,1}\delta)^2 \right\},$$

*with $\nu$, $\mathbf{D}$, and $\mathbf{Q}^{(c)}$ given by Eq. 7, Eq. 4, and Eq. 8, and $\tilde{c}_{min}$ is defined in Assumption 3.*

Theorem 1 says that $\Delta_i(\mathbf{A}, \widetilde{\mathbf{A}}) \approx \mathcal{O}\left(\frac{1}{\rho_n m} + \frac{r^2}{\sqrt{m}\rho_n^3}\right)$ provided $\Delta_i(\bar{\mathbf{A}}, \widetilde{\bar{\mathbf{A}}})$ is small. Thus, for fixed $n$, MOR based estimate $\boldsymbol{\theta}_{\text{coarse},\widetilde{\mathbf{A}}}$ approximates $\boldsymbol{\theta}_{\text{group},\mathbf{A}}$ if the graph density $\rho_n$ or number of c-nodes is large, which we validate using numerical simulations as well. It can be shown that the bias term $\Delta_i(\bar{\mathbf{A}}, \widetilde{\bar{\mathbf{A}}})$ is exactly zero if the communities are synchronized with the coarsening process (see Remark 1) and that $\mathbf{Q}_{ii} = p > 0$ and $\mathbf{Q}_{ij} = q > 0$, for $i \neq j$.

## 6   LEARNING APPROACH FOR GROUP AVERAGE CONTROLLABILITY

We present our learning based approach to estimate $\boldsymbol{\theta}_{\text{group},\mathbf{A}}$ in Eq. 10. Unlike the MOR based approach that relies on $\mathcal{S}_{\text{fine}}$, we directly estimate elements in $\boldsymbol{\theta}_{\text{group},\mathbf{A}}$ based on the popular mixed membership (MM) community learning algorithms (Mao et al., 2020; Huang et al., 2019; Mao et al., 2018; 2017; Aicher et al., 2015). Specifically, we work with the MM algorithm by (Mao et al., 2020) which is not only numerically efficient but also has strong theoretical guarantees.

**Lemma 2.** *Let $\boldsymbol{\theta}_{\text{group},\bar{\mathbf{A}}}$ be obtained by replacing $\mathbf{A}$ with the expected matrix $\bar{\mathbf{A}}$ in $\boldsymbol{\theta}_{\text{group},\mathbf{A}}$, given by Eq. 10. Let $\mathbf{P}$ and $\boldsymbol{\Phi}$ be as in Eq. 4 and Eq. 6. Suppose that Assumption 1 hold. Then,*

$$\boldsymbol{\theta}_{\text{group},\bar{\mathbf{A}}} = \left(\mathbf{1}_m + d\boldsymbol{\Phi}\text{diag}(\boldsymbol{\Upsilon})\right)/r, \tag{15}$$

*where $d = 1/(n\text{tr}[\mathbf{D}\mathbf{Q}^{(c)}])$, $\mathbf{D} = (1/n)\mathbf{P}\mathbf{P}^{\mathsf{T}}$, and $\mathbf{Q}^{(c)}$ is given by Eq. 8, and*

$$\Upsilon \triangleq (nd)\mathbf{Q}^{(c)}\mathbf{D}\mathbf{Q}^{(c)}[\mathbf{I} - ((nd)\mathbf{D}\mathbf{Q}^{(c)})^2]^{-1}. \tag{16}$$

Lemma 2 gives us a formula to compute the group average controllability vector associated with the expected matrix $\bar{\mathbf{A}}$ (see Supplemental material for complementary explanation and interpretation). Theorem 3 (see Appendix) shows that $\Delta_i(\mathbf{A}, \widetilde{\mathbf{A}}) \leq \epsilon$, for arbitrary $\epsilon > 0$, hold with high probability. In view of this fact, we propose our candidate estimator as

$$\hat{\boldsymbol{\theta}}_{\text{group}} \triangleq \mathbf{1}_m + \hat{\boldsymbol{\Phi}}\text{diag}(\hat{\Upsilon}), \tag{17}$$

where $\hat{\Upsilon} \triangleq \frac{\hat{\mathbf{Q}}\hat{\mathbf{D}}\hat{\mathbf{Q}}}{\text{tr}(\hat{\mathbf{D}}\hat{\mathbf{Q}})}(\mathbf{I} - (\frac{\hat{\mathbf{D}}\hat{\mathbf{Q}}}{\text{tr}(\hat{\mathbf{D}}\hat{\mathbf{Q}})})^2)^{-1}$. The hatted quantities $\hat{\boldsymbol{\Phi}}$ and $\hat{\mathbf{Q}}$ are obtained from Algorithm[5] 2, which takes as input $\widetilde{\mathbf{A}}$ and number of communities $K$. Instead, we obtain $\hat{\mathbf{D}}$ from Algorithm 1. Importantly, $\hat{\boldsymbol{\theta}}_{\text{group}}$ in Eq. 17 is obtained from coarsened matrix $\widetilde{\mathbf{A}}$ but not the fine scale matrix $\mathbf{A}$.

**Theorem 2.** *(**Component wise error bound between $\hat{\boldsymbol{\theta}}_{\text{group}}$ and $\boldsymbol{\theta}_{\text{group},\mathbf{A}}$**): Suppose that there exist constants $\tilde{c}_{min}$ and $\tilde{c}_{max}$ that satisfy Assumption 3. Then, under the hypotheses stated in Lemma 1,*

$$\underbrace{\left| \frac{r\boldsymbol{\theta}_{\text{group},\mathbf{A}}^{(i)} - 1}{\sum_{i=1}^m (r\boldsymbol{\theta}_{\text{group},\mathbf{A}}^{(i)} - 1)} - \frac{\hat{\boldsymbol{\theta}}_{\text{group}}^{(i)} - 1}{\sum_{i=1}^m (\hat{\boldsymbol{\theta}}_{\text{group}}^{(i)} - 1)} \right|}_{\triangleq \hat{\Delta}_i(\widetilde{\mathbf{A}})} = \mathcal{O}\left( \frac{1}{m}[\frac{1}{\rho_n} + ||\mathbf{E}_{\boldsymbol{\Phi}}||_{max} + ||\mathbf{E}_{\mathbf{Q}}||_{max} + ||\mathbf{E}_{\mathbf{D}}||_{max}] \right)$$

*holds with probability at least $1 - 3\exp(-2\hat{\kappa}(\mathbf{Q}^{(c)}, \mathbf{D}, m, n, \rho_n))$, where $\mathbf{E}_{\boldsymbol{\Phi}} \triangleq \hat{\boldsymbol{\Phi}} - \boldsymbol{\Phi}$, $\mathbf{E}_{\mathbf{Q}} \triangleq \hat{\mathbf{Q}} - \mathbf{Q}$, and $\mathbf{E}_{\mathbf{D}} \triangleq \hat{\mathbf{D}} - \mathbf{D}$ are the error matrices. Further, for a constant $0 < \zeta < 1$, the exponent $\hat{\kappa}(\mathbf{Q}^{(c)}, \mathbf{D}, m, n, \rho_n) = \min\left\{ n\rho_n^2(\text{tr}(\mathbf{D}\mathbf{Q}^{(c)})\zeta)^2, \ mn\rho_n^2(\tilde{c}_{min}c_{min}||\mathbf{Q}^{(c)}||_{1,1}\zeta)^2 \right\}.$*

Theorem 2 suggests that for sufficiently large $m$, the estimate $\hat{\boldsymbol{\theta}}_{\text{group}}$ approximates $\boldsymbol{\theta}_{\text{group},\mathbf{A}}$ to an arbitrary precision if: (a) the graph is dense enough (larger $\rho_n$), (b) the coarse community membership matrix is well estimated (smaller[6] $||\mathbf{E}_{\boldsymbol{\Phi}}||_{max}$), (c) the cross-community probability estimation do not suffer from high error (smaller $||\mathbf{E}_{\mathbf{Q}}||_{max}$), and the relative community sizes estimated from the coarse graph are close the ones in the fine graph (smaller $||\mathbf{E}_{\mathbf{D}}||$). The result of Theorem 2 is important because one can directly infer the most influential control node set $\mathcal{K}_i$ (one with high average controllability) via the most influential $i$-the c-nodes, and vice versa.

---

**Algorithm 1:** Direct Inference of the Group Average Controllability

---

**Require:** estimates $\hat{\boldsymbol{\Phi}}$ and $\hat{\mathbf{Q}}$ from Algorithm 2, and the number of communities $K$

1: compute $\hat{\mathbf{D}} = \text{diag}(\frac{\hat{\boldsymbol{\Phi}}^{\mathsf{T}}\mathbf{1}_m}{\mathbf{1}_K\hat{\boldsymbol{\Phi}}^{\mathsf{T}}\mathbf{1}_m})$

2: **return** $\hat{\boldsymbol{\theta}}_{\text{group}} = \mathbf{1}_m + \hat{\boldsymbol{\Phi}}\text{diag}\left( \frac{\hat{\mathbf{Q}}\hat{\mathbf{D}}\hat{\mathbf{Q}}}{\text{tr}(\hat{\mathbf{D}}\hat{\mathbf{Q}})}(\mathbf{I} - (\frac{\hat{\mathbf{D}}\hat{\mathbf{Q}}}{\text{tr}(\hat{\mathbf{D}}\hat{\mathbf{Q}})})^2)^{-1}) \right)$

---

**Algorithm 2:** Mixed Membership Community Estimation Algorithm (Mao et al., 2020)

---

**Require:** Coarse adjacency matrix $\widetilde{\mathbf{A}}$, number of communities $K$

1: compute the highest $K$ eigen-decomposition of $\widetilde{\mathbf{A}}$ as $\hat{V}\hat{\Lambda}\hat{V}^{\mathsf{T}}$ and set $\mathcal{S}_{\text{pruned}} = \text{Prune}(\hat{V})$

2: set $X = \hat{V}([m] \setminus \mathcal{S}_{\text{p.runed}}, :)$ and compute $\mathcal{S}_{\text{pure}} = \text{Successive Projection Algorithm}(X^{\mathsf{T}})$

3: set $X_{\text{pure}} = X(\mathcal{S}_{\text{pure}}, :)$ and compute un-normalized $\hat{\boldsymbol{\Phi}}^{\text{un-nom}} = \hat{V}X_{\text{pure}}^{-1}$

4: $\hat{\boldsymbol{\Phi}}_{ik}^{\text{un-nom}} \leftarrow 0$ if $\hat{\boldsymbol{\Phi}}_{ik}^{\text{un-nom}} < e^{-12}, \forall i \in [m], k \in [K]$

5: **return** $\hat{\boldsymbol{\Phi}} = \text{Diag}^{-1}(\hat{\boldsymbol{\Phi}}^{\text{un-nom}}\mathbf{1}_K)\hat{\boldsymbol{\Phi}}^{\text{un-nom}}$ and $\hat{\mathbf{Q}} = X_{\text{pure}}\hat{\Lambda}X_{\text{pure}}^{\mathsf{T}}$

---

[5]This MM algorithm adapted from Mao et al. (2020) is a type of spectral clustering method that first performs eigen decomposition of $\widetilde{\mathbf{A}}$ to find the overlapping membership ($\boldsymbol{\Phi}_{ik}$) of the fine nodes. A pruning step is also included (see steps 4 and 5 in Algorithm 2) to speed up the algorithm performance.

[6] (Mao et al., 2020) showed that $||\mathbf{E}_{\boldsymbol{\Phi}}||_{max}$ and $||\mathbf{E}_{\mathbf{Q}}||_{max}$ stated in Theorem 2 approach zero under some conditions, as $m \to \infty$. But these conditions might not be applicable to our setup because of our coarsening operation. However, our simulations show that $\hat{\Delta}_i(\widetilde{\mathbf{A}})$ decreases with $m$. The behavior of $||\mathbf{E}_{\boldsymbol{\Phi}}||_{max}$ and $||\mathbf{E}_{\mathbf{Q}}||_{max}$ with respect to graph scaling is left for future work.

## 7 SIMULATIONS

We validate our theoretical results by plotting[7] the errors $\frac{1}{m}\sum_{i=1}^{m}\Delta_i(\mathbf{A},\widetilde{\mathbf{A}})$ and $\frac{1}{m}\sum_{i=1}^{m}\widehat{\Delta}_i(\widetilde{\mathbf{A}})$ and show that these errors are comparable to the bounds we obtained in Theorems 1 and 2. We generate $\mathbf{A} \sim \text{SBM}(n, \mathbf{Q}, \mathbf{p})$ and then determine $\widetilde{\mathbf{A}} = \mathbf{WAW}^{\mathsf{T}}$ (see Supplemental material for generating $\mathbf{W}$). We set number of fine nodes $n = 5000$, the overlap parameter $\eta = 0.1$, and the number of communities $K = 5$. Finally, for $\mathbf{Q} \in \mathbb{R}^{K \times K}$, we set $\mathbf{Q}_{kk} = p = 0.05$ and $\mathbf{Q}_{kk'} = q = 0.01$ (for $k \neq k'$). If not specified, the number of c-nodes $m = 100$ and coverage size per c-node $r = 4$. Fig. 2(a)-2(d) illustrate the qualitative behavior of the errors with respect to changes in $m$, $\rho_n$, and the degree of (non-)synchronization in coarse nodes (i.e., $\eta$), and $r$. To fairly compare these errors, we also consider a base line error: $\sum_{i=1}^{m}\left|\frac{\mu_i - 1}{\sum_{i=1}^{m}(\mu_i - 1)} - \frac{r\boldsymbol{\theta}_{\text{group},\mathbf{A}}^{(i)}{}^{-1}}{\sum_{i=1}^{m}(r\boldsymbol{\theta}_{\text{group},\mathbf{A}}^{(i)}{}^{-1})}\right|/m$, where $\mu \in [1,2]^m$ contains i.i.d. uniform random variable drawn independently of $\widetilde{\mathbf{A}}$.

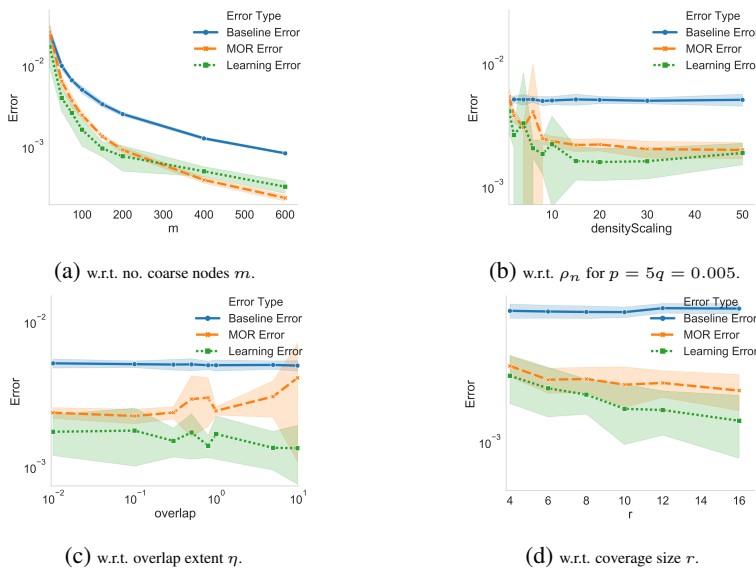

(a) w.r.t. no. coarse nodes $m$.

(b) w.r.t. $\rho_n$ for $p = 5q = 0.005$.

(c) w.r.t. overlap extent $\eta$.

(d) w.r.t. coverage size $r$.

Figure 2: Estimation error based on MOR and Learning approaches. MOR Error=$\sum_{i=1}^{m}\Delta_i(\mathbf{A},\widetilde{\mathbf{A}})/m$ and Learning Error=$\sum_{i=1}^{m}\widehat{\Delta}_i(\widetilde{\mathbf{A}})/m$. The shaded region in the figures represent one standard deviation computed for 20 independent realizations. We make the following observations. First, both the learning and MOR based errors are consistently better than the random baseline. Second, the learning based approach has consistently smaller error than that of the MOR approach for large parametric regimes. Third, (a) shows that all errors monotonically decrease as $m$ increases. This is consistent with our bounds in Theorems 1 and 2. Fourth, (b) shows that errors decrease as $\rho_n$ increases. This is expected because larger values of $\rho_n$ result in more distant in- and cross-community edge densities. This makes community representation extraction and controllability estimation easier. Fifth, (c) demonstrates the higher tolerance of the Learning approach to situations whenin coarse measurements are less synchronized, in comparison to the MOR method. Finally, (d) shows that error decreases with $r$. This should be the case as larger $r$ means more fine nodes are sampled during coarsening.

## 8 CONCLUSION AND FUTURE WORK

We introduced a learning-based framework that exploits the power of community-based representation learning to infer average controllability of fine graphs from coarse summary data. We compared the performance of this approach with that of MOR approach. For both these methods, we derived high probability error bounds on the deviation between the error estimate and ground truth, and validated the theory with numerical simulations. Our results highlight the role of fine- and coarse-network sizes, graph density, and community synchronization bias (see Remark 1 in modulating the estimation errors. Interestingly, for the latter approach, we show that the estimation error decreases with network size albeit the synchronization bias, which is not the case with the MOR-based approach. For future, we plan to implement our theory to study the role of coarsening, community structures , and synchronization aspects on the controllability of brain networks.

---

[7]The Python code to reproduce the results is attached to the submitted file.

## 9 ETHICS STATEMENT

Although our work mainly takes a theoretical perspective to the controllability of coarse graphs motivated by therapeutic neuroscience applications, our results can potentially involve negative impacts if employed in other applications. For instance, identifying the most influential groups of nodes (or equivalently individuals) in a social network as a result of estimating the network group average controllability, may motivate manipulative actions; i.e. the most influential node group may be selected for control, in order to steer the whole network towards an unethical goal (like manipulating individuals in a social network to vote in favour of a particular election candidate). The negative impact may also result in bias against less-influential nodes, as they will be ignored when it comes to the selection of node groups for control actuation.

## 10 REPRODUCIBILITY STATEMENT

All the results presented in this paper are reproducible. The theoretical findings are annotated and step-by-step elaborated in the appendix. The data generation process and the parameter values used for numerical simulations are fully explained. In addition, the Python code from which the simulation figures are generated is attached to the submission files. The code is also on Github and the repository will go public upon submission acceptance.

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

## A APPENDIX

In this appendix, we first provide further detailed explanation for the numerical simulations that was missing due to space constraints. Next we have a remark that adds complementary interpretation of the group controllability formula estimated from the learning approach. Finally, we provide proofs for all results stated in our paper.

### A.1 MORE ON NUMERICAL SIMULATIONS: HOW TO GENERATE $\mathbf{W}$

For a realization of $\widetilde{\mathbf{A}}$, we obtain the support of each row of $\mathbf{W}$ independently by first generating a random vector $\pi \in \{\eta\}^K$ using a Dirichlet distribution; larger positive $\eta$ result in greater overlap and more communities. For each community $k$ where $\pi_k > 0$, $\lfloor r \cdot \pi_k \rfloor$ fine nodes are randomly chosen from $\mathcal{V}_k^{\text{non-sel}}$ (i.e., non-selected fine indices in community $k$) and set as the support of $\mathbf{w}_i$. The chosen indices are removed from $\mathcal{V}_k^{\text{non-sel}}$. This process continues until the support of all $\mathbf{w}_i$s are selected.

### A.2 MORE ON THE ESTIMATED GROUP CONTROLLABILITY USING THE LEARNING APPROACH

**Remark 3.** *(Synchronization versus controllability) Recall that* $\mathbf{\Phi}_{i,k} = |\{v : v \in \mathcal{V}_k \cap supp(\mathbf{w}_i)\}|/r$ *(for all* $k \in [K]$, $i \in [m]$*) is the fraction of c-node $i$'s overlap with community* $k$. *Thus, from Eq. 15, it follows that* $\boldsymbol{\theta}_{\text{group},\bar{\mathbf{A}}}^{(i)} \propto \sum_{k \in [K]} \mathbf{\Phi}_{i,k} \Upsilon_{kk}$. *In view of this observation and Lemma 2, we observe that c-nodes that have the largest overlap with communities of strongest* $\Upsilon_{kk}$ *are the most controllable.*

**Additional notation**: For $n \times m$ dimensional real matrix $\mathbf{M}$, denote $\|\mathbf{M}\|_F = \sqrt{\sum_u \sum_v \mathbf{M}_{uv}^2} = \sqrt{\text{tr}(\mathbf{M}^\mathsf{T}\mathbf{M})}$. For a symmetric matrix $\mathbf{M}$, denote $\lambda_{\max}(\mathbf{M})$ to be the maximum eigenvalue. $\text{diag}(\mathbf{M}) = [\mathbf{M}_{11}, \ldots, \mathbf{M}_{nn}]^\mathsf{T} \in \mathbb{R}^n$, and $\text{Diag}(\mathbf{M})$ sets the off-diagonal entries of $\mathbf{M}$ to zero.

**Useful matrix norm bounds**:

1. Let $\mathbf{Z} = \mathbf{XY}$. Then, $\|\mathbf{Z}\|_F \leq \|\mathbf{X}\|_2 \|\mathbf{Y}\|_F \leq \|\mathbf{X}\|_F \|\mathbf{Y}\|_F$.
2. *Cauchy-Schwartz inequality*: $\text{tr}(\mathbf{X}^\mathsf{T}\mathbf{Y}) \leq \|\mathbf{X}\|_F \|\mathbf{Y}\|_F$.
3. For any norm: If $\|\mathbf{M}\| < 1 \Rightarrow \|(\mathbf{I} - \mathbf{M})^{-1}\| < \frac{1}{1 - \|\mathbf{M}\|}$ (Keinert).
4. $\|\mathbf{M}\|_\infty \leq \sqrt{n}\|\mathbf{M}\|_2$
5. $\|\mathbf{M}\|_1 \leq \sqrt{n}\|\mathbf{M}\|_2$
6. For any $m \times n$ matrix $\mathbf{M}$, we have $\|\mathbf{M}\|_2 \leq \sqrt{m}\|\mathbf{M}\|_\infty$.
7. $\|\mathbf{MXY}\|_{\max} \leq \|\mathbf{M}\|_\infty \|\mathbf{X}\|_{\max} \|\mathbf{Y}\|_1$
8. $\|\mathbf{M}^2 - \mathbf{M}'^2\| \leq \|\mathbf{M} - \mathbf{M}'\|(\|\mathbf{M}\| + \|\mathbf{M}'\|)$

    *Proof.* $\|\mathbf{M}^2 - \mathbf{M}'^2\| = \|\mathbf{M}^2 - \mathbf{MM}' + \mathbf{MM}' - \mathbf{M}'^2\| = \|\mathbf{M}(\mathbf{M} - \mathbf{M}') + (\mathbf{M} - \mathbf{M}')\mathbf{M}'\| \leq \|\mathbf{M} - \mathbf{M}'\|(\|\mathbf{M}\| + \|\mathbf{M}'\|)$ □

9. $\mathbf{M}^{-1} - \mathbf{M}'^{-1} = \mathbf{M}^{-1}(\mathbf{M}' - \mathbf{M})\mathbf{M}'^{-1}$

**Lemma 3.** *(Lower-Bound Probability of Joint Events:)* *For the intersection of two events* $\mho_1$ *and* $\mho_2$ *we have:* $\mathbb{P}\{\mho_1 \cap \mho_2\} \geq \mathbb{P}\{\mho_1\} + \mathbb{P}\{\mho_2\} - 1$.

### A.3 PROOFS FOR SECTION 5: MOR APPROACH FOR GROUP AVERAGE CONTROLLABILITY

#### A.3.1 PROOF OF LEMMA 1

*Proof.* We prove only the left inequality in Eq. 12. The right inequality in Eq. 12 can be proved using similar steps, and the details are omitted. Because $\mathbf{A}_{\text{nom}} = \mathbf{A}/(c \cdot \text{tr}(\mathbf{A}))$ is symmetric, it follows that $\rho(\mathbf{A}_{\text{nom}}) = \|\mathbf{A}_{\text{nom}}\|_2$. From this observation and the fact that $\|\mathbf{A}_{\text{nom}}\|_2 \leq \|\mathbf{A}_{\text{nom}}\|_\infty$, we have

$$\mathbb{P}[\rho(\mathbf{A}_{\text{nom}}) \geq \beta] = \mathbb{P}[\|\mathbf{A}_{\text{nom}}\|_2 \geq \beta] \leq \mathbb{P}[\|\mathbf{A}_{\text{nom}}\|_\infty \geq \beta]$$
$$= \mathbb{P}[\|\mathbf{A}\|_\infty - c\beta\text{tr}(\mathbf{A}) \geq 0]. \qquad (18)$$

Let $\mathbf{F} = \|\mathbf{A}\|_\infty - c\beta \mathrm{tr}(\mathbf{A})$ and note that $\mathbb{E}[\mathbf{F}] = \mathbb{E}\|\mathbf{A}\|_\infty - c\beta \mathrm{tr}(\bar{\mathbf{A}}) \geq \|\bar{\mathbf{A}}\|_\infty - c\beta \mathrm{tr}(\bar{\mathbf{A}})$. The inequality follows because $\|\mathbf{A}\|_\infty = \max_{i \in [n]} \sum_{j=1}^n |a_{ij}|$ and that $\mathbb{E}[\max\{X_1, \ldots, X_t\}] \geq \max\{\mathbb{E}[X_1], \ldots, \mathbb{E}[X_t]\}$. From these observations, inequality in Eq. 18 can be further bounded as

$$\begin{aligned}
\mathbb{P}\left[\mathbf{F} \geq 0\right] = \mathbb{P}\left[\mathbf{F} - \mathbb{E}[\mathbf{F}] \geq -\mathbb{E}[\mathbf{F}]\right] \\
\leq \mathbb{P}\left[\mathbf{F} - \mathbb{E}[\mathbf{F}] \geq c\beta \mathrm{tr}(\bar{\mathbf{A}}) - \|\bar{\mathbf{A}}\|_\infty\right].
\end{aligned} \tag{19}$$

We show that $\mathbf{F}$ is a sub-Gaussian random variable and then bound the right term in Eq. 19 using the well-known concentration inequality results. Rewrite $\mathbf{F}$ as follows

$$\begin{aligned}
\mathbf{F} &= \max\left\{\sum_{j=1}^n |a_{1j}|, \sum_{j=1}^n |a_{2j}|, \ldots, \sum_{j=1}^n |a_{nj}|\right\} - c\beta \mathrm{tr}(\mathbf{A}) \\
&= \max\left\{\sum_{j=1}^n |a_{1j}| - c\beta \mathrm{tr}(\mathbf{A}), \sum_{j=1}^n |a_{2j}| - c\beta \mathrm{tr}(\mathbf{A}), \ldots, \sum_{j=1}^n |a_{nj}| - c\beta \mathrm{tr}(\mathbf{A})\right\} \\
&= \max\left\{\sum_{j=1}^n (a_{1j} - c\beta a_{jj}), \sum_{j=1}^n (a_{2j} - c\beta a_{jj}), \ldots, \sum_{j=1}^n (a_{nj} - c\beta a_{jj})\right\}.
\end{aligned} \tag{20}$$

In the last equality, we drop the absolute values because $a_{ij} \in [0, 1]$. From the latter fact, we also note that $(a_{kl} - a_{kk}) \in [-c\beta, 1]$. Thus, for all $k \neq l$, $(a_{kl} - a_{kk})$ is bounded, and hence, sub-Gaussian with parameter at most $\sqrt{1 + (c\beta)^2}/2$. Instead, for $k = l$, $(a_{kl} - a_{kk})$ is sub-Gaussian with parameter at most $|1 - c\beta|/2$. Finally, from Definition 1, notice that each term in the summation $\sum_{j=1}^n (a_{kj} - c\beta a_{jj})$ is independent. From these facts and the linearity of sub-Gaussians, we note that, for all $k \in [n]$, summand $\sum_{j=1}^n (a_{kj} - c\beta a_{jj})$ is sub-Gaussian with parameter at most:

$$\sigma = \frac{1}{2}\sqrt{(n-1)(1 + (c\beta)^2) + (1 - c\beta)^2}.$$

Putting all these pieces together in conjunction with the facts that maxima of sub-Gaussians concentrates near its expectation and $c\beta \mathrm{tr}(\bar{\mathbf{A}}) - \|\bar{\mathbf{A}}\|_\infty \geq 0$ (by assumption), from Eq. 19, we have

$$\begin{aligned}
\mathbb{P}\left[\mathbf{F} - \mathbb{E}[\mathbf{F}] \geq c\beta \mathrm{tr}(\bar{\mathbf{A}}) - \|\bar{\mathbf{A}}\|_\infty\right] &\leq n \exp\left(-(c\beta \mathrm{tr}(\bar{\mathbf{A}}) - \|\bar{\mathbf{A}}\|_\infty)^2/2\sigma^2\right) \\
&\leq n \exp\left(-2(c\beta \mathrm{tr}(\bar{\mathbf{A}}) - \|\bar{\mathbf{A}}\|_\infty)^2/(n(1 - c\beta)^2)\right).
\end{aligned} \tag{21}$$

The last inequality follows because $2\sigma^2 \geq n/2(1 - c\beta)^2$. The left inequality in Eq. 12 follows by combining inequalities in Eq. 18 and Eq. 21. The proof is now complete. $\quad\square$

### A.3.2 Lemma 4: Upper and Lower bounds of the trace of the fine- and coarse-scale matrices

**Lemma 4.** *(Upper and Lower bounds of the trace of the fine- and coarse-scale matrices). For constants $0 < \delta, \zeta < 1$:*

$$\begin{aligned}
\mathbb{P}\left[n\rho_n \zeta_{(l)} \leq \mathrm{tr}(\mathbf{A}) \leq n\rho_n \zeta_{(u)}\right] &\geq 1 - 2\exp\left(-2n\rho_n^2(\mathrm{tr}(\mathbf{D}\mathbf{Q}^{(c)})\zeta)^2\right), \\
\mathbb{P}\left[m\rho_n \nu^2 \delta_{(l)} \leq \mathrm{tr}(\widetilde{\mathbf{A}}) \leq m\rho_n \delta_{(u)}\right] &\geq 1 - 2\exp\left(-2m(\rho_n \nu^2 \mathrm{tr}(Q^{(c)})\delta)^2\right), \\
\mathbb{P}\left[\mathbf{1}^\intercal \mathbf{W}\mathbf{A}\mathbf{W}^\intercal \mathbf{1} \geq \tilde{c}_{min}^2 \rho_n m^2 \|\mathbf{Q}^{(c)}\|_{1,1}(\delta + 1)\right] &\geq 1 - \exp\left(-2\rho_n^2 m^2(\tilde{c}_{min}^2 \|\mathbf{Q}^{(c)}\|_{1,1}\delta)^2\right), \\
\mathbb{P}\left[\mathbf{1}_m^\intercal \mathbf{W}\mathbf{A}\mathbf{1}_n \geq \tilde{c}_{min}c_{min}\rho_n mn \|\mathbf{Q}^{(c)}\|_{1,1}(\zeta + 1)\right] &\geq 1 - \exp\left(-2\rho_n^2 mn(\tilde{c}_{min}c_{min}\|\mathbf{Q}^{(c)}\|_{1,1}\zeta)^2\right)
\end{aligned} \tag{22}$$

*where $\nu$ is the coarsening resolution parameter (c.f. Eq. 7), and*

$$\begin{aligned}
\zeta_{(l)} &\triangleq \mathrm{tr}(\mathbf{D}\mathbf{Q}^{(c)})(1 - \zeta), \zeta_{(u)} \triangleq \mathrm{tr}(\mathbf{D}\mathbf{Q}^{(c)})(1 + \zeta) \\
\delta_{(l)} &\triangleq \mathrm{tr}(Q^{(c)})(1 - \delta), \delta_{(u)} \triangleq \mathrm{tr}(Q^{(c)})(1 + \delta)
\end{aligned} \tag{23}$$

### A.3.3 PROOF OF LEMMA 4

*Proof.* From definition, $\mathbb{E}[\text{tr}(\mathbf{A})] = \text{tr}(\mathbf{P}^\mathsf{T}\mathbf{Q}\mathbf{P}) = n\rho_n\text{tr}(\mathbf{D}\mathbf{Q}^{(c)})$ and $\mathbb{E}[\text{tr}(\widetilde{\mathbf{A}})] = \rho_n\text{tr}(\boldsymbol{\Phi}\mathbf{Q}^{(c)}\boldsymbol{\Phi}^\mathsf{T})$. Using the Hoeffding's inequality for tail bounding independent random variables, for a constant $0 < \zeta < 1$ we have:

$$\mathbf{P}\left[|\text{tr}(\mathbf{A}) - n\rho_n\text{tr}(\mathbf{D}\mathbf{Q}^{(c)})| \geq n\rho_n\text{tr}(\mathbf{D}\mathbf{Q}^{(c)})\zeta\right] \leq 2\exp\left(\frac{-2(n\rho_n\text{tr}(\mathbf{D}\mathbf{Q}^{(c)})\zeta)^2}{n}\right) = 2\exp\left(-2n\rho_n^2\underbrace{(\text{tr}(\mathbf{D}\mathbf{Q}^{(c)})\zeta)^2}_{=\frac{\zeta_{(u)}-\zeta_{(l)}}{2}}\right)$$

(24)

Hence

$$n\rho_n\underbrace{\text{tr}(\mathbf{D}\mathbf{Q}^{(c)})(1-\zeta)}_{\zeta_{(l)}} \leq \text{tr}(\mathbf{A}) \leq n\rho_n\underbrace{\text{tr}(\mathbf{D}\mathbf{Q}^{(c)})(1+\zeta)}_{\zeta_{(u)}}$$

(25)

We follow similar steps for $\widetilde{\mathbf{A}}$. For all $i \in [m]$:

$$\sum_{k,k'\in[K]} \boldsymbol{\Phi}_{ik}\mathbf{Q}_{kk'}^{(c)}\boldsymbol{\Phi}_{ik'} \geq \sum_{k\in[K]:\boldsymbol{\Phi}_{ik}>0}[\boldsymbol{\Phi}_{ik}^2\mathbf{Q}_{kk}^{(c)} + 2\boldsymbol{\Phi}_{ik}\sum_{k'>k}\mathbf{Q}_{kk'}^{(c)}\boldsymbol{\Phi}_{ik'}]$$
$$\geq \nu^2\text{tr}(Q^{(c)}) + 2\nu^2||Q^{(c)}||_{\min}$$
$$\geq \nu^2\text{tr}(Q^{(c)})$$

(26)

based on which the following upper and lower bounds on $\text{tr}(\widetilde{\mathbf{A}})$ can be found.

$$m\rho_n\nu^2\text{tr}(Q^{(c)}) \leq \text{tr}(\widetilde{\mathbf{A}}) = \rho_n\text{tr}(\boldsymbol{\Phi}\mathbf{Q}^{(c)}\boldsymbol{\Phi}^\mathsf{T}) = \rho_n\sum_{i=1}^{m}\sum_{k,k'\in[K]}\boldsymbol{\Phi}_{ik}\mathbf{Q}_{kk'}^{(c)}\boldsymbol{\Phi}_{ik'} \leq m\rho_n\text{tr}(Q^{(c)})$$

(27)

We can now use Hoeffding's inequality to obtain the tail bound for a constant $0 < \delta < 1$:

$$\mathbf{P}\left[|\text{tr}(\widetilde{\mathbf{A}}) - \rho_n\text{tr}(\boldsymbol{\Phi}\mathbf{Q}^{(c)}\boldsymbol{\Phi}^\mathsf{T})| \geq \rho_n\text{tr}(\boldsymbol{\Phi}\mathbf{Q}^{(c)}\boldsymbol{\Phi}^\mathsf{T})\delta\right] \leq 2\exp\left(\frac{-2(\rho_n\text{tr}(\boldsymbol{\Phi}\mathbf{Q}^{(c)}\boldsymbol{\Phi}^\mathsf{T}))^2}{m}\right)$$
$$\leq 2\exp\left(-2m(\rho_n\nu^2\text{tr}(Q^{(c)})\delta)^2\right).$$

(28)

The event in Eq. 28 leads to the following upper and lower bound on $\text{tr}(\widetilde{\mathbf{A}})$:

$$m\rho_n\nu^2\underbrace{\text{tr}(Q^{(c)})(1-\delta)}_{\delta_{(l)}} \leq \rho_n\text{tr}(\boldsymbol{\Phi}\mathbf{Q}^{(c)}\boldsymbol{\Phi}^\mathsf{T})(1-\delta) \leq \text{tr}(\widetilde{\mathbf{A}}) \leq \rho_n\text{tr}(\boldsymbol{\Phi}\mathbf{Q}^{(c)}\boldsymbol{\Phi}^\mathsf{T})(1+\delta) \leq m\rho_n\underbrace{\text{tr}(Q^{(c)})(1+\delta)}_{\delta_{(u)}}.$$

(29)

We then replacing $\delta_{(l)}, \delta_{(u)}$ defined in Eq. 29 into Eq. 28.

Similarly, event

$$\mathbf{1}^\mathsf{T}\mathbf{W}\mathbf{A}\mathbf{W}^\mathsf{T}\mathbf{1} \geq \underbrace{\mathbf{1}^\mathsf{T}\boldsymbol{\Phi}}_{\geq\tilde{c}_{\min}m\mathbf{1}_K}\mathbf{Q}\boldsymbol{\Phi}^\mathsf{T}\mathbf{1}(1+\delta) \geq \tilde{c}_{\min}^2\rho_n m^2||\mathbf{Q}^{(c)}||_{1,1}(1+\delta)$$

(30)

happens with probability at least

$$1 - \mathbb{P}\left[\mathbf{1}^\mathsf{T}\mathbf{W}\mathbf{A}\mathbf{W}^\mathsf{T}\mathbf{1} - \mathbf{1}^\mathsf{T}\boldsymbol{\Phi}\mathbf{Q}\boldsymbol{\Phi}^\mathsf{T}\mathbf{1} < \tilde{c}_{\min}^2\rho_n m^2||\mathbf{Q}^{(c)}||_{1,1}\delta\right] \geq 1 - \exp\left(\frac{-2(\tilde{c}_{\min}^2\rho_n m^2||\mathbf{Q}^{(c)}||_{1,1}\delta)^2}{m^2}\right)$$
$$\geq 1 - \exp\left(-2\rho_n^2 m^2(\tilde{c}_{\min}^2||\mathbf{Q}^{(c)}||_{1,1}\delta)^2\right)$$

(31)

using Assumption 3 and following using one-sided Hoeffding's concentration inequality.

Finally, similar arguments can be applied to the following event

$$\mathbf{1}_m^\mathsf{T}\mathbf{W}\mathbf{A}\mathbf{1}_n \geq \underbrace{\mathbf{1}_m\boldsymbol{\Phi}}_{\geq\tilde{c}_{\min}m\mathbf{1}_K}\mathbf{Q}\underbrace{\mathbf{P}\mathbf{1}_n}_{=n\text{diag}(\mathbf{D})(1+\zeta)\geq nc_{\min}\mathbf{1}_K} \geq \tilde{c}_{\min}c_{\min}\rho_n mn||\mathbf{Q}^{(c)}||_{1,1}(1+\zeta)$$

(32)

that occurs with probability at least

$$
\begin{aligned}
1 - \mathbb{P}\left[\mathbf{1}_m^\mathsf{T}\mathbf{W}\mathbf{A}\mathbf{1}_n - \mathbf{1}_m\boldsymbol{\Phi}\mathbf{Q}\mathbf{P}\mathbf{1}_n < \tilde{c}_{\min}c_{\min}\rho_n mn\|\mathbf{Q}^{(c)}\|_{1,1}\zeta\right] & \geq 1 - \exp\left(-2\frac{(\tilde{c}_{\min}c_{\min}\rho_n mn\|\mathbf{Q}^{(c)}\|_{1,1}\zeta)^2}{mn}\right) \\
& \geq 1 - \exp\left(-2\rho_n^2 mn(\tilde{c}_{\min}c_{\min}\|\mathbf{Q}^{(c)}\|_{1,1}\zeta)^2\right)
\end{aligned}
\tag{33}
$$

using one-sided Hoeffding's inequality. This concludes the proof.

$\square$

### A.3.4  LEMMA: ERROR BETWEEN THE GRAMIANS OF RANDOM AND EXPECTED LTI SYSTEMS

Let $\overline{\mathcal{S}}_{\text{fine}}$ and $\overline{\mathcal{S}}_{\text{coarse}}$ denote the expected dynamics of LTI Eq. 9 when $\mathbf{A}$ and $\widetilde{\mathbf{A}}$ are replaced with the expected quantities $\bar{\mathbf{A}}$ and $\bar{\widetilde{\mathbf{A}}}$. The following result provides an error bound between the difference of Gramians of $\mathcal{S}_{\text{fine}}$ and $\overline{\mathcal{S}}_{\text{fine}}$ and that of $\mathcal{S}_{\text{coarse}}$ and $\overline{\mathcal{S}}_{\text{coarse}}$.

**Lemma 5.** *(**Error between the Gramians of random and expected LTI systems**): Under the assumptions stated in Lemma 1, the following holds with probability at least $1 - 2\exp\left(-2n\rho_n^2(tr(\mathbf{D}\mathbf{Q}^{(c)})\zeta)^2\right)$:*

$$
\alpha_n \triangleq \|\mathcal{C}(\mathbf{A}_{nom}, \mathbf{I}_{n\times n}) - \mathcal{C}(\bar{\mathbf{A}}_{nom}, \mathbf{I}_{n\times n})\|_{max} \leq \frac{\left[1 + \frac{\|\mathbf{Q}^{(c)}\|_{max}}{tr(\mathbf{D}\mathbf{Q}^{(c)})}\right]\left[\frac{1}{\zeta_{(l)}} + \frac{\rho_n\|\mathbf{Q}^{(c)}\|_{max}}{tr(\mathbf{D}\mathbf{Q}^{(c)})}\right]}{c^2(1-\beta^2)\zeta_{(l)}\left[1 - \frac{1}{c^2\cdot tr^2(\mathbf{D}\mathbf{Q}^{(c)})}\right]}\frac{1}{\rho_n^2 n} = \mathcal{O}(\frac{1}{\rho_n^2 n}),
\tag{34}
$$

*and with probability at least $1 - 2\exp\left(-2m(\rho_n\nu^2 tr(Q^{(c)})\delta)^2\right)$:*

$$
\widetilde{\alpha}_n \triangleq \|\mathcal{C}(\widetilde{\mathbf{A}}_{nom}, \mathbf{I}_{m\times m}) - \mathcal{C}(\bar{\widetilde{\mathbf{A}}}_{nom}, \mathbf{I}_{m\times m})\|_{max} \leq \frac{\left[1 + \frac{\|Q^{(c)}\|_{max}}{\nu^2 tr(Q^{(c)})}\right]\left[\frac{1}{\delta_{(l)}} + \frac{\rho_n\|Q^{(c)}\|_{max}}{tr(Q^{(c)})}\right]}{\tilde{c}^2(1-\beta^2)\delta_{(l)}(1 - (\frac{1}{\tilde{c}\nu^2 tr(Q^{(c)})})^2)}\frac{1}{m\rho_n^2\nu^4} = \mathcal{O}\left(\frac{1}{m\rho_n^2\nu^4}\right),
\tag{35}
$$

*where $r = |supp(\mathbf{w_i})|$ is the homogeneous coarsening parameter.*

Note that $\mathcal{C}(\mathbf{D}, \mathbf{I}) = (\mathbf{I} - \mathbf{D}^2)^{-1}$, where $\mathbf{D}$ can take $\mathbf{A}_{\text{nom}}$, $\bar{\mathbf{A}}_{\text{nom}}$, $\widetilde{\mathbf{A}}_{\text{nom}}$, or $\bar{\widetilde{\mathbf{A}}}_{\text{nom}}$. Thus, Theorem 5 is effectively bounding the difference of resolvents $(z\mathbf{I} - \mathbf{D}^2)^{-1}$ evaluated at $z = 1$. For $n = m$, we have $r = 1, \nu = 1$ and both $\alpha_n$ and $\widetilde{\alpha}_n$ coincide. Lemma 5 is basically a concentration result for the Gramians of $\mathcal{S}_{\text{fine}}$ (or $\mathcal{S}_{\text{coarse}}$) and $\overline{\mathcal{S}}_{\text{fine}}$ (or $\overline{\mathcal{S}}_{\text{coarse}}$); however the rate at which the difference goes to zero is different for fine- and coarse systems. Further, smaller the stability margin $1 - \beta$, looser are the bounds in Eq. 34 and Eq. 35.

### A.3.5  PROOF OF LEMMA 5

We begin by proving the inequality in Eq. 34. From Eq. 2 and Lemma 1, the limit below exists with the probability stated in the statement of lemma.

$$
\begin{aligned}
\mathcal{C}(\mathbf{A}_{\text{nom}}, \mathbf{I}_{n\times n}) & \triangleq \lim_{T\to\infty}\mathcal{C}_T(\mathbf{A}_{\text{nom}}, \mathbf{I}_{n\times n}) = \lim_{T\to\infty}\sum_{t=0}^{T-1}(\mathbf{A}_{\text{nom}})^t(\mathbf{A}_{\text{nom}}^\mathsf{T})^t \\
& = \lim_{T\to\infty}\sum_{t=0}^{T-1}(\mathbf{A}_{\text{nom}})^{2t} = (\mathbf{I} - \mathbf{A}_{\text{nom}}^2)^{-1}.
\end{aligned}
\tag{36}
$$

The last but one equality follows because $\mathbf{A}$ is a symmetric matrix, $\|\mathbf{A}_{\text{nom}}\|_2 = \|\frac{\mathbf{A}}{c\cdot tr(\mathbf{A})}\|_2 \leq \beta < 1$ (follows from the lemma's hypothesis), the last one from the Neumann series formula. Also, we have

$$
\|\bar{\mathbf{A}}_{\text{nom}}\|_2 = \|\frac{\bar{\mathbf{A}}}{c\cdot tr(\bar{\mathbf{A}})}\|_2 = \|\frac{\mathbf{P}^\mathsf{T}\mathbf{Q}^{(c)}\mathbf{P}}{c\cdot tr(\mathbf{P}^\mathsf{T}\mathbf{Q}^{(c)}\mathbf{P})}\|_2 \leq \frac{\sqrt{\|\mathbf{P}^\mathsf{T}\mathbf{Q}^{(c)}\mathbf{P}\|_\infty\|\mathbf{P}^\mathsf{T}\mathbf{Q}^{(c)}\mathbf{P}\|_1}}{c\cdot n tr(\mathbf{D}\mathbf{Q}^{(c)})} \leq \frac{1}{c\cdot tr(\mathbf{D}\mathbf{Q}^{(c)})} < 1,
\tag{37}
$$

since $[\mathbf{P}^\mathsf{T}\mathbf{Q}^{(c)}\mathbf{P}]_{\ell v} \leq 1$ for all $\ell, v \in [n]$. Similarly

$$
\mathcal{C}(\bar{\mathbf{A}}_{\text{nom}}, \mathbf{I}_{n\times n}) \triangleq \lim_{T\to\infty}\mathcal{C}_T(\bar{\mathbf{A}}_{\text{nom}}, \mathbf{I}_{n\times n}) = (\mathbf{I} - \bar{\mathbf{A}}_{\text{nom}}^2)^{-1}.
$$

From these observations, we establish the following identity (explanations for each step succeeds the equations):

$$
\begin{aligned}
\alpha_n &= \|\boldsymbol{\mathcal{C}}(\mathbf{A}_{\mathrm{nom}}, \mathbf{I}_{n\times n}) - \boldsymbol{\mathcal{C}}(\bar{\mathbf{A}}_{\mathrm{nom}}, \mathbf{I}_{n\times n})\|_{\max} \\
&= \|(\mathbf{I} - (\mathbf{A}_{\mathrm{nom}})^2)^{-1} - (\mathbf{I} - (\bar{\mathbf{A}}_{\mathrm{nom}})^2)^{-1}\|_{\max} \\
&\overset{(a)}{=} \|(\mathbf{I} - \mathbf{A}_{\mathrm{nom}})^{-1}(\mathbf{A}_{\mathrm{nom}}^2 - \bar{\mathbf{A}}_{\mathrm{nom}}^2)(\mathbf{I} - \bar{\mathbf{A}}_{\mathrm{nom}}^2)^{-1}\|_{\max} \\
&\overset{(b)}{\leq} \|(\mathbf{I} - \mathbf{A}_{\mathrm{nom}}^2)^{-1}\|_{\infty} \|\mathbf{A}_{\mathrm{nom}}^2 - (\bar{\mathbf{A}}_{\mathrm{nom}}^2\|_{\max} \|(\mathbf{I} - \bar{\mathbf{A}}_{\mathrm{nom}}^2)^{-1}\|_1 \\
&\overset{(c)}{\leq} \sqrt{n}\|(\mathbf{I} - \mathbf{A}_{\mathrm{nom}}^2)^{-1}\|_2 \|\mathbf{A}_{\mathrm{nom}}^2 - \bar{\mathbf{A}}_{\mathrm{nom}}^2\|_{\max} \sqrt{n}\|(\mathbf{I} - \bar{\mathbf{A}}_{\mathrm{nom}}^2)^{-1}\|_2 \\
&\overset{(d)}{\leq} n\frac{1}{1-\|\mathbf{A}_{\mathrm{nom}}^2\|_2} \|\mathbf{A}_{\mathrm{nom}}^2 - \bar{\mathbf{A}}_{\mathrm{nom}}^2\|_{\max} \frac{1}{1-\|\bar{\mathbf{A}}_{\mathrm{nom}}^2\|_2} \\
&\overset{(e)}{\leq} n\frac{1}{1-\beta^2} \|\mathbf{A}_{\mathrm{nom}}^2 - \bar{\mathbf{A}}_{\mathrm{nom}}^2\|_{\max} \frac{1}{1-\frac{1}{c^2\cdot\mathrm{tr}^2(\mathbf{DQ}^{(c)})}} \\
&\overset{(f)}{\leq} n\frac{1}{1-\beta^2} \|\mathbf{A}_{\mathrm{nom}} - \bar{\mathbf{A}}_{\mathrm{nom}}\|_{\max}[\|\mathbf{A}_{\mathrm{nom}}\|_{\max} + \|\bar{\mathbf{A}}_{\mathrm{nom}}\|_{\max}]\frac{1}{1-\frac{1}{c^2\cdot\mathrm{tr}^2(\mathbf{DQ}^{(c)})}} \\
&\leq n\frac{1}{1-\beta^2}\|\frac{\mathbf{A}}{c\cdot\mathrm{tr}(\mathbf{A})} - \frac{\bar{\mathbf{A}}}{c\cdot\mathrm{tr}(\bar{\mathbf{A}})}\|_{\max}[\frac{1}{cn\rho_n\zeta_{(l)}} + \frac{c\rho_n\|\mathbf{Q}^{(c)}\|_{\max}}{n\rho_n\mathrm{tr}(\mathbf{DQ}^{(c)})}]\frac{1}{1-\frac{1}{c^2\cdot\mathrm{tr}^2(\mathbf{DQ}^{(c)})}} \\
&\leq n\frac{1}{1-\beta^2}\frac{1}{c\cdot\mathrm{tr}(\mathbf{A})}\|\mathbf{A} - \bar{\mathbf{A}}\|_{\max}[1 + \frac{n\|\bar{\mathbf{A}}\|_{\max}}{\mathrm{tr}(\bar{\mathbf{A}})}]\frac{1}{cn\rho_n}[\frac{1}{\zeta_{(l)}} + \frac{\rho_n\|\mathbf{Q}^{(c)}\|_{\max}}{\mathrm{tr}(\mathbf{DQ}^{(c)})}]\frac{1}{1-\frac{1}{c^2\cdot\mathrm{tr}^2(\mathbf{DQ}^{(c)})}} \\
&\overset{(g)}{\leq} n\frac{1}{1-\beta^2}\frac{1}{cn\rho_n\zeta_{(l)}}[1 + \frac{n\rho_n\|\mathbf{Q}^{(c)}\|_{\max}}{\rho_n n\mathrm{tr}(\mathbf{DQ}^{(c)})}]\frac{1}{cn\rho_n}[\frac{1}{\zeta_{(l)}} + \frac{\rho_n\|\mathbf{Q}^{(c)}\|_{\max}}{\mathrm{tr}(\mathbf{DQ}^{(c)})}]\frac{1}{1-\frac{1}{c^2\cdot\mathrm{tr}^2(\mathbf{DQ}^{(c)})}} \\
&\leq \underbrace{\frac{[1 + \frac{\|\mathbf{Q}^{(c)}\|_{\max}}{\mathrm{tr}(\mathbf{DQ}^{(c)})}][\frac{1}{\zeta_{(l)}} + \frac{\rho_n\|\mathbf{Q}^{(c)}\|_{\max}}{\mathrm{tr}(\mathbf{DQ}^{(c)})}]}{c^2(1-\beta^2)\zeta_{(l)}\left[1 - \frac{1}{c^2\cdot\mathrm{tr}^2(\mathbf{DQ}^{(c)})}\right]}}_{\mathcal{O}(1)}\frac{1}{\rho_n^2 n} \\
&= \mathcal{O}\big(\frac{1}{\rho_n^2 n}\big).
\end{aligned}
\tag{38}
$$

where (a)-(d),(f) follow the inequalities itemized at the beginning of the appendix; (e) is because of the assumption in Lemma 1, Eq. 37, and that:

$$
\|(\mathbf{I} - \mathbf{A}_{\mathrm{nom}}^2)^{-1}\|_2 \leq 1/(1 - \|\mathbf{A}_{\mathrm{nom}}^2\|_2) \leq 1/(1 - \|\mathbf{A}_{\mathrm{nom}}\|_2^2) \leq 1/(1 - \beta^2);
\tag{39}
$$

and the rest of inequalities follow from definitions of $\mathbf{A}_{\mathrm{nom}}$, $\bar{\mathbf{A}}_{\mathrm{nom}}$ in the paragraphs processing Eq. 1 and Eq. 11.

The proof for the inequality in Eq. 35 follows similar lines as above, and hence, we provide a sketch, but not the full details. From Eq. 2, note that the following

$$
\begin{aligned}
\boldsymbol{\mathcal{C}}(\widetilde{\mathbf{A}}_{\mathrm{nom}}, \mathbf{I}_{m\times m}) &\triangleq \lim_{T\to\infty} \boldsymbol{\mathcal{C}}_T(\widetilde{\mathbf{A}}_{\mathrm{nom}}, \mathbf{I}_{m\times m}) = \lim_{T\to\infty}\sum_{t=0}^{T-1}\widetilde{\mathbf{A}}_{\mathrm{nom}}^t\left(\widetilde{\mathbf{A}}_{\mathrm{nom}}^\mathsf{T}\right)^t \\
&= \lim_{T\to\infty}\sum_{t=0}^{T-1}\widetilde{\mathbf{A}}_{\mathrm{nom}}^{2t} = (\mathbf{I} - \widetilde{\mathbf{A}}_{\mathrm{nom}}^2)^{-1},
\end{aligned}
\tag{40}
$$

where $\|\widetilde{\mathbf{A}}_{\mathrm{nom}}\|_2 = \|\frac{\widetilde{\mathbf{A}}}{\widetilde{c}\cdot\mathrm{tr}(\widetilde{\mathbf{A}})}\|_2 \leq \beta < 1$ (from the lemma's hypothesis). Moreover, we have $\boldsymbol{\mathcal{C}}(\bar{\widetilde{\mathbf{A}}}_{\mathrm{nom}}, \mathbf{I}_{m\times m}) = (\mathbf{I} - \bar{\widetilde{\mathbf{A}}}_{\mathrm{nom}}^2)^{-1}$, where

$$
\|\bar{\widetilde{\mathbf{A}}}_{\mathrm{nom}}\|_2 = \|\frac{\bar{\widetilde{\mathbf{A}}}}{\widetilde{c}\mathrm{tr}(\widetilde{\mathbf{A}})}\|_2 = \|\frac{\sqrt{\|\boldsymbol{\Phi}\mathbf{Q}^{(c)}\boldsymbol{\Phi}^\mathsf{T}\|_{\infty}\|\boldsymbol{\Phi}\mathbf{Q}^{(c)}\boldsymbol{\Phi}^\mathsf{T}\|_1}}{\widetilde{c}\mathrm{tr}(\boldsymbol{\Phi}\mathbf{Q}^{(c)}\boldsymbol{\Phi}^\mathsf{T})}\|_2 \leq \frac{\sqrt{mm}}{\widetilde{c}m\nu^2\mathrm{tr}(Q^{(c)})} \leq \frac{1}{\widetilde{c}\nu^2\mathrm{tr}(Q^{(c)})}
\tag{41}
$$

following the fact that $[\boldsymbol{\Phi}\mathbf{Q}^{(c)}\boldsymbol{\Phi}^\mathsf{T}]_{ij} \leq 1$.

Using these observations, we obtain the following inequality by taking similar steps as those for $\alpha_n$ in Eq. 38 (inner steps are removed due to redundancy) :

$$
\begin{aligned}
\tilde{\alpha}_n &\triangleq \|\mathcal{C}(\widetilde{\mathbf{A}}_{\text{nom}}, \mathbf{I}_{m\times n}) - \mathcal{C}(\bar{\widetilde{\mathbf{A}}}_{\text{nom}}, \mathbf{I}_{m\times m})\|_{\max} \\
&\le m\frac{1}{1-\|(\widetilde{\mathbf{A}}_{\text{nom}}^2\|_2}\frac{1}{\tilde{c}\text{tr}(\widetilde{\mathbf{A}})}\|\widetilde{\mathbf{A}} - \bar{\widetilde{\mathbf{A}}}\|_{\max}[1 + \frac{m\|\widetilde{\mathbf{A}}\|_{\max}}{\text{tr}(\widetilde{\mathbf{A}})}][\|\widetilde{\mathbf{A}}_{\text{nom}}\|_{\max} + \|\bar{\widetilde{\mathbf{A}}}_{\text{nom}}\|_{\max}]\frac{1}{1-\|\bar{\widetilde{\mathbf{A}}}_{\text{nom}}^2\|_2} \\
&\le m\frac{1}{1-\beta^2}\frac{1}{m\rho_n\nu^2\delta_{(l)}}[1 + \frac{m\rho_n\|Q^{(c)}\|_{\max}}{m\rho_n\nu^2\text{tr}(Q^{(c)})}][\frac{1}{\tilde{c}m\rho_n\nu^2\delta_{(l)}} + \frac{\rho_n\|Q^{(c)}\|_{\max}}{\tilde{c}m\rho_n\nu^2\text{tr}(Q^{(c)})}]\frac{1}{1-\|(\frac{\bar{\widetilde{\mathbf{A}}}}{\text{tr}(\widetilde{\mathbf{A}})})^2\|_2} \\
&\le m\frac{1}{1-\beta^2}\frac{1}{m\rho_n\nu^2\delta_{(l)}}[1 + \frac{\|Q^{(c)}\|_{\max}}{\nu^2\text{tr}(Q^{(c)})}][\frac{1}{\tilde{c}m\rho_n\nu^2\delta_{(l)}} + \frac{\rho_n\|Q^{(c)}\|_{\max}}{\tilde{c}m\rho_n\nu^2\text{tr}(Q^{(c)})}]\frac{1}{1-(\frac{1}{\tilde{c}\nu^2\text{tr}(Q^{(c)})})^2} \\
&\le \underbrace{\frac{[1 + \frac{\|Q^{(c)}\|_{\max}}{\nu^2\text{tr}(Q^{(c)})}][\frac{1}{\delta_{(l)}} + \frac{\rho_n\|Q^{(c)}\|_{\max}}{\text{tr}(Q^{(c)})}]}{\tilde{c}^2(1-\beta^2)\delta_{(l)}(1-(\frac{1}{\tilde{c}\nu^2\text{tr}(Q^{(c)})})^2)}}_{\mathcal{O}(1)}\frac{1}{m\rho_n^2\nu^4} \\
&= \mathcal{O}\left(\frac{1}{m\rho_n^2\nu^4}\right).
\end{aligned}
\tag{42}
$$

The proof is now complete.

### A.3.6 PROOF OF THEOREM 1

The proof of the theorem makes use of Lemma 5. Let $i \in [m]$, and recall that $\mathbf{B}_i = \text{diag}(\mathbf{w}_i^\top)$ and $\widetilde{\mathbf{B}}_i = r\mathbf{W}\text{diag}(\mathbf{w}_i^\top)$. From Eq. 10 and Eq. 13, consider the following bound

$$
\begin{aligned}
\Delta_i(\mathbf{A}, \widetilde{\mathbf{A}}) &\triangleq |\frac{r\boldsymbol{\theta}_{\text{group},\mathbf{A}}^{(i)}-1}{\sum_{i=1}^{m}(r\boldsymbol{\theta}_{\text{group},\mathbf{A}}^{(i)}-1)} - \frac{\boldsymbol{\theta}_{\text{coarse},\widetilde{\mathbf{A}}}^{(i)}-1}{\sum_{i=1}^{m}(\boldsymbol{\theta}_{\text{coarse},\widetilde{\mathbf{A}}}^{(i)}-1)}| \\
&\le \underbrace{|\frac{r\boldsymbol{\theta}_{\text{group},\mathbf{A}}^{(i)}-1}{\sum_{i=1}^{m}(r\boldsymbol{\theta}_{\text{group},\mathbf{A}}^{(i)}-1)} - \frac{r\boldsymbol{\theta}_{\text{group},\bar{\mathbf{A}}}^{(i)}-1}{\sum_{i=1}^{m}(r\boldsymbol{\theta}_{\text{group},\bar{\mathbf{A}}}^{(i)}-1)}|}_{=\Delta_i(\mathbf{A},\bar{\mathbf{A}})} \\
&+ \underbrace{|\frac{\boldsymbol{\theta}_{\text{coarse},\widetilde{\mathbf{A}}}^{(i)}-1}{\sum_{i=1}^{m}(\boldsymbol{\theta}_{\text{coarse},\widetilde{\mathbf{A}}}^{(i)}-1)} - \frac{\boldsymbol{\theta}_{\text{coarse},\bar{\widetilde{\mathbf{A}}}}^{(i)}-1}{\sum_{i=1}^{m}(\boldsymbol{\theta}_{\text{coarse},\widetilde{\mathbf{A}}}^{(i)}-1)}|}_{=\Delta_i(\widetilde{\mathbf{A}},\widetilde{\mathbf{A}})} \\
&+ \underbrace{|\frac{r\boldsymbol{\theta}_{\text{group},\bar{\mathbf{A}}}^{(i)}-1}{\sum_{i=1}^{m}(r\boldsymbol{\theta}_{\text{group},\bar{\mathbf{A}}}^{(i)}-1)} - \frac{\boldsymbol{\theta}_{\text{coarse},\bar{\widetilde{\mathbf{A}}}}^{(i)}-1}{\sum_{i=1}^{m}(\boldsymbol{\theta}_{\text{coarse},\bar{\widetilde{\mathbf{A}}}}^{(i)}-1)}||}_{\text{bias}=\Delta_i(\bar{\mathbf{A}}),\widetilde{\mathbf{A}})} \\
&\le \Delta_i(\mathbf{A}, \bar{\mathbf{A}}) + \Delta_i(\widetilde{\mathbf{A}}, \widetilde{\mathbf{A}}) + \text{bias}.
\end{aligned}
\tag{43}
$$

The complete proof of the bound on $\Delta_i(\mathbf{A}, \bar{\mathbf{A}})$ will be elaborated in Thm. 3 and it follows:

$$
\begin{aligned}
&\Delta_i(\mathbf{A}, \bar{\mathbf{A}}) \\
&\le \underbrace{\frac{\zeta_{(u)}^2}{\tilde{c}_{\min}c_{\min}\|\mathbf{Q}^{(c)}\|_{1,1}(1+\zeta)}\frac{[1 + \frac{\|\mathbf{Q}^{(c)}\|_{\max}}{\text{tr}(\mathbf{DQ}^{(c)})}][\frac{1}{\zeta_{(l)}} + \frac{\rho_n\|\mathbf{Q}^{(c)}\|_{\max}}{\text{tr}(\mathbf{DQ}^{(c)})}]}{(1-\beta^2)\zeta_{(l)}[1 - \frac{1}{c^2\cdot\text{tr}^2(\mathbf{DQ}^{(c)})}]}\left[1 + \frac{K\|\mathbf{Q}^{(c)}\mathbf{DQ}^{(c)}\|_{\max}}{\|\mathbf{Q}^{(c)}\mathbf{DQ}^{(c)}\|_{\min}(1 - (\frac{\|\mathbf{DQ}^{(c)}\|_F}{\text{tr}(\mathbf{DQ}^{(c)})})^2)}\right]}_{\mathcal{O}(1)}\frac{1}{\rho_n m} \\
&= \mathcal{O}(\frac{1}{\rho_n m})
\end{aligned}
\tag{44}
$$

We now derive an upper bound on $\Delta_i(\widetilde{\mathbf{A}}, \widetilde{\widetilde{\mathbf{A}}})$.

$$
\begin{aligned}
\Delta_i(\widetilde{\mathbf{A}}, \widetilde{\widetilde{\mathbf{A}}}) &= |\frac{\boldsymbol{\theta}^{(i)}_{\text{coarse},\tilde{\mathbf{A}}}{}^{-1}}{\sum_{i=1}^{m}(\boldsymbol{\theta}^{(i)}_{\text{coarse},\widetilde{\mathbf{A}}} - 1)} - \frac{\boldsymbol{\theta}^{(i)}_{\text{coarse},\tilde{\mathbf{A}}}{}^{-1}}{\sum_{i=1}^{m}(\boldsymbol{\theta}^{(i)}_{\text{coarse},\widetilde{\widetilde{\mathbf{A}}}} - 1)}| \\
&\leq \frac{1}{\sum_{i=1}^{m}(\boldsymbol{\theta}^{(i)}_{\text{coarse},\widetilde{\mathbf{A}}} - 1)} |\boldsymbol{\theta}^{(i)}_{\text{coarse},\widetilde{\mathbf{A}}} - \boldsymbol{\theta}^{(i)}_{\text{coarse},\widetilde{\widetilde{\mathbf{A}}}} + \frac{\sum_{i=1}^{m}(\boldsymbol{\theta}^{(i)}_{\text{coarse},\widetilde{\mathbf{A}}} - \boldsymbol{\theta}^{(i)}_{\text{coarse},\widetilde{\widetilde{\mathbf{A}}}})}{\sum_{i=1}^{m}(\boldsymbol{\theta}^{(i)}_{\text{coarse},\widetilde{\widetilde{\mathbf{A}}}} - 1)}[\boldsymbol{\theta}^{(i)}_{\text{coarse},\widetilde{\widetilde{\mathbf{A}}}} - 1]| \\
&\leq \frac{1}{\sum_{i=1}^{m}(\boldsymbol{\theta}^{(i)}_{\text{coarse},\widetilde{\mathbf{A}}} - 1)} \|\boldsymbol{\theta}_{\text{coarse},\widetilde{\mathbf{A}}} - \boldsymbol{\theta}_{\text{coarse},\widetilde{\widetilde{\mathbf{A}}}}\|_{\max} \left[1 + |\frac{m}{\sum_{i=1}^{m}(\boldsymbol{\theta}^{(i)}_{\text{coarse},\widetilde{\widetilde{\mathbf{A}}}} - 1)}[\boldsymbol{\theta}^{(i)}_{\text{coarse},\widetilde{\widetilde{\mathbf{A}}}} - 1]|\right] \\
&\leq \frac{1}{\sum_{i=1}^{m}(\boldsymbol{\theta}^{(i)}_{\text{coarse},\widetilde{\mathbf{A}}} - 1)} \tilde{\alpha}_n \left[1 + \frac{m\|\boldsymbol{\theta}_{\text{coarse},\widetilde{\widetilde{\mathbf{A}}}} - 1\|_{\max}}{\sum_{i=1}^{m}(\boldsymbol{\theta}^{(i)}_{\text{coarse},\widetilde{\widetilde{\mathbf{A}}}} - 1)}\right],
\end{aligned}
\tag{45}
$$

From the equality after Eq. 40, we have

$$
\boldsymbol{\theta}^{(i)}_{\text{coarse},\widetilde{\widetilde{\mathbf{A}}}} = \text{diag}_i\left((\mathbf{I} - \bar{\mathbf{A}}^2_{\text{nom}})^{-1}\right)
\tag{46}
$$

Furthermore, since $\boldsymbol{\Phi}\mathbf{Q}^{(c)}\boldsymbol{\Phi}^{\mathsf{T}} \leq \mathbf{1}\mathbf{1}^{\mathsf{T}}$ then $(\boldsymbol{\Phi}\mathbf{Q}^{(c)}\boldsymbol{\Phi}^{\mathsf{T}})^2 \leq \mathbf{1}\mathbf{1}^{\mathsf{T}}\mathbf{1}\mathbf{1}^{\mathsf{T}} = m\mathbf{1}\mathbf{1}^{\mathsf{T}}$

$$
\begin{aligned}
\|\boldsymbol{\theta}_{\text{coarse},\widetilde{\widetilde{\mathbf{A}}}} - 1\|_{\max} &= \|(\mathbf{I} - \bar{\mathbf{A}}^2_{\text{nom}})^{-1}\bar{\mathbf{A}}^2_{\text{nom}}\|_{\max} \\
&\leq \frac{1}{\text{tr}^2(\boldsymbol{\Phi}\mathbf{Q}^{(c)}\boldsymbol{\Phi}^{\mathsf{T}})}\|(\mathbf{I} - \bar{\mathbf{A}}^2_{\text{nom}})^{-1}m\mathbf{1}\mathbf{1}^{\mathsf{T}}\|_{\max} \\
&\leq \frac{m}{\text{tr}^2(\boldsymbol{\Phi}\mathbf{Q}^{(c)}\boldsymbol{\Phi}^{\mathsf{T}})}\|(\mathbf{I} - \bar{\mathbf{A}}^2_{\text{nom}})^{-1}\|_{\infty} \\
&\leq \frac{m\sqrt{m}}{\text{tr}^2(\boldsymbol{\Phi}\mathbf{Q}^{(c)}\boldsymbol{\Phi}^{\mathsf{T}})}\|(\mathbf{I} - \bar{\mathbf{A}}^2_{\text{nom}})^{-1}\|_2 \\
&\leq \frac{m\sqrt{m}}{(m\rho_n\nu^2\text{tr}(Q^{(c)})^2}\|(\mathbf{I} - \bar{\mathbf{A}}^2_{\text{nom}})^{-1}\|_2 \\
&\leq \frac{1}{\sqrt{m}\rho_n^2\nu^4\text{tr}^2(Q^{(c)})(1-(\frac{1}{\tilde{c}\nu^2\text{tr}(Q^{(c)})})^2)}.
\end{aligned}
\tag{47}
$$

Similarly

$$
\begin{aligned}
\sum_{i=1}^{m}(\boldsymbol{\theta}^{(i)}_{\text{coarse},\widetilde{\mathbf{A}}} - 1) &= \text{tr}[(\mathbf{I} - \widetilde{\widetilde{\mathbf{A}}}^2_{\text{nom}})^{-1} - \mathbf{I}] \\
&= \text{tr}[\widetilde{\widetilde{\mathbf{A}}}^2_{\text{nom}}(\mathbf{I} - \widetilde{\widetilde{\mathbf{A}}}^2_{\text{nom}})^{-1}] \\
&\geq \text{tr}[\widetilde{\widetilde{\mathbf{A}}}^2_{\text{nom}}] \\
&\geq \frac{\text{tr}[(\boldsymbol{\Phi}\mathbf{Q}^{(c)}\boldsymbol{\Phi}^{\mathsf{T}})^2]}{\text{tr}^2[\boldsymbol{\Phi}\mathbf{Q}^{(c)}\boldsymbol{\Phi}^{\mathsf{T}}]} \\
&\geq \frac{m^2\nu^3\tilde{c}_{\min}\text{tr}[(Q^{(c)})^2]}{(\tilde{c}m\text{tr}[Q^{(c)}])^2} \\
&= \frac{\nu^3\tilde{c}_{\min}\text{tr}[(Q^{(c)})^2]}{\tilde{c}^2\text{tr}^2[Q^{(c)}]}
\end{aligned}
\tag{48}
$$

where all inequalities follow well-known matrix norm axiom. The last line in Eq. 48 is based on Assumption 2, and

$$
\begin{aligned}
\text{tr}[(\boldsymbol{\Phi}\mathbf{Q}^{(c)}\boldsymbol{\Phi}^{\mathsf{T}})^2] &\geq \nu\tilde{c}_{\min}m\text{tr}[\boldsymbol{\Phi}(\mathbf{Q}^{(c)})^2\boldsymbol{\Phi}^{\mathsf{T}}] \\
&\geq \nu\tilde{c}_{\min}mm\nu^2\text{tr}[(Q^{(c)})^2] \\
&= m^2\nu^3\tilde{c}_{\min}\text{tr}[(Q^{(c)})^2]
\end{aligned}
\tag{49}
$$

similar to the derivation of Eq. 27 since for all $k \in [K]$:

$$[\mathbf{\Phi}^\mathsf{T}\mathbf{\Phi}]_{kk} = \sum_{i\in[m]} \mathbf{\Phi}_{ik}^2 \geq \nu \sum_{i\in[m]} \mathbf{\Phi}_{ik} \geq \nu\tilde{c}_{\min}m \tag{50}$$

We use the definition of $\boldsymbol{\theta}_{\mathrm{coarse},\widetilde{\mathbf{A}}}$ in Eq. 13:

$$
\begin{aligned}
|\sum_{i=1}^{m}(\boldsymbol{\theta}_{\mathrm{coarse},\widetilde{\mathbf{A}}}^{(i)} - 1)| &= \sum_{i=1}^{m}\mathrm{diag}_i\left((\mathbf{I}-\widetilde{\mathbf{A}}_{\mathrm{nom}}^2)^{-1}-\mathbf{I}\right)\\
&= \mathrm{tr}\left((\mathbf{I}-\widetilde{\mathbf{A}}_{\mathrm{nom}}^2)^{-1}-\mathbf{I}\right)\\
&\geq \mathrm{tr}(\widetilde{\mathbf{A}}_{\mathrm{nom}}^2)\\
&\geq \frac{\mathrm{tr}(\widetilde{\mathbf{A}}^2)}{\mathrm{tr}^2(\widetilde{\mathbf{A}})}\\
&\overset{(a)}{\geq} \frac{m^2\rho_n||\mathbf{Q}^{(c)}||_{1,1}}{r^2}\frac{\tilde{c}_{\min}^2(1+\delta)}{(m\rho_n\delta_{(u)})^2}\\
&= \frac{1}{\rho_n r^2}\frac{\tilde{c}_{\min}^2(1+\delta)||\mathbf{Q}^{(c)}||_{1,1}}{\delta_{(u)}^2},
\end{aligned}
\tag{51}
$$

where the inequality (a) in Eq. 51 comes from

$$
\begin{aligned}
\mathrm{tr}(\widetilde{\mathbf{A}}^2) &= \sum_{i,j}\widetilde{\mathbf{A}}_{ij}^2\\
&= \sum_{i,j}\left(\frac{1}{r^2}\sum_{v\in\mathcal{K}_i,\ell\in\mathcal{K}_j}\mathbf{A}_{v\ell}\right)^2\\
&= \frac{1}{r^4}\sum_{i,j}\sum_{v,v'\in\mathcal{K}_i,\ell,\ell'\in\mathcal{K}_j}\mathbf{A}_{v\ell}\mathbf{A}_{v'\ell'}\\
&= \frac{1}{r^4}\left[\sum_{v\in\cup_i\mathcal{K}_i,\ell\in\cup_j\mathcal{K}_j}\mathbf{A}_{v\ell} + \sum_{v\neq v'\in\cup_i\mathcal{K}_i\,\text{or}\,\ell\neq\ell'\in\cup_j\mathcal{K}_j}\mathbf{A}_{v\ell}\mathbf{A}_{v'\ell'}\right]\\
&\geq \frac{1}{r^2}\sum_{i,j}\mathbf{w}_i^\mathsf{T}\mathbf{A}\mathbf{w}_j\\
&= \frac{1}{r^2}\mathbf{1}^\mathsf{T}\mathbf{W}\mathbf{A}\mathbf{W}^\mathsf{T}\mathbf{1}\\
&\geq \frac{1}{r^2}\underbrace{\mathbf{1}^\mathsf{T}\mathbf{\Phi}}_{\geq\tilde{c}_{\min}m\mathbf{1}_K}\mathbf{Q}\underbrace{\mathbf{\Phi}^\mathsf{T}\mathbf{1}}_{\geq\tilde{c}_{\min}m\mathbf{1}_K^\mathsf{T}}(1+\delta)\\
&\geq \frac{m^2\rho_n||\mathbf{Q}^{(c)}||_{1,1}}{r^2}\tilde{c}_{\min}^2(1+\delta),
\end{aligned}
\tag{52}
$$

where the event defined in Eq. 30 is used whose corresponding probability is Eq. 31.

Replacing Eq. 47, Eq. 48, Eq. 51, and Eq. 42 into Eq. 45 yields:

$$
\begin{aligned}
\Delta_i(\widetilde{\mathbf{A}},\widetilde{\mathbf{A}}) &\leq \rho_n r^2\frac{\delta_{(u)}^2}{\tilde{c}_{\min}^2(1+\delta)||\mathbf{Q}^{(c)}||_{1,1}}\tilde{\alpha}_n\left[1+\frac{m\frac{1}{\sqrt{m}\rho_n^2\nu^4\mathrm{tr}^2(Q^{(c)})(1-(\frac{1}{\tilde{c}\nu^2\mathrm{tr}(Q^{(c)})})^2)}}{\frac{\nu^3\tilde{c}_{\min}\mathrm{tr}[(Q^{(c)})^2]}{\tilde{c}^2\mathrm{tr}^2[Q^{(c)}]}}\right]\\
&= \mathcal{O}\left(\frac{\rho_n r^2\sqrt{m}}{m\rho_n^2\rho_n^2}\right)\\
&= \mathcal{O}\left(\frac{r^2}{\sqrt{m}\rho_n^3}\right)
\end{aligned}
\tag{53}
$$

The statement of the theorem follows by invoking Eq. 44 and Eq. 53 into Eq. 43:

$$\Delta_i\left(\tfrac{1}{\mathrm{tr}(\mathbf{A})}\mathbf{A},\tfrac{1}{\mathrm{tr}(\widetilde{\mathbf{A}})}\widetilde{\mathbf{A}}\right) = \mathrm{bias} + \mathcal{O}\left(\frac{1}{\rho_n m}+\frac{r^2}{\sqrt{m}\rho_n^3}\right). \tag{54}$$

with a joint probability of at least

$$
\begin{aligned}
1 - 2\exp\left(-2n\rho_n^2(\mathrm{tr}(\mathbf{D}\mathbf{Q}^{(c)})\zeta)^2\right) - \exp\left(-2\rho_n^2 mn(\tilde{c}_{\min}c_{\min}||\mathbf{Q}^{(c)}||_{1,1}\zeta)^2\right)\\
- 2\exp\left(-2m(\rho_n\nu^2\mathrm{tr}(Q^{(c)})\delta)^2\right) - \exp\left(-2\rho_n^2 m^2(\tilde{c}_{\min}^2||\mathbf{Q}^{(c)}||_{1,1}\delta)^2\right)
\end{aligned}
\tag{55}
$$

per Lemma 3. We then get the minimum of the four exponents in Eq. 70. The proof is now complete.

### A.4 PROOFS FOR SECTION 6: LEARNING APPROACH FOR GROUP AVERAGE CONTROLLABILITY

#### A.4.1 PROOF OF LEMMA 2

We start by defining $\boldsymbol{\theta}_{\text{fine},\mathbf{M}}$ similar to Eq. 10 and Eq. 13 ($\mathbf{M}$ will be later substituted with $\mathbf{A}$ and $\bar{\mathbf{A}}$),

$$\boldsymbol{\theta}_{\text{fine},\mathbf{M}}^{\mathsf{T}} \triangleq [\text{tr}[\mathcal{C}(\mathbf{M}_{\text{nom}}, \mathbf{e}_1)] \quad \dots \quad \text{tr}[\mathcal{C}(\mathbf{M}_{\text{nom}}, \mathbf{e}_n)]] \in \mathbb{R}^{n \times 1}. \tag{56}$$

Using the Gramian definition in Eq. 2, we have:

$$
\begin{aligned}
\boldsymbol{\theta}_{\text{fine},\mathbf{M}}^{(i)} &= \text{tr}\left[\mathcal{C}(\mathbf{M}_{\text{nom}}, \mathbf{e}_i)\right] \\
&= \text{tr}\left[\sum_{\tau=0}^{\infty} \mathbf{M}_{\text{nom}}^{\tau} \mathbf{e}_i \mathbf{e}_i^{\mathsf{T}} \mathbf{M}_{\text{nom}}^{\tau}\right] \\
&= \sum_{\tau=0}^{\infty} \text{tr}(\mathbf{M}_{\text{nom}}^{\tau} \mathbf{e}_i \mathbf{e}_i^{\mathsf{T}} \mathbf{M}_{\text{nom}}^{\tau}) \\
&= \sum_{\tau=0}^{\infty} \text{tr}(\mathbf{e}_i^{\mathsf{T}} \mathbf{M}_{\text{nom}}^{2\tau} \mathbf{e}_i) \\
&= \sum_{\tau=0}^{\infty} \text{diag}_i(\mathbf{M}_{\text{nom}}^{2\tau}) \\
&= \text{diag}_i\left(\sum_{\tau=0}^{\infty} \mathbf{M}_{\text{nom}}^{2\tau}\right) \\
&= \text{diag}_i\left((\mathbf{I} - \mathbf{M}_{\text{nom}}^2)^{-1}\right).
\end{aligned}
\tag{57}
$$

We first simplify the term $\boldsymbol{\theta}_{\text{fine},\bar{\mathbf{A}}}$ in Eq. 60, by substituting $\bar{\mathbf{A}}$ with $\mathbf{P}^{\mathsf{T}}\mathbf{Q}\mathbf{P}$ introduced prior to Eq. 6:

$$
\begin{aligned}
\boldsymbol{\theta}_{\text{fine},\bar{\mathbf{A}}} &= \begin{bmatrix} \text{tr}[\mathcal{C}(\bar{\mathbf{A}}_{\text{nom}}, \mathbf{e}_1)] \\ \dots \\ \text{tr}[\mathcal{C}(\bar{\mathbf{A}}_{\text{nom}}, \mathbf{e}_n)] \end{bmatrix} = \text{diag}\left(\sum_{\tau=0}^{\infty} \bar{\mathbf{A}}_{\text{nom}}^{2\tau}\right) \\
&= \text{diag}(\sum_{\tau=0}^{\infty}(\frac{1}{\text{tr}(\mathbf{P}^{\mathsf{T}}\mathbf{Q}\mathbf{P})}\mathbf{P}^{\mathsf{T}}\mathbf{Q}\mathbf{P})^{2\tau}) = \text{diag}(\sum_{\tau=0}^{\infty}(\frac{1}{n\text{tr}(\mathbf{D}\mathbf{Q}^{(c)})}\mathbf{P}^{\mathsf{T}}\mathbf{Q}^{(c)}\mathbf{P})^{2\tau}) \\
&= \text{diag}(I + (\frac{1}{n\text{tr}(\mathbf{D}\mathbf{Q}^{(c)})})^2\mathbf{P}^{\mathsf{T}}\mathbf{Q}^{(c)}\mathbf{P}\mathbf{P}^{\mathsf{T}}\mathbf{Q}^{(c)}\mathbf{P} + (\frac{1}{n\text{tr}(\mathbf{D}\mathbf{Q}^{(c)})})^4\mathbf{P}^{\mathsf{T}}\mathbf{Q}^{(c)}\mathbf{P}\mathbf{P}^{\mathsf{T}}\mathbf{Q}^{(c)}\mathbf{P} + \cdots) \\
&= \text{diag}(I + \frac{1}{n}(\frac{1}{\text{tr}(\mathbf{D}\mathbf{Q}^{(c)})})^2\mathbf{P}^{\mathsf{T}}\mathbf{Q}^{(c)}\underbrace{\frac{1}{n}\mathbf{P}\mathbf{P}^{\mathsf{T}}}_{\triangleq \mathbf{D}}\mathbf{Q}^{(c)}\mathbf{P} \\
&\qquad + \frac{1}{n}(\frac{1}{\text{tr}(\mathbf{D}\mathbf{Q}^{(c)})})^4\mathbf{P}^{\mathsf{T}}\mathbf{Q}^{(c)}\underbrace{\frac{1}{n}\mathbf{P}\mathbf{P}^{\mathsf{T}}}_{}\mathbf{Q}^{(c)}\underbrace{\frac{1}{n}\mathbf{P}\mathbf{P}^{\mathsf{T}}}_{}\mathbf{Q}^{(c)}\underbrace{\frac{1}{n}\mathbf{P}\mathbf{P}^{\mathsf{T}}}_{}\mathbf{Q}^{(c)}\mathbf{P} + \cdots) \\
&= \mathbf{1}_n + \frac{1}{n\text{tr}(\mathbf{D}\mathbf{Q}^{(c)})}\text{diag}(\mathbf{P}^{\mathsf{T}}\mathbf{Q}^{(c)}\frac{\mathbf{D}\mathbf{Q}^{(c)}}{\text{tr}(\mathbf{D}\mathbf{Q}^{(c)})}\mathbf{P} + \frac{1}{\text{tr}(\mathbf{A})}\mathbf{P}^{\mathsf{T}}\mathbf{Q}^{(c)}\frac{\mathbf{D}\mathbf{Q}^{(c)}}{\text{tr}(\mathbf{D}\mathbf{Q}^{(c)})}\frac{\mathbf{D}\mathbf{Q}^{(c)}}{\text{tr}(\mathbf{D}\mathbf{Q}^{(c)})}\frac{\mathbf{D}\mathbf{Q}^{(c)}}{\text{tr}(\mathbf{D}\mathbf{Q}^{(c)})}\mathbf{P} + \cdots) \\
&= \mathbf{1}_n + \frac{1}{n\text{tr}(\mathbf{D}\mathbf{Q}^{(c)})}\text{diag}(\mathbf{P}^{\mathsf{T}}\underbrace{\frac{\mathbf{Q}^{(c)}\mathbf{D}\mathbf{Q}^{(c)}}{\text{tr}(\mathbf{D}\mathbf{Q}^{(c)})}[\mathbf{I} + (\frac{\mathbf{D}\mathbf{Q}^{(c)}}{\text{tr}(\mathbf{D}\mathbf{Q}^{(c)})})^2 + \cdots]}_{\triangleq \Upsilon(\mathbf{Q}^{(c)}, \mathbf{D})}\mathbf{P}) \\
&= \mathbf{1}_n + \frac{1}{n\text{tr}(\mathbf{D}\mathbf{Q}^{(c)})}\text{diag}(\mathbf{P}^{\mathsf{T}}\Upsilon(\mathbf{Q}^{(c)}, \mathbf{D})\mathbf{P}) \\
&\stackrel{(a)}{=} \mathbf{1}_n + \frac{1}{n\text{tr}(\mathbf{D}\mathbf{Q}^{(c)})}(\mathbf{P} \circ \mathbf{P})^{\mathsf{T}}\text{diag}(\Upsilon(\mathbf{Q}^{(c)}, \mathbf{D})) \\
&\stackrel{(b)}{=} \mathbf{1}_n + \frac{1}{n\text{tr}(\mathbf{D}\mathbf{Q}^{(c)})}\mathbf{P}^{\mathsf{T}}\text{diag}(\Upsilon),
\end{aligned}
\tag{58}
$$

where (a) is due to the special structure of $\mathbf{P}$ Eq. 4 since for an arbitrary matrix $\mathbf{M}$ of appropriate size:

$$
\begin{aligned}
\text{diag}_i(\mathbf{P}^{\mathsf{T}}\mathbf{M}\mathbf{P}) &= \sum_{k,k' \in [K]} \mathbf{P}_{ki}\mathbf{M}_{k,k'}\mathbf{P}_{k'i} = \sum_{k,k' \in [K]} \mathbf{P}_{ki}\mathbf{M}_{k,k'}\mathbf{P}_{k'i} \\
&= \sum_{k \in [K]} \mathbf{P}_{ki}\mathbf{M}_{k,k}\mathbf{P}_{ki} = \sum_{k \in [K]} \mathbf{P}_{ki}^2 \mathbf{M}_{k,k} \\
&= (\mathbf{P}_i \circ \mathbf{P}_i)\text{diag}(\mathbf{M}),
\end{aligned}
\tag{59}
$$

and (b) is true because $\mathbf{P}$ is binary. Replacing Eq. 58 into Eq. 60 yields:

$$\boldsymbol{\theta}_{\text{group},\bar{\mathbf{A}}} = \frac{1}{r}\mathbf{W}\left(\mathbf{1}_n + \frac{1}{n\text{tr}(\mathbf{DQ}^{(c)})}\mathbf{P}^{\mathsf{T}}\text{diag}(\Upsilon))\right) = \frac{1}{r}\left(\mathbf{1}_m + \frac{1}{n\text{tr}(\mathbf{DQ}^{(c)})}\boldsymbol{\Phi}\text{diag}(\Upsilon))\right) \quad (60)$$

The proof is now complete.

### A.4.2 THEOREM 3 AND PROOF

Define the error metric similar to Eq. 14:

$$\Delta_i(\mathbf{A},\bar{\mathbf{A}}) \triangleq \left| \frac{r\boldsymbol{\theta}_{\text{group},\mathbf{A}}^{(i)} - 1}{\sum_{i=1}^m [r\boldsymbol{\theta}_{\text{group},\mathbf{A}}^{(i)} - 1]} - \frac{r\boldsymbol{\theta}_{\text{group},\widetilde{\mathbf{A}}}^{(i)} - 1}{\sum_{i=1}^m [r\boldsymbol{\theta}_{\text{group},\bar{\mathbf{A}}}^{(i)} - 1]} \right|, \quad \text{for all } i \in [m]. \quad (61)$$

**Theorem 3.** *(Component wise error bound between $\boldsymbol{\theta}_{\text{group},\mathbf{A}}$ and $\boldsymbol{\theta}_{\text{group},\bar{\mathbf{A}}}$): Let $\Delta_i(\mathbf{A},\bar{\mathbf{A}})$ be defined as above and $\nu$ be the resolution parameter given by Eq. 7. Under the assumptions stated in Section 3 and Lemma 1, the following holds:*

$$\Delta_i(\mathbf{A},\bar{\mathbf{A}}) \leq \underbrace{\frac{\zeta_{(u)}^2}{\tilde{c}_{min}c_{min}||\mathbf{Q}^{(c)}||_{1,1}(1+\zeta)} \frac{[1+\frac{||\mathbf{Q}^{(c)}||_{max}}{tr(\mathbf{DQ}^{(c)})}][\frac{1}{\zeta_{(l)}} + \frac{\rho_n||\mathbf{Q}^{(c)}||_{max}}{tr(\mathbf{DQ}^{(c)})}]}{(1-\beta^2)\zeta_{(l)}\left[1 - \frac{1}{c^2 \cdot tr^2(\mathbf{DQ}^{(c)})}\right]} \left[1 + \frac{K||\mathbf{Q}^{(c)}\mathbf{DQ}^{(c)}||_{max}}{||\mathbf{Q}^{(c)}\mathbf{DQ}^{(c)}||_{min}(1 - (\frac{||\mathbf{DQ}^{(c)}||_F}{tr(\mathbf{DQ}^{(c)})})^2)}\right]}_{\mathcal{O}(1)} \frac{1}{\rho_n m}$$

$$(62)$$

*with probability at least $1 - 3\exp\left(-2\hat{\kappa}(\mathbf{Q}^{(c)},\mathbf{D},m,n,\rho_n)\right)$, where Further, for a constant $0 < \zeta < 1$, the exponent is $\hat{\kappa}(\mathbf{Q}^{(c)},\mathbf{D},m,n,\rho_n) = \min\left\{n\rho_n^2(tr(\mathbf{DQ}^{(c)})\zeta)^2, \ mn\rho_n^2(\tilde{c}_{min}c_{min}||\mathbf{Q}^{(c)}||_{1,1}\zeta)^2\right\}$.*

### A.4.3 PROPOSITION 4 AND PROOF

**Proposition 4.** *(Group Average Controllability for $\bar{\mathbf{A}}$) The group average controllability vector for $\bar{\mathbf{A}}$ is*

$$\boldsymbol{\theta}_{\text{group},\mathbf{M}}^{(i)} = \frac{1}{r}\mathbf{w}_i\boldsymbol{\theta}_{\text{fine},\mathbf{M}}. \quad (63)$$

*Proof.* We begin by simplifying the group average controllability using its definition in Eq. 10 and the definition of Gramian in Eq. 2, for a general matrix notation $\mathbf{M}$ which can be replaced by either $\mathbf{A}$ or $\bar{\mathbf{A}}$:

$$
\begin{aligned}
\boldsymbol{\theta}_{\text{group},\mathbf{M}}^{(i)} &= \text{tr}\left[\mathcal{C}(\mathbf{M}_{\text{nom}},\text{diag}(\mathbf{w}_i^{\mathsf{T}}))\right] \\
&= \text{tr}\left[\sum_{\tau=0}^{\infty}\mathbf{M}_{\text{nom}}^{\tau}\text{diag}(\mathbf{w}_i^{\mathsf{T}})\text{diag}(\mathbf{w}_i^{\mathsf{T}})^{\mathsf{T}}\mathbf{M}_{\text{nom}}^{\tau}\right] \\
&= \text{tr}\left[\sum_{\tau=0}^{\infty}\mathbf{M}_{\text{nom}}^{\tau}\text{diag}((\mathbf{w}_i \circ \mathbf{w}_i)^{\mathsf{T}})\mathbf{M}_{\text{nom}}^{\tau}\right] \\
&\overset{(a)}{=} \frac{1}{r^2}\text{tr}\left[\sum_{\tau=0}^{\infty}\mathbf{M}_{\text{nom}}^{\tau}(\sum_{v\in\text{supp}(\mathbf{w}_i)}\mathbf{e}_v\mathbf{e}_v^{\mathsf{T}})\mathbf{M}_{\text{nom}}^{\tau}\right] \\
&= \frac{1}{r^2}\sum_{v\in\text{supp}(\mathbf{w}_i)}\text{tr}\left[\sum_{\tau=0}^{\infty}\mathbf{M}_{\text{nom}}^{\tau}\mathbf{e}_v\mathbf{e}_v^{\mathsf{T}}\mathbf{M}_{\text{nom}}^{\tau}\right] \\
&= \frac{1}{r^2}\sum_{v\in\text{supp}(\mathbf{w}_i)}\underbrace{\text{tr}\left[\sum_{\tau=0}^{\infty}\mathbf{M}_{\text{nom}}^{\tau}\mathbf{e}_v\mathbf{e}_v^{\mathsf{T}}\mathbf{M}_{\text{nom}}^{\tau}\right]}_{\boldsymbol{\theta}_{\text{fine},\mathbf{M}}^{(v)}},
\end{aligned}
\quad (64)
$$

where (a) is due to the assumption of $r$-homogeneous $\mathbf{W}$, $\circ$ denotes the Hadamard product, and we have already defined $\boldsymbol{\theta}_{\text{fine},\mathbf{M}}$ in Eq. 56. Putting Eq. 64 in vector form concludes the proof. $\square$

### A.4.4 Proof of Theorem 3

*Proof.* We start by substituting $\mathbf{M}$ into Eq. 63, from Proposition 4, with $\mathbf{A}$ and $\bar{\mathbf{A}}$, yields :

$$
\begin{aligned}
\Delta_i(\mathbf{A}, \bar{\mathbf{A}}) &= \left| \frac{r\boldsymbol{\theta}_{\text{group},\mathbf{A}}^{(i)}-1}{\sum\limits_{i=1}^{m}(r\boldsymbol{\theta}_{\text{group},\mathbf{A}}^{(i)}-1)} - \frac{r\boldsymbol{\theta}_{\text{group},\bar{\mathbf{A}}}^{(i)}-1}{\sum\limits_{i=1}^{m}(r\boldsymbol{\theta}_{\text{group},\bar{\mathbf{A}}}^{(i)}-1)} \right| \\
&= \left| \frac{\mathbf{w}_i\boldsymbol{\theta}_{\text{fine},\mathbf{A}}-1}{\sum\limits_{i=1}^{m}(\mathbf{w}_i\boldsymbol{\theta}_{\text{fine},\mathbf{A}}-1)} - \frac{\mathbf{w}_i\boldsymbol{\theta}_{\text{fine},\bar{\mathbf{A}}}-1}{\sum\limits_{i=1}^{m}(\mathbf{w}_i\boldsymbol{\theta}_{\text{fine},\bar{\mathbf{A}}}-1)} \right| \\
&= \frac{1}{\sum\limits_{i=1}^{m}(\mathbf{w}_i\boldsymbol{\theta}_{\text{fine},\mathbf{A}}-1)} \left| \mathbf{w}_i\boldsymbol{\theta}_{\text{fine},\mathbf{A}}-1 - \frac{\sum\limits_{i=1}^{m}(\mathbf{w}_i\boldsymbol{\theta}_{\text{fine},\mathbf{A}}-1)}{\sum\limits_{i=1}^{m}(\mathbf{w}_i\boldsymbol{\theta}_{\text{fine},\bar{\mathbf{A}}}-1)}[\mathbf{w}_i\boldsymbol{\theta}_{\text{fine},\bar{\mathbf{A}}}-1] \right| \\
&= \frac{1}{\sum\limits_{i=1}^{m}(\mathbf{w}_i\boldsymbol{\theta}_{\text{fine},\mathbf{A}}-1)} \left| \mathbf{w}_i(\boldsymbol{\theta}_{\text{fine},\mathbf{A}}-\boldsymbol{\theta}_{\text{fine},\bar{\mathbf{A}}}) - \left[ \frac{\sum\limits_{i=1}^{m}(\mathbf{w}_i\boldsymbol{\theta}_{\text{fine},\mathbf{A}}-1)}{\sum\limits_{i=1}^{m}(\mathbf{w}_i\boldsymbol{\theta}_{\text{fine},\bar{\mathbf{A}}}-1)} - 1 \right] [\mathbf{w}_i\boldsymbol{\theta}_{\text{fine},\bar{\mathbf{A}}}-1] \right| \\
&\leq \frac{1}{\sum\limits_{i=1}^{m}(\mathbf{w}_i\boldsymbol{\theta}_{\text{fine},\mathbf{A}}-1)} \left[ |\mathbf{w}_i(\boldsymbol{\theta}_{\text{fine},\mathbf{A}}-\boldsymbol{\theta}_{\text{fine},\bar{\mathbf{A}}})| + \left| \frac{\sum\limits_{i=1}^{m}\mathbf{w}_i(\boldsymbol{\theta}_{\text{fine},\mathbf{A}}-\boldsymbol{\theta}_{\text{fine},\bar{\mathbf{A}}})}{\sum\limits_{i=1}^{m}(\mathbf{w}_i\boldsymbol{\theta}_{\text{fine},\bar{\mathbf{A}}}-1)}[\mathbf{w}_i\boldsymbol{\theta}_{\text{fine},\bar{\mathbf{A}}}-1] \right| \right] \\
&\leq \frac{1}{\sum\limits_{i=1}^{m}(\mathbf{w}_i\boldsymbol{\theta}_{\text{fine},\mathbf{A}}-1)} \left[ |\mathbf{w}_i(\boldsymbol{\theta}_{\text{fine},\mathbf{A}}-\boldsymbol{\theta}_{\text{fine},\bar{\mathbf{A}}})| + \left| \frac{\sum\limits_{i=1}^{m}\mathbf{w}_i(\boldsymbol{\theta}_{\text{fine},\mathbf{A}}-\boldsymbol{\theta}_{\text{fine},\bar{\mathbf{A}}})}{\sum\limits_{i=1}^{m}\frac{1}{n\text{tr}(\mathbf{D}\mathbf{Q}^{(c)})}\boldsymbol{\Phi}^{(i)}\text{diag}(\Upsilon)} \frac{1}{n\text{tr}(\mathbf{D}\mathbf{Q}^{(c)})}\boldsymbol{\Phi}^{(i)}\text{diag}(\Upsilon) \right| \right] \\
&\leq \frac{1}{\sum\limits_{i=1}^{m}(\mathbf{w}_i\boldsymbol{\theta}_{\text{fine},\mathbf{A}}-1)} \left[ |\mathbf{w}_i(\boldsymbol{\theta}_{\text{fine},\mathbf{A}}-\boldsymbol{\theta}_{\text{fine},\bar{\mathbf{A}}})| + \left| \frac{\sum\limits_{i=1}^{m}\mathbf{w}_i(\boldsymbol{\theta}_{\text{fine},\mathbf{A}}-\boldsymbol{\theta}_{\text{fine},\bar{\mathbf{A}}})}{\sum\limits_{i=1}^{m}\boldsymbol{\Phi}^{(i)}\text{diag}(\Upsilon)}\boldsymbol{\Phi}^{(i)}\text{diag}(\Upsilon) \right| \right] \\
&\leq \frac{1}{\sum\limits_{i=1}^{m}(\mathbf{w}_i\boldsymbol{\theta}_{\text{fine},\mathbf{A}}-1)} \left[ ||\boldsymbol{\theta}_{\text{fine},\mathbf{A}}-\boldsymbol{\theta}_{\text{fine},\bar{\mathbf{A}}}||_{\max} + \frac{m||\boldsymbol{\theta}_{\text{fine},\mathbf{A}}-\boldsymbol{\theta}_{\text{fine},\bar{\mathbf{A}}}||_{\max}}{m||\Upsilon||_{\min}}||\Upsilon||_{\max} \right] \\
&\leq \frac{1}{\sum\limits_{i=1}^{m}(\mathbf{w}_i\boldsymbol{\theta}_{\text{fine},\mathbf{A}}-1)} ||\boldsymbol{\theta}_{\text{fine},\mathbf{A}}-\boldsymbol{\theta}_{\text{fine},\bar{\mathbf{A}}}||_{\max} \left[ 1 + \frac{||\Upsilon||_{\max}}{||\Upsilon||_{\min}} \right] \\
&\leq \frac{1}{\sum\limits_{i=1}^{m}(\mathbf{w}_i\boldsymbol{\theta}_{\text{fine},\mathbf{A}}-1)} \alpha_n \left[ 1 + \frac{||\Upsilon||_{\max}}{||\Upsilon||_{\min}} \right],
\end{aligned}
$$
(65)

where (a) is due to Cauchy-Schwartz inequality, $\alpha_n$ is defined in Eq. 34, $\boldsymbol{\theta}_{\text{fine},\bar{\mathbf{A}}}$ is substituted from Eq. 58, and (b) is the result of the properties of the coarsening matrix in Definition 2. We use the

definition of $\boldsymbol{\theta}_{\text{fine},\mathbf{A}}$ in Eq. 57:

$$
\begin{aligned}
\sum_{i=1}^{m}(\mathbf{w}_i\boldsymbol{\theta}_{\text{fine},\mathbf{A}}-1) \quad &= \sum_{i=1}^{m}\mathbf{w}_i\text{diag}\left((\mathbf{I}-\mathbf{A}_{\text{nom}}^2)^{-1}-\mathbf{I}\right)\\
&= \mathbf{1}_m\mathbf{W}\text{diag}\left((\mathbf{I}-\mathbf{A}_{\text{nom}}^2)^{-1}-\mathbf{I}\right)\\
&= \mathbf{1}_m\mathbf{W}\text{diag}\left(\mathbf{A}_{\text{nom}}^2\right)\\
&= \mathbf{1}_m\mathbf{W}\text{diag}\left(\mathbf{A}_{\text{nom}}^2\right)\\
&= \tfrac{1}{c^2\cdot\text{tr}^2(\mathbf{A})}\mathbf{1}_m\mathbf{W}\text{diag}\left(\mathbf{A}^2\right)\\
&= \tfrac{1}{c^2\cdot\text{tr}^2(\mathbf{A})}\mathbf{1}_m\mathbf{W}\mathbf{A}\mathbf{1}_n\\
&\overset{(a)}{\geq} \tfrac{\tilde{c}_{\min}c_{\min}\rho_n mn||\mathbf{Q}^{(c)}||_{1,1}(1+\zeta)}{c^2(\zeta_{(u)}\rho_n n)^2}\\
&= \tfrac{\tilde{c}_{\min}c_{\min}||\mathbf{Q}^{(c)}||_{1,1}(1+\zeta)}{c^2\zeta_{(u)}^2}\tfrac{m}{\rho_n n}.
\end{aligned}
\tag{66}
$$

inequality (a) uses the event defined in Eq. 32 is used whose corresponding probability is Eq. 33. Using the definition of $\Upsilon$ in Eq. 16

$$
\begin{aligned}
||\Upsilon||_{\max} \quad &\leq ||\tfrac{\mathbf{Q}^{(c)}\mathbf{D}\mathbf{Q}^{(c)}}{\text{tr}(\mathbf{D}\mathbf{Q}^{(c)})}(\mathbf{I}-(\tfrac{\mathbf{D}\mathbf{Q}^{(c)}}{\text{tr}(\mathbf{D}\mathbf{Q}^{(c)})})^2)^{-1}||_{\max}\\
&\leq \tfrac{K||\mathbf{Q}^{(c)}\mathbf{D}\mathbf{Q}^{(c)}||_{\max}}{\text{tr}(\mathbf{D}\mathbf{Q}^{(c)})(1-(\tfrac{||\mathbf{D}\mathbf{Q}^{(c)}||_F}{\text{tr}(\mathbf{D}\mathbf{Q}^{(c)})})^2)}\\
&= \mathcal{O}(1)
\end{aligned}
\tag{67}
$$

$$
\begin{aligned}
||\Upsilon||_{\min} \quad &\geq ||\tfrac{\mathbf{Q}^{(c)}\mathbf{D}\mathbf{Q}^{(c)}}{\text{tr}(\mathbf{D}\mathbf{Q}^{(c)})}(\mathbf{I}-(\tfrac{\mathbf{D}\mathbf{Q}^{(c)}}{\text{tr}(\mathbf{D}\mathbf{Q}^{(c)})})^2)^{-1}||_{\min}\\
&\geq \tfrac{||\mathbf{Q}^{(c)}\mathbf{D}\mathbf{Q}^{(c)}||_{\min}}{\text{tr}(\mathbf{D}\mathbf{Q}^{(c)})}\\
&= \Omega(1),
\end{aligned}
\tag{68}
$$

where $\Omega(.)$ is the opposite scaling of $\mathcal{O}(.)$; we write $f(m)=\Omega(h(m))$ iff there exist positive reals $c_0$ and $m_0$ such that $|f(n)|\geq c_0 h(m)$ for all $m\geq m_0$.

We substitute Eq. 66 and Eq. 34 (with probability Eq. 24) into Eq. 65

$$
\Delta_i(\mathbf{A},\bar{\mathbf{A}}) \quad \leq \tfrac{\zeta_{(u)}^2}{\tilde{c}_{\min}c_{\min}||\mathbf{Q}^{(c)}||_{1,1}(1+\zeta)}\tfrac{[1+\tfrac{||\mathbf{Q}^{(c)}||_{\max}}{\text{tr}(\mathbf{D}\mathbf{Q}^{(c)})}][\tfrac{1}{\zeta_{(l)}}+\tfrac{\rho_n||\mathbf{Q}^{(c)}||_{\max}}{\text{tr}(\mathbf{D}\mathbf{Q}^{(c)})}]}{(1-\beta^2)\zeta_{(l)}\left[1-\tfrac{1}{c^2\cdot\text{tr}^2(\mathbf{D}\mathbf{Q}^{(c)})}\right]}\left[1+\tfrac{K||\mathbf{Q}^{(c)}\mathbf{D}\mathbf{Q}^{(c)}||_{\max}}{||\mathbf{Q}^{(c)}\mathbf{D}\mathbf{Q}^{(c)}||_{\min}(1-(\tfrac{||\mathbf{D}\mathbf{Q}^{(c)}||_F}{\text{tr}(\mathbf{D}\mathbf{Q}^{(c)})})^2)}\right]\tfrac{1}{\rho_n m}
\tag{69}
$$

with a joint probability of at least

$$
1-2\exp\left(-2n\rho_n^2(\text{tr}(\mathbf{D}\mathbf{Q}^{(c)})\zeta)^2\right)-\exp\left(-2\rho_n^2 mn(\tilde{c}_{\min}c_{\min}||\mathbf{Q}^{(c)}||_{1,1}\zeta)^2\right)
\tag{70}
$$

per Lemma 3. We then get the minimum of the two exponents in Eq. 70 which concludes the proof.

$\square$

### A.4.5 PROOF OF THEOREM 2

We start by using the triangle inequality for absolute values:

$$
\widehat{\Delta}_i(\widetilde{\mathbf{A}}) = \left|\tfrac{\hat{\boldsymbol{\theta}}_{\text{group}}^{(i)}-1}{\sum\limits_{i=1}^{m}(\hat{\boldsymbol{\theta}}_{\text{group}}^{(i)}-1)}-\tfrac{r\boldsymbol{\theta}_{\text{group},\mathbf{A}}^{(i)}-1}{\sum\limits_{i=1}^{m}(r\boldsymbol{\theta}_{\text{group},\mathbf{A}}^{(i)}-1)}\right|
$$

$$
\leq \underbrace{\left|\tfrac{r\boldsymbol{\theta}_{\text{group},\bar{\mathbf{A}}}^{(i)}-1}{\sum\limits_{i=1}^{m}(r\boldsymbol{\theta}_{\text{group},\bar{\mathbf{A}}}^{(i)}-1)}-\tfrac{r\boldsymbol{\theta}_{\text{group},\mathbf{A}}^{(i)}-1}{\sum\limits_{i=1}^{m}(r\boldsymbol{\theta}_{\text{group},\mathbf{A}}^{(i)}-1)}\right|}_{\Delta_i(\mathbf{A},\bar{\mathbf{A}})}+\underbrace{\left|\tfrac{\hat{\boldsymbol{\theta}}_{\text{group}}^{(i)}-1}{\sum\limits_{i=1}^{m}(\hat{\boldsymbol{\theta}}_{\text{group}}^{(i)}-1)}-\tfrac{r\boldsymbol{\theta}_{\text{group},\bar{\mathbf{A}}}^{(i)}-1}{\sum\limits_{i=1}^{m}(r\boldsymbol{\theta}_{\text{group},\bar{\mathbf{A}}}^{(i)}-1)}\right|}_{\triangleq\widehat{\Delta}_i(\tilde{\mathbf{A}},\bar{\mathbf{A}})}
\tag{71}
$$

The first term on the RHS of Eq. 71 has already been bounded in Thm.3. Next, we bound the second term in Eq. 71 and combine the two bounds at the end.

We substitute $\hat{\boldsymbol{\theta}}_{\text{group}}^{(i)}$ from the output of Algorithm 1, and $\boldsymbol{\theta}_{\text{group},\bar{\mathbf{A}}}^{(i)}$ from Eq. 60, for $\Upsilon$ defined in Eq. 16. For notation simplicity we write $\Upsilon(\mathbf{Q}^{(c)}, \mathbf{D})$ as $\Upsilon$ and $\Upsilon(\hat{\mathbf{Q}}^{(c)}, \hat{\mathbf{D}})$ as $\hat{\Upsilon}$ yields

$$
\widehat{\Delta}_i(\tilde{\mathbf{A}}, \bar{\mathbf{A}}) = \Big| \frac{\frac{\hat{\boldsymbol{\theta}}_{\text{group}}^{(i)} - 1}{m}}{\sum_{i=1}^{m}(\hat{\boldsymbol{\theta}}_{\text{group}}^{(i)} - 1)} - \frac{\frac{r\boldsymbol{\theta}_{\text{group},\bar{\mathbf{A}}}^{(i)} - 1}{m}}{\sum_{i=1}^{m}(r\boldsymbol{\theta}_{\text{group},\bar{\mathbf{A}}}^{(i)} - 1)} \Big| = \Bigg| \frac{\frac{\hat{\boldsymbol{\Phi}}^{(i)}\text{diag}(\hat{\Upsilon})}{m}}{\sum_{i=1}^{m}\hat{\boldsymbol{\Phi}}^{(i)}\text{diag}(\hat{\Upsilon})} - \frac{\frac{\boldsymbol{\Phi}^{(i)}\text{diag}(\Upsilon)}{m}}{\sum_{i=1}^{m}\boldsymbol{\Phi}^{(i)}\text{diag}(\Upsilon)} \Bigg|
$$

$$
= \frac{1}{\sum_{i=1}^{m}\hat{\boldsymbol{\Phi}}^{(i)}\text{diag}(\hat{\Upsilon})} \Bigg| \hat{\boldsymbol{\Phi}}^{(i)}\text{diag}(\hat{\Upsilon}) - \frac{\sum_{i=1}^{m}\hat{\boldsymbol{\Phi}}^{(i)}\text{diag}(\hat{\Upsilon})}{\sum_{i=1}^{m}\boldsymbol{\Phi}^{(i)}\text{diag}(\Upsilon)}\boldsymbol{\Phi}^{(i)}\text{diag}(\Upsilon) \Bigg| .
$$

$$(72)$$

We set $\hat{\boldsymbol{\Phi}}^{(i)} = \boldsymbol{\Phi}^{(i)} + \mathbf{E}_{\boldsymbol{\Phi}}, \hat{\mathbf{Q}}^{(c)} = \mathbf{Q}^{(c)} + \mathbf{E}_{\mathbf{Q}}, \hat{\mathbf{D}} = \mathbf{D} + \mathbf{E}_{\mathbf{D}}$, and $\hat{\Upsilon} = \Upsilon + \bar{E}$ where $\mathbf{E}_{\boldsymbol{\Phi}}, \mathbf{E}_{\mathbf{Q}}$ and $\mathbf{E}_{\mathbf{D}}$ are error matrices of appropriate sizes. Substitution of theses error matrices into Eq. 72, as well as multiple applications of the triangle inequality, gives:

$$
\widehat{\Delta}_i(\tilde{\mathbf{A}}, \bar{\mathbf{A}})
$$

$$
= \frac{1}{\sum_{i=1}^{m}\hat{\boldsymbol{\Phi}}^{(i)}\text{diag}(\hat{\Upsilon})} \Bigg| (\boldsymbol{\Phi}^{(i)} + \mathbf{E}_{\boldsymbol{\Phi}})\text{diag}(\Upsilon + \bar{E}) - \frac{\sum_{i=1}^{m}(\boldsymbol{\Phi}^{(i)} + \mathbf{E}_{\boldsymbol{\Phi}})\text{diag}(\Upsilon + \bar{E})}{\sum_{i=1}^{m}\boldsymbol{\Phi}^{(i)}\text{diag}(\Upsilon)}\boldsymbol{\Phi}^{(i)}\text{diag}(\Upsilon) \Bigg|
$$

$$
= \frac{1}{\sum_{i=1}^{m}\hat{\boldsymbol{\Phi}}^{(i)}\text{diag}(\hat{\Upsilon})} \Bigg| (\boldsymbol{\Phi}^{(i)} + \mathbf{E}_{\boldsymbol{\Phi}})[\text{diag}(\Upsilon) + \text{diag}(\bar{E})] - \frac{\sum_{i=1}^{m}(\boldsymbol{\Phi}^{(i)} + \mathbf{E}_{\boldsymbol{\Phi}})[\text{diag}(\Upsilon) + \text{diag}(\bar{E})]}{\sum_{i=1}^{m}\boldsymbol{\Phi}^{(i)}\text{diag}(\Upsilon)}\boldsymbol{\Phi}^{(i)}\text{diag}(\Upsilon) \Bigg|
$$

$$
= \frac{1}{\sum_{i=1}^{m}\hat{\boldsymbol{\Phi}}^{(i)}\text{diag}(\hat{\Upsilon})} \Bigg| \boldsymbol{\Phi}^{(i)}\text{diag}(\bar{E}) + \mathbf{E}_{\boldsymbol{\Phi}}[\text{diag}(\Upsilon) + \text{diag}(\bar{E})] - \frac{\sum_{i=1}^{m}\boldsymbol{\Phi}^{(i)}\text{diag}(\bar{E}) + m\mathbf{E}_{\boldsymbol{\Phi}}[\text{diag}(\Upsilon) + \text{diag}(\bar{E})]}{\sum_{i=1}^{m}\boldsymbol{\Phi}^{(i)}\text{diag}(\Upsilon)}\boldsymbol{\Phi}^{(i)}\text{diag}(\Upsilon) \Bigg|
$$

$$
= \frac{1}{\sum_{i=1}^{m}\hat{\boldsymbol{\Phi}}^{(i)}\text{diag}(\hat{\Upsilon})} \Bigg| [\boldsymbol{\Phi}^{(i)} + \mathbf{E}_{\boldsymbol{\Phi}}]\text{diag}(\bar{E}) + [\mathbf{E}_{\boldsymbol{\Phi}} - \frac{\sum_{i=1}^{m}\boldsymbol{\Phi}^{(i)}\text{diag}(\bar{E}) + m\mathbf{E}_{\boldsymbol{\Phi}}[\text{diag}(\Upsilon) + \text{diag}(\bar{E})]}{\sum_{i=1}^{m}\boldsymbol{\Phi}^{(i)}\text{diag}(\Upsilon)}\boldsymbol{\Phi}^{(i)}]\text{diag}(\Upsilon) \Bigg|
$$

$$
\leq \frac{1}{\sum_{i=1}^{m}\hat{\boldsymbol{\Phi}}^{(i)}\text{diag}(\hat{\Upsilon})} \Bigg[ (1 + ||\mathbf{E}_{\boldsymbol{\Phi}}||_1)||\bar{E}||_{\max} + [||\mathbf{E}_{\boldsymbol{\Phi}}||_1 + \frac{\sum_{i=1}^{m}\boldsymbol{\Phi}^{(i)}\text{diag}(\bar{E}) + m\mathbf{E}_{\boldsymbol{\Phi}}[\text{diag}(\Upsilon) + \text{diag}(\bar{E})]}{\sum_{i=1}^{m}\boldsymbol{\Phi}^{(i)}\text{diag}(\Upsilon)}\boldsymbol{\Phi}^{(i)}]||\Upsilon||_{\max} \Bigg] .
$$

$$(73)$$

To further simplify Eq. 73, we find upper bounds on the terms inside. The two terms $||\Upsilon||_{\max}$ and $||\Upsilon||_{\min}$ have already been bounded in Eq. 67 and Eq. 68 and we have:

$$
\frac{1}{\sum_{i=1}^{m}\hat{\boldsymbol{\Phi}}^{(i)}\text{diag}(\hat{\Upsilon})} \leq \frac{1}{m||\text{diag}(\hat{\Upsilon})||_{\min}}
$$

$$
\leq \frac{1}{m||\hat{\Upsilon}||_{\min}}
$$

$$
= \mathcal{O}(\frac{1}{m})
$$

$$(74)$$

The following inequality holds from the definition of norms:

$$
||\mathbf{E}_{\boldsymbol{\Phi}}||_1 \leq K||\mathbf{E}_{\boldsymbol{\Phi}}||_{\max}
$$

$$(75)$$

Using Eq. 74 and Eq. 75, the inner term in the last line of Eq. 73 is simplified as:

$$
\frac{\sum_{i=1}^{m} \mathbf{\Phi}^{(i)} \mathrm{diag}(\bar{E}) + m \mathbf{E}_{\mathbf{\Phi}}[\mathrm{diag}(\Upsilon) + \mathrm{diag}(\bar{E})]}{\sum_{i=1}^{m} \mathbf{\Phi}^{(i)} \mathrm{diag}(\Upsilon)} 
\begin{aligned}
&= \frac{\sum_{i=1}^{m} \mathbf{\Phi}^{(i)} \mathrm{diag}(\bar{E})}{\sum_{i=1}^{m} \mathbf{\Phi}^{(i)} \mathrm{diag}(\Upsilon)} + m \frac{\mathbf{E}_{\mathbf{\Phi}}[\mathrm{diag}(\Upsilon) + \mathrm{diag}(\bar{E})]}{\sum_{i=1}^{m} \mathbf{\Phi}^{(i)} \mathrm{diag}(\Upsilon)} \\
&\leq \frac{m||\mathrm{diag}(\bar{E})||_{\max}}{m||\mathrm{diag}(\Upsilon)||_{\min}} + mK \frac{||\mathbf{E}_{\mathbf{\Phi}}||_{\max}[||\mathrm{diag}(\Upsilon)||_{\max} + ||\mathrm{diag}(\bar{E})||_{\max}]}{m||\mathrm{diag}(\Upsilon)||_{\min}} \\
&\leq \frac{||\bar{E}||_{\max}}{||\Upsilon||_{\min}} + K \frac{||\mathbf{E}_{\mathbf{\Phi}}||_{\max}[||\Upsilon||_{\max} + ||\bar{E}||_{\max}]}{||\Upsilon||_{\min}} \\
&= \mathcal{O}(||\mathbf{E}_{\mathbf{\Phi}}||_{\max} + ||\bar{E}||_{\max})
\end{aligned}
\tag{76}
$$

Replacing Eq. 67, Eq. 68, Eq. 74, Eq. 75, and Eq. 76 into Eq. 73 simplifies it as:

$$
\left| \frac{\frac{\hat{\boldsymbol{\theta}}_{\mathrm{group}}^{(i)} - 1}{m}}{\sum_{i=1}^{m}(\hat{\boldsymbol{\theta}}_{\mathrm{group}}^{(i)} - 1)} - \frac{\frac{r\boldsymbol{\theta}_{\mathrm{group},\bar{\mathbf{A}}}^{(i)} - 1}{m}}{\sum_{i=1}^{m}(r\boldsymbol{\theta}_{\mathrm{group},\bar{\mathbf{A}}}^{(i)} - 1)} \right| = \mathcal{O}\left(\frac{1}{m}[||\mathbf{E}_{\mathbf{\Phi}}||_{\max} + ||\bar{E}||_{\max}]\right).
\tag{77}
$$

We now simplify the term $||\bar{E}||_{\max}$ in Eq. 79:

$$
\begin{aligned}
||\bar{E}||_{\max} &= ||\hat{\Upsilon} - \Upsilon||_{\max} \\
&= \left|\left| \frac{\hat{\mathbf{Q}}^{(c)} \hat{\mathbf{D}} \hat{\mathbf{Q}}^{(c)}}{\mathrm{tr}(\hat{\mathbf{D}}\hat{\mathbf{Q}}^{(c)})} \left(\mathbf{I} - \left(\frac{\hat{\mathbf{D}}\hat{\mathbf{Q}}^{(c)}}{\mathrm{tr}(\hat{\mathbf{D}}\hat{\mathbf{Q}}^{(c)})}\right)^2\right)^{-1} - \frac{\mathbf{Q}^{(c)} \mathbf{D} \mathbf{Q}^{(c)}}{\mathrm{tr}(\mathbf{D}\mathbf{Q}^{(c)})} \left(\mathbf{I} - \left(\frac{\mathbf{D}\mathbf{Q}^{(c)}}{\mathrm{tr}(\mathbf{D}\mathbf{Q}^{(c)})}\right)^2\right)^{-1} \right|\right|_{\max} \\
&= \frac{1}{\mathrm{tr}(\hat{\mathbf{D}}\hat{\mathbf{Q}}^{(c)})} \left|\left| \hat{\mathbf{Q}}^{(c)} \hat{\mathbf{D}} \hat{\mathbf{Q}}^{(c)} \left(\mathbf{I} - \left(\frac{\hat{\mathbf{D}}\hat{\mathbf{Q}}^{(c)}}{\mathrm{tr}(\hat{\mathbf{D}}\hat{\mathbf{Q}}^{(c)})}\right)^2\right)^{-1} - \frac{\mathrm{tr}(\hat{\mathbf{D}}\hat{\mathbf{Q}}^{(c)})}{\mathrm{tr}(\mathbf{D}\mathbf{Q}^{(c)})} \mathbf{Q}^{(c)} \mathbf{D} \mathbf{Q}^{(c)} \left(\mathbf{I} - \left(\frac{\mathbf{D}\mathbf{Q}^{(c)}}{\mathrm{tr}(\mathbf{D}\mathbf{Q}^{(c)})}\right)^2\right)^{-1} \right|\right|_{\max}.
\end{aligned}
\tag{78}
$$

To continue the simplification of $||\bar{E}||_{\max}$, we bring the two error matrices $\mathbf{E}_{\mathbf{Q}}$ and $\mathbf{E}_{\mathbf{D}}$ introduced in the statement of the theorem into play:

$$
\begin{aligned}
(\mathbf{Q}^{(c)} + \mathbf{E}_{\mathbf{Q}}) \quad (\mathbf{D} + \mathbf{E}_{\mathbf{D}})(\mathbf{Q}^{(c)} + \mathbf{E}_{\mathbf{Q}}) &= (\mathbf{Q}^{(c)} + \mathbf{E}_{\mathbf{Q}})(\mathbf{D}\mathbf{Q}^{(c)} + \mathbf{D}\mathbf{E}_{\mathbf{Q}} + \mathbf{E}_{\mathbf{D}}\mathbf{Q}^{(c)} + \mathbf{E}_{\mathbf{D}}\mathbf{E}_{\mathbf{Q}}) \\
&= \mathbf{Q}^{(c)}\mathbf{D}\mathbf{Q}^{(c)} + \mathbf{\Gamma},
\end{aligned}
\tag{79}
$$

where we define

$$
\mathbf{\Gamma} \triangleq \mathbf{Q}^{(c)}\mathbf{D}\mathbf{E}_{\mathbf{Q}} + \mathbf{Q}^{(c)}\mathbf{E}_{\mathbf{D}}\mathbf{Q}^{(c)} + \mathbf{Q}^{(c)}\mathbf{E}_{\mathbf{D}}\mathbf{E}_{\mathbf{Q}} + \mathbf{E}_{\mathbf{Q}}\mathbf{D}\mathbf{Q}^{(c)} + \mathbf{E}_{\mathbf{Q}}\mathbf{D}\mathbf{E}_{\mathbf{Q}} + \mathbf{E}_{\mathbf{Q}}\mathbf{E}_{\mathbf{D}}\mathbf{Q}^{(c)} + \mathbf{E}_{\mathbf{Q}}\mathbf{E}_{\mathbf{D}}\mathbf{E}_{\mathbf{Q}}
\tag{80}
$$

and

$$
(\mathbf{D} + \mathbf{E}_{\mathbf{D}}) \quad (\mathbf{Q}^{(c)} + \mathbf{E}_{\mathbf{Q}}) = \mathbf{D}\mathbf{Q}^{(c)} + \underbrace{\mathbf{D}\mathbf{E}_{\mathbf{Q}} + \mathbf{E}_{\mathbf{D}}\mathbf{Q}^{(c)} + \mathbf{E}_{\mathbf{D}}\mathbf{E}_{\mathbf{Q}}}_{\mathfrak{E}}.
\tag{81}
$$

Replacing Eq. 79 and Eq. 81 into Eq. 78 yields:

$$
\begin{aligned}
||\bar{E}||_{\max} &= \frac{1}{\mathrm{tr}(\hat{\mathbf{D}}\hat{\mathbf{Q}}^{(c)})} ||(\mathbf{Q}^{(c)}\mathbf{D}\mathbf{Q}^{(c)} + \mathbf{\Gamma})\left(\mathbf{I} - \left(\frac{\hat{\mathbf{D}}\hat{\mathbf{Q}}^{(c)}}{\mathrm{tr}(\hat{\mathbf{D}}\hat{\mathbf{Q}}^{(c)})}\right)^2\right)^{-1} - \frac{\mathrm{tr}(\mathbf{D}\mathbf{Q}^{(c)} + \mathfrak{E})}{\mathrm{tr}(\mathbf{D}\mathbf{Q}^{(c)})} \mathbf{Q}^{(c)}\mathbf{D}\mathbf{Q}^{(c)} \left(\mathbf{I} - \left(\frac{\mathbf{D}\mathbf{Q}^{(c)}}{\mathrm{tr}(\mathbf{D}\mathbf{Q}^{(c)})}\right)^2\right)^{-1}||_{\max} \\
&= \frac{1}{\mathrm{tr}(\hat{\mathbf{D}}\hat{\mathbf{Q}}^{(c)})} ||\mathbf{Q}^{(c)}\mathbf{D}\mathbf{Q}^{(c)}\left[\left(\mathbf{I} - \left(\frac{\hat{\mathbf{D}}\hat{\mathbf{Q}}^{(c)}}{\mathrm{tr}(\hat{\mathbf{D}}\hat{\mathbf{Q}}^{(c)})}\right)^2\right)^{-1} - \left(\mathbf{I} - \left(\frac{\mathbf{D}\mathbf{Q}^{(c)}}{\mathrm{tr}(\mathbf{D}\mathbf{Q}^{(c)})}\right)^2\right)^{-1}\right] \\
&\qquad + \mathbf{\Gamma}\left(\mathbf{I} - \left(\frac{\hat{\mathbf{D}}\hat{\mathbf{Q}}^{(c)}}{\mathrm{tr}(\hat{\mathbf{D}}\hat{\mathbf{Q}}^{(c)})}\right)^2\right)^{-1} - \frac{\mathrm{tr}(\mathfrak{E})}{\mathrm{tr}(\mathbf{D}\mathbf{Q}^{(c)})} \mathbf{Q}^{(c)}\mathbf{D}\mathbf{Q}^{(c)}\left(\mathbf{I} - \left(\frac{\mathbf{D}\mathbf{Q}^{(c)}}{\mathrm{tr}(\mathbf{D}\mathbf{Q}^{(c)})}\right)^2\right)^{-1}||_{\max} \\
&= \frac{1}{\mathrm{tr}(\hat{\mathbf{D}}\hat{\mathbf{Q}}^{(c)})} [||\mathbf{Q}^{(c)}\mathbf{D}\mathbf{Q}^{(c)}\underbrace{\left[\sum_{\ell=1}^{\infty}\left(\frac{\hat{\mathbf{D}}\hat{\mathbf{Q}}^{(c)}}{\mathrm{tr}(\hat{\mathbf{D}}\hat{\mathbf{Q}}^{(c)})}\right)^{2\ell} - \left(\frac{\mathbf{D}\mathbf{Q}^{(c)}}{\mathrm{tr}(\mathbf{D}\mathbf{Q}^{(c)})}\right)^{2\ell}\right]}_{\triangleq \epsilon_1} \\
&\qquad + \mathbf{\Gamma}\left(\mathbf{I} - \left(\frac{\hat{\mathbf{D}}\hat{\mathbf{Q}}^{(c)}}{\mathrm{tr}(\hat{\mathbf{D}}\hat{\mathbf{Q}}^{(c)})}\right)^2\right)^{-1} - \frac{\mathrm{tr}(\mathfrak{E})}{\mathrm{tr}(\mathbf{D}\mathbf{Q}^{(c)})} \mathbf{Q}^{(c)}\mathbf{D}\mathbf{Q}^{(c)}\left(\mathbf{I} - \left(\frac{\mathbf{D}\mathbf{Q}^{(c)}}{\mathrm{tr}(\mathbf{D}\mathbf{Q}^{(c)})}\right)^2\right)^{-1}||_{\max}] \\
&\leq \frac{K}{\mathrm{tr}(\hat{\mathbf{D}}\hat{\mathbf{Q}}^{(c)})} [||\mathbf{Q}^{(c)}\mathbf{D}\mathbf{Q}^{(c)}||_{\max}||\epsilon_1||_{\max} + ||\mathbf{\Gamma}||_{\max}||\left(\mathbf{I} - \left(\frac{\hat{\mathbf{D}}\hat{\mathbf{Q}}^{(c)}}{\mathrm{tr}(\hat{\mathbf{D}}\hat{\mathbf{Q}}^{(c)})}\right)^2\right)^{-1}||_{\max} \\
&\qquad + \frac{\mathrm{tr}(\mathfrak{E})}{\mathrm{tr}(\mathbf{D}\mathbf{Q}^{(c)})} ||\mathbf{Q}^{(c)}\mathbf{D}\mathbf{Q}^{(c)}||_{\max}||\left(\mathbf{I} - \left(\frac{\mathbf{D}\mathbf{Q}^{(c)}}{\mathrm{tr}(\mathbf{D}\mathbf{Q}^{(c)})}\right)^2\right)^{-1}||_{\max}] \\
&= \mathcal{O}(||\epsilon_1||_{\max} + ||\mathbf{\Gamma}||_{\max} + \mathrm{tr}(\mathfrak{E})) \\
&= \mathcal{O}(||\epsilon_1||_{\max} + ||\mathbf{\Gamma}||_{\max} + ||\mathfrak{E}||_{\max})
\end{aligned}
\tag{82}
$$

We define another error term:

$$\left(\mathbf{D}\mathbf{Q}^{(c)} + \mathfrak{E}\right)^{2\ell} = (\mathbf{D}\mathbf{Q}^{(c)})^{2\ell} + \underbrace{\sum_{\nu=1}^{2\ell} \cdots}_{\triangleq \mathcal{R} = \mathcal{O}(\mathfrak{E})} . \tag{83}$$

We can then simplify $\epsilon_1$ for $\ell \geq 1$ as:

$$
\begin{aligned}
||\epsilon_1||_{\max} &= ||\sum_{\ell=1}^{\infty} \frac{1}{(\mathrm{tr}(\hat{\mathbf{D}}\hat{\mathbf{Q}}^{(c)}))^{2\ell}} \left[ (\hat{\mathbf{D}}\hat{\mathbf{Q}}^{(c)})^{2\ell} - (\frac{\mathrm{tr}(\hat{\mathbf{D}}\hat{\mathbf{Q}}^{(c)})}{\mathrm{tr}(\mathbf{D}\mathbf{Q}^{(c)})})^{2\ell}(\mathbf{D}\mathbf{Q}^{(c)})^{2\ell} \right] ||_{\max} \\
&= ||\sum_{\ell=1}^{\infty} \frac{1}{(\mathrm{tr}(\hat{\mathbf{D}}\hat{\mathbf{Q}}^{(c)}))^{2\ell}} \left[ (\mathbf{D}\mathbf{Q}^{(c)})^{2\ell} + \mathcal{R} - (1 + \frac{\mathrm{tr}(\mathcal{R})}{\mathrm{tr}(\mathbf{D}\mathbf{Q}^{(c)})})^{2\ell}(\mathbf{D}\mathbf{Q}^{(c)})^{2\ell} \right] ||_{\max} \\
&= ||\sum_{\ell=1}^{\infty} \frac{1}{(\mathrm{tr}(\hat{\mathbf{D}}\hat{\mathbf{Q}}^{(c)}))^{2\ell}} \left[ \mathcal{R} - (\frac{\mathrm{tr}(\mathcal{R})}{\mathrm{tr}(\mathbf{D}\mathbf{Q}^{(c)})})^{2\ell}(\mathbf{D}\mathbf{Q}^{(c)})^{2\ell} \right] ||_{\max} \\
&= ||[(1 - (\frac{1}{\mathrm{tr}(\hat{\mathbf{D}}\hat{\mathbf{Q}}^{(c)})})^2)^{-1} - 1]\mathcal{R} - \sum_{\ell=1}^{\infty} \left[ (\frac{\mathrm{tr}(\mathcal{R})}{\mathrm{tr}(\mathbf{D}\mathbf{Q}^{(c)})\mathrm{tr}(\hat{\mathbf{D}}\hat{\mathbf{Q}}^{(c)})})^{2\ell}(\mathbf{D}\mathbf{Q}^{(c)})^{2\ell} \right] ||_{\max} \\
&= ||[(1 - (\frac{1}{\mathrm{tr}(\hat{\mathbf{D}}\hat{\mathbf{Q}}^{(c)})})^2)^{-1} - 1]\mathcal{R} - [(\mathbf{I} - (\frac{\mathrm{tr}(\mathcal{R})}{\mathrm{tr}(\mathbf{D}\mathbf{Q}^{(c)})\mathrm{tr}(\hat{\mathbf{D}}\hat{\mathbf{Q}}^{(c)})}\mathbf{D}\mathbf{Q}^{(c)})^2)^{-1} - \mathbf{I}]||_{\max} \\
&\leq [(1 - (\frac{1}{\mathrm{tr}(\hat{\mathbf{D}}\hat{\mathbf{Q}}^{(c)})})^2)^{-1} - 1]||\mathcal{R}||_{\max} + ||[(\mathbf{I} - (\frac{\mathrm{tr}(\mathcal{R})}{\mathrm{tr}(\mathbf{D}\mathbf{Q}^{(c)})\mathrm{tr}(\hat{\mathbf{D}}\hat{\mathbf{Q}}^{(c)})}\mathbf{D}\mathbf{Q}^{(c)})^2)^{-1} - \mathbf{I}]||_{\max} \\
&= \mathcal{O}(||\mathcal{R}||_{\max}) \\
&= \mathcal{O}(||\mathfrak{E}||_{\max})
\end{aligned}
\tag{84}
$$

Similarly, we can rewrite an upper bound on $||\mathbf{\Gamma}||_{\max}$ as:

$$
\begin{aligned}
||\mathbf{\Gamma}||_{\max} &= K^2(||\mathbf{Q}^{(c)}\mathbf{D}||_{\max}||\mathbf{E_Q}||_{\max} + ||\mathbf{Q}^{(c)}||_{\max}||\mathbf{E_D}||_{\max}||\mathbf{Q}||_{\max} + ||\mathbf{Q}^{(c)}||_{\max}||\mathbf{E_D}||_{\max}||\mathbf{E_Q}||_{\max} \\
&\qquad + ||\mathbf{E_Q}||_{\max}||\mathbf{D}\mathbf{Q}^{(c)}||_{\max} + ||\mathbf{E_Q}||_{\max}||\mathbf{D}||_{\max}||\mathbf{E_Q}||_{\max} \\
&\qquad\qquad + ||\mathbf{E_Q}||_{\max}||\mathbf{E_D}||_{\max}||\mathbf{Q}^{(c)}||_{\max} + ||\mathbf{E_Q}||_{\max}||\mathbf{E_D}||_{\max}||\mathbf{E_Q}||_{\max}) \\
&= K^2(||\mathbf{E_Q}||_{\max} + ||\mathbf{E_D}||_{\max} + ||\mathbf{E_D}||_{\max}||\mathbf{E_Q}||_{\max} + ||\mathbf{E_Q}||_{\max} + ||\mathbf{E_Q}||_{\max}^2 \\
&\qquad + ||\mathbf{E_Q}||_{\max}||\mathbf{E_D}||_{\max} + ||\mathbf{E_Q}||_{\max}^2||\mathbf{E_D}||_{\max}) \\
&= \mathcal{O}(||\mathbf{E_Q}||_{\max} + ||\mathbf{E_D}||_{\max}),
\end{aligned}
\tag{85}
$$

and

$$
\begin{aligned}
||\mathfrak{E}||_{\max} &= ||\mathbf{D}\mathbf{E_Q} + \mathbf{E_D}\mathbf{Q}^{(c)} + \mathbf{E_D}\mathbf{E_Q}||_{\max} \\
&= K(||\mathbf{D}||_{\max}||\mathbf{E_Q}||_{\max} + ||\mathbf{E_D}||_{\max}||\mathbf{Q}^{(c)}||_{\max} + ||\mathbf{E_D}||_{\max}||\mathbf{E_Q}||_{\max}) \\
&= K(||\mathbf{E_Q}||_{\max} + ||\mathbf{E_D}||_{\max} + ||\mathbf{E_D}||_{\max}||\mathbf{E_Q}||_{\max}) \\
&= \mathcal{O}(||\mathbf{E_Q}||_{\max} + ||\mathbf{E_D}||_{\max}).
\end{aligned}
\tag{86}
$$

By substituting Eq. 84, Eq. 86, and Eq. 85 into Eq. 82, we get:

$$||\bar{E}||_{\max} = \mathcal{O}(||\mathbf{E_Q}||_{\max} + ||\mathbf{E_D}||_{\max}). \tag{87}$$

Replacing Eq. 87 into the original error in Eq. 79 yields:

$$\left| \frac{\frac{\hat{\boldsymbol{\theta}}_{\mathrm{group}}^{(i)}-1}{m}}{\sum_{i=1}^{m}(\hat{\boldsymbol{\theta}}_{\mathrm{group}}^{(i)} - 1)} - \frac{\frac{r\boldsymbol{\theta}_{\mathrm{group},\bar{\mathbf{A}}}^{(i)}-1}{m}}{\sum_{i=1}^{m}(r\boldsymbol{\theta}_{\mathrm{group},\bar{\mathbf{A}}}^{(i)} - 1)} \right| = \mathcal{O}\left( \frac{1}{m}[||\mathbf{E_\Phi}||_{\max} + ||\mathbf{E_Q}||_{\max} + ||\mathbf{E_D}||_{\max}] \right). \tag{88}$$

which happens with the same lower bound probability as in Eq. 70. The proof is now complete.

