# OpenReview forum: "Quantifying the Controllability of Coarsely Characterized Networked Dynamical Systems"
_ICLR.cc/2022/Conference — ICLR 2022 Submitted_

### Official Review · Reviewer_AEYY · 2021-11-01

**Correctness:** 3
**Technical Novelty And Significance:** 3
**Empirical Novelty And Significance:** 3
**Recommendation:** 6
**Confidence:** 4

**Main Review:**

In general, I think this is an interesting topic, which has so far received not a lot of attention in the literature. Learning network properties from aggregate descriptions is clearly of relevance in a number of settings.

However, I have several reservations about some of the results presented here.

The motivation in terms of brain networks and linear control system is questionable, I think. Yes, there is a large literature now that uses linear control theory to study brain networks; but the brain is clearly a nonlinear system.

It seems the authors have neglected that there exists a branch in the literature concerned with "blind estimation" of community structure (such as SBMs) and other network properties, which has a very similar flavor to the work presented here. I think this should be discussed in more detail. See, e.g.,
Wai, Hoi-To, et al. "Blind community detection from low-rank excitations of a graph filter." IEEE Transactions on signal processing 68 (2019): 436-451.
Roddenberry, T. Mitchell, et al. "Exact blind community detection from signals on multiple graphs." IEEE Transactions on Signal Processing 68 (2020): 5016-5030.
Schaub, Michael T., Santiago Segarra, and John N. Tsitsiklis. "Blind identification of stochastic block models from dynamical observations." SIAM Journal on Mathematics of Data Science 2.2 (2020): 335-367.

Further, while the choice of the SBM is understandable from a theoretical perspective, from a practical point of view this is highly questionable: SBMs (and graphons) more generally, lead to dense graphs, while in practice we encounter highly sparse graphs. One could at least consider the degree corrected variant of the SBM.
Moreover, SBMs will not concentrate well in spectral norm in the "sparse regime" (node degrees being constant as the number of nodes increases), which is problematic for the setup considered here.
I think the authors should at least discuss these issues in more detail.

Assumption 4 would merit more discussion -- it seems to a pretty strong assumption.


There are also some flaws in the technical presentation.

The adjacency matrix of a network has zero diagonal, hence normalizing by the trace is a bad idea -- the authors effectively assume the existence of self-loops here (presumably from the low-rank SBM assumption). For the coarse grained version (which will have self-loops) this is not a problem.

Similarly, the expression for the expectation of the SBM corresponds to a version in which self-loops are allowed -- though this is not unusual in the literature this should at least be commented upon.

I am not sure that the statement about the controllability for T vs n steps is correct. This is true for continuous time systems, but for discrete time systems it it seems choosing a small T could prevent a system from being fully controllable, which is controllable for T=n.


**Summary Of The Paper:**

The paper studies the problem of quantifying certain controllability metrics, based on coarse scale observations of a network, which is assume to have community structure.
The main contributions are some theoretical characterizations on when and how well this is possible, given the chosen setup.

**Summary Of The Review:**

The paper makes progress in an interesting direction: learning of network properties from aggregate observations.

What should be improved in a revised version  (see above)
a) the relations to previous work in the literature should be explained more clearly (ideas from concentration of measure, blind community detection and MOR)
b) a more careful discussion about the SBM and the weaknesses of this modelling assumption should be given.
c) a number of technical mistakes should be fixed

---

> ### Author Response · Authors · 2021-11-20
> **We discussed the linearity assumption of brain networked systems. We commented on the connection between our work and that of the blind estimation of community structure. We also explained the reason behind the SBM variant choice in our work.**
>
> *[R3: 1] The motivation in terms of brain networks and linear control system is questionable as the brain is clearly a nonlinear system.*
>
> We appreciate the reviewer for bringing up this point. A phenomenal amount of literature has focused on modeling brain network behaviour using LTIs [1,2]. We repeat the reasons behind such linearization using the arguments in [1]: First, linearization is an accurate approximation of non-linear systems in certain scenarios, such as small-scale deviations around an operating point.  Second, the controllability of a linearized system renders local controllability for the non-linear system. We ensure to add these complementary explanations to the updated paper.
>
> *[R3: 2] Connection to the "blind estimation" of community structure (such as SBMs) and other network properties is missing*
>
> We thank the reviewer for the suggestion of further related literature. We acknowledge that the aforementioned papers are very interesting, to some extent relate to our work, and will add them to our literature review. However, there exists two main differences between this line of work and ours:
>
> First, there is no indication of controllability derivation in the blind community recovery; the blind community recovery is all about methods of detecting communities from a graph. These methods, similar to many other existing methods (c.f. the first paragraph of Section 6), have essentially similar outcomes as what we get from Mao’s spectral clustering-based method, i.e. to recover mixed-membership communities given an input graph [3]. We would only apply this method, or similar approaches, as a part of our algorithm to obtain the end goal: computing coarse controllability from the proposed learning-based method. We will add these references to the introduction paragraph of Section 6.
>
> Second, the blind community recovery methods rely on multiple data points (signals or samples) on the graph so as to estimate its covariance matrix. The covariance matrix is then regarded as the adjacency matrix. In contrast, in our paper, we assume a one-shot topology of the graph. In that sense, we do not necessarily need multiple samples per graph.  One graph is sufficient from which the graph edges can be considered as samples.
>
> *[R3: 3] While the choice of the SBM is understandable from a theoretical perspective, from a practical point of view this is highly questionable: SBMs (and graphons) more generally, lead to dense graphs, while in practice we encounter highly sparse graphs.*
>
> Thank you for the apropos comment. SBMs usually include a parameter that controls the sparsity of the graph. The parameter decreases as the graph size grows (i.e. larger graphs become sparser) and the scaling behaviour is Ω(1/n) or Ω(log n /n). We appreciate if the reviewer could elaborate on what scenarios they have in mind that SBMs generate dense graphs necessarily.
>
> *[R3: 4] One could at least consider the degree corrected variant of the SBM. Moreover, SBMs will not concentrate well in spectral norm in the "sparse regime" (node degrees being constant as the number of nodes increases), which is problematic for the setup considered here. I think the authors should at least discuss these issues in more detail.*
>
> We thank the reviewer for bringing up this point. The vanilla SBM we used in the paper, although restrictive, provides a powerful modeling framework to facilitate fundamental understanding of the graph community organization, a predominant feature in many real-world networks. We agree that other, more sophisticated versions of SBMs, provide richer and more realistic modeling frameworks. Nevertheless, as a first work characterizing coarse controllability of SBMs, we focused on the vanilla model and studying the variants are postponed to future work.
> We make sure we clarify the response to concern in the final version.
>
>
> *[1] Gu, Shi, et al. "Controllability of structural brain networks." Nature communications 6.1 (2015): 1-10.*
>
> *[2] Tang, Evelyn, et al. "Control of brain network dynamics across diverse scales of space and time." Physical Review E 101.6 (2020): 062301.*
>
> *[3] Mao, X., Sarkar, P., & Chakrabarti, D. (2020). Estimating mixed memberships with sharp eigenvector deviations. Journal of the American Statistical Association, 1-13.*

---

> > ### Author Response · Authors · 2021-11-20
> > **We commented on the reasoning behind Assumption 4, as well as the mistype of a word after controllability definition.**
> >
> > *[R3: 5] Assumption 4 would merit more discussion -- it seems to a pretty strong assumption.*
> >
> > We appreciate the reviewer shedding light on this point. Assumption 4 indicates controllability of networks from groups of nodes that are mapped to c-nodes. Note that this is weaker than asking a fine network to be controllable from every single node. At the heart of Assumption 4 are assumptions in [1], that brain networks are controllable from single regions, though this may require large input energy (equivalent to the smallest eigenvalue being nonzero, but very small). We consider controllability from a single region (which includes multiple fine nodes) as a starting point of developing the theory. See also Remark 2. For future work, we consider generalizing to multi-region controllability, which is known to be a difficult combinatorial problem [4]. We will include this complementary point to the final paper.
> >
> > *[R3: 6] There are also some flaws in the technical presentation.*
> >
> > We will certainly do another detailed pass on the document to resolve the possible errors. We appreciate it if the reviewer can point out the specific parts where they have seen erroneous presentations.
> >
> > *[R3: 7] I am not sure that the statement about the controllability for T vs n steps is correct. This is true for continuous time systems, but for discrete time systems it seems choosing a small T could prevent a system from being fully controllable, which is controllable for T=n.*
> >
> > Thanks for the great detailed attention to this statement. If we understand your question correctly, it is about the “iff” statement that follows after our controllability definition. “Iff” must be changed to a simple “if”, since, as correctly
> >
> > *[4] Summers, Tyler H., Fabrizio L. Cortesi, and John Lygeros. "On submodularity and controllability in complex dynamical networks." IEEE Transactions on Control of Network Systems 3.1 (2015): 91-101.*

---

> > > ### Author Response · Authors · 2021-11-20
> > > **We discussed our choice of the normalization factor of the adjacency matrix. We also provided complementary explanations for the self-loop existence assumption.**
> > >
> > > *[R3: 8] The adjacency matrix of a network has zero diagonal, hence normalizing by the trace is a bad idea -- the authors effectively assume the existence of self-loops here (presumably from the low-rank SBM assumption). For the coarse grained version (which will have self-loops) this is not a problem. Similarly, the expression for the expectation of the SBM corresponds to a version in which self-loops are allowed -- though this is not unusual in the literature this should at least be commented upon.*
> > >
> > > This is a great point. Prior to submission, we had a remark dedicated solely to clarify this point. However, we had to remove that remark due to space constraints. The remark, which will be added to the final version, is as follows:
> > >
> > > **Remark:** The normalization factor $\frac{1}{c \cdot \text{tr}(\mathbf{A})}$ in Eq.9 plays a key role in our analysis. Of course one may use other normalization factors including $\frac{1}{ρ(\mathbf{A})+ϵ}$, for ϵ >0, where ρ(.) denotes the spectral radius. However, as we work with large networks realized by an SBM, we need a factor that scales with the network size $n$ and is amenable to our analysis. Thus, we chose $\frac{1}{c \cdot \text{tr}(\mathbf{A})}$ as the normalization factor. The choice of trace as a normalization coefficient has multiple advantages. First it helps bring out the dominant dimensions when analysing the system. Second, it contributes to simplifying bound derivations. We should note that such normalization stabilizes systems iff they satisfy
> > > $\frac{|\lambda_{\text{max}}(\mathbf{A})|}{\sum_{\ell}\lambda_\ell(\mathbf{A})}<1$.
> > >
> > > There are various coefficients that would maintain the stability of system dynamics even when A itself is not in conformity with the aforementioned condition. For instance we can replace $\frac{1}{\text{tr}(A)}\mathbf{A}$ with $\frac{1}{\text{tr}(A)} \mathbf{A} -c_1I$ for a constant $c_1$. We can also use $\frac{1}{c_2 \text{tr}(\mathbf{A})} A$ for a constant $c_2>1$ that would ensure $\frac{|\lambda_{\text{max}}(\mathbf{A})|}{c_2\displaystyle\sum_{\ell}\lambda_\ell(\mathbf{A})}<1$. Nevertheless, to keep the derivations in the paper tractable, we stick with $\frac{1}{c \cdot \text{tr}(\mathbf{A})}$  as the stabilizing coefficient, and we assume $\|\frac{1}{c \cdot \text{tr}(A)}\mathbf{A}\|_2\leq \beta< 1$. 
> > >
> > > Regarding the *self-loop* assumption in SBMs vs brain network, please see our answer to Reviewer 2 [R2: 4]: First, we should point out that the existence of self-loops usually depends on the modeling choice. E.g. brain networks computed from e.g. brain imaging data contain self-loops [2,3]. The reason for the existence of these self-loops is that regions of interest (ROIs) when parcelling the brain imaging data cannot be very small, and so each ROI covers a number of neurons that have inter-connections. Hence, SBM (i.e. with self-loops) is still a valid modeling for the brain networks. However, as correctly mentioned by the reviewer, it is common to remove the self-loops when a brain graph is used in an LTI system dynamics [2,3]. This way, paper derivations remain mainly the same except for minor modifications.
> > > From the Remark, we can see that the normalization coefficient $\frac{1}{c \cdot \text{tr}(\mathbf{A})}$ in Eq.9 is important since it needs to scale with $\frac{1}{n}$ and that it facilitates error characterization analysis. When we’re not seeking error characterization, other normalization coefficients that enjoy similar scaling behaviour include standard $\frac{1}{ρ(\mathbf{A})+1}$, $\frac{1}{||A||_2}$, $\frac{1}{||A||_\infty}$.

---

> > > > ### Comment · Reviewer_AEYY · 2021-11-29
> > > > **Thanks for the rebuttal**
> > > >
> > > > I would like to thank the authors for their responses. While my assessment has not changed in a major way, I still think the paper is above the threshold for acceptance.

---

### Official Review · Reviewer_SdpL · 2021-11-02

**Correctness:** 3
**Technical Novelty And Significance:** 3
**Empirical Novelty And Significance:** Not applicable
**Recommendation:** 6
**Confidence:** 4

**Main Review:**

Strengths: It provides a clue about how to infer the global property of network controllability with partial knowledge of the connectome. As interdisciplinary research linking control theory with network science, this work is inspiring with relatively rigorous mathematical support.

Weakness:
1. The scope is a bit limited for the computer science community. The framework can be described in a more general sense.
2. Since the network controllability has its application in practice, it would be better if the authors can provide illustrative examples on when such kind of approximation is needed.
3. Is there any statistical significance for the results shown in Figure 2?
4. The assumptions on the diagonal elements are not consistent throughout the article. In the case of brain networks, the self-loop does not exist. In the case of the SBMs, the self-loop does exist from the probabilistic assumption. Indeed, the stability of Eqn (11) only requires a parameter to shrink the largest eigenvalues of $\tilde{A}$, which is unnecessary to be related to $tr(\tilde{A})$.
5. Is the result a corollary from the inference on the hidden stochastic block structure?  The two steps of estimating controllability and block structures seem to be isolated with each other.

**Summary Of The Paper:**

This paper studies network controllability with partial knowledge of the network structure. It provides insight into how the controllability can be approximated from the topology.

**Summary Of The Review:**

This paper established a framework for inferring the global controllability measurement based on the community information. The scope might be narrow for the computer science group. But the topic itself is very interesting and lays on the frontier of control theory and network neuroscience fields.

---

> ### Author Response · Authors · 2021-11-20
> **We clarified the reason our work is relevant and significant for the computer science community. We elaborated on the concerns around generalizability or our work, as well as complementary examples where controllability estimation from coarse networks become important. We explained the statistical significance shown in Fig.2 and the self-loop existence assumptions in our modeling. We clarified the connection between controllability estimation and community recovery.**
>
> *[R2: 1] Limited scope for the computer science community; Framework generality*
>
> Thank you for the feedback. Please see our answer to [R1: 10]: SBMs, community structure and inference, as well as low-rank approximations (closely related to the coarsening) are extremely popular among the computer science (CS) community. At the same time, and from a different angle, control theory has been revived into CS through reinforcement learning. Therefore, we believe that as the use of ML tools in CS provides greater opportunities to solve important problems in control systems, one example of which investigated in our paper.
>
> *Framework generality:* We emphasize that our work is the first of its kind that characterizes the trade-off between controllability of coarse-measured systems and that of fine-scale networks. Our motivation to consider SBMs for networks stems from the structure of many real-world networks, including: brain networks, which are, at least empirically, known to have community structure across various spatial scales [1]; social and power networks. However, we agree with the reviewer that SBM may restrict the generalizability of our results to other important network models, which we leave as a subject for future research.
>
> *[R2: 2] Examples of controllability estimation*
>
> Thanks for this recommendation. The coarse network controllability framework presented find various examples in real-world applications. For instance, recent efforts are going towards improving neuromodulation techniques by finding optimal locations (in other words, the most controllable nodes; using brain imaging data) from which the brain is stimulated. This approach has therapeutic benefits in patients of neurological disorders (e.g. epilepsy) who do not respond to other treatment methods. Another application is found in social networks where a coarse graph is available for inference making. This can be for reasons such as keeping anonymity of people or lacking sufficient memory to keep the data of a large population of people. The objective is then finding which groups of people (each mapped to a coarse node) play most significant roles in changing the behaviour of the entire system, like their political tendencies. Another example is related to pandemic and vaccination data. Let the coarse graph represent the partial formation of the vaccination percentage of groups of people in a social network. The important question is then prioritizing the groups by the significance of their immunization to achieve minimum infection/death rate. We will add these illustrative examples to the paper introduction.
>
> *[R2: 3] Statistical significance of the results in Fig.2?*
>
> We thank the reviewer for the detailed attention. As mentioned in the Figure caption, the shaded regions show one standard deviation below and over the mean values, over a set of 20 independent samples. This means the statistical significance of values is ~68%. We can provide updated Figures that indicate two standard deviations, which will then enjoy a ~95% confidence interval.
>
> *[R2: 4] The assumptions on the self loops*
>
> We appreciate the reviewer’s close attention and thoughts. First, we should point out that the existence of self-loops usually depends on the modeling choice. E.g. brain networks computed from e.g. brain imaging data contain self-loops [2,3]. The reason for the existence of these self-loops is that regions of interest (ROIs) when parcelling the brain imaging data cannot be very small, and so each ROI covers a number of neurons that have inter-connections. Hence, SBM (i.e. with self-loops) is still a valid modeling for the brain networks. However, as correctly mentioned by the reviewer, it is common to remove the self-loops when a brain graph is used in an LTI system dynamics [2,3]. This way, paper derivations remain mainly the same except for minor modifications.
>
> *[R2: 5] The stability parameter in Eq.(11) is unrelated to tr(A~).*
>
> This is a correct statement. The constant parameter is there to ensure stability of the system, i.e. $\rho(\tilde{A})/\tilde{c} tr(\tilde{A})<1$. Please also see our reply to Reviewer 3 [R3:8].
>
> *[R2: 6] Seemingly isolated controllability estimation and block structure estimation.*
>
> The main results of the paper are: a) Thm.1 and Thm.2 that characterize error bounds on the classic (MOR) approach and the proposed learning approach, respectively. b) the learning-based algorithm to estimate coarse controllability described in Alg.1. One step of Alg.1 is recovering the underlying community memberships, one possible method of which is described in Alg. 2.
>
> *[1] Sporns, et al. "Modular brain networks." Annual review of psychology 67 (2016): 613-640.*
>
> *[2] Gu, et al. (2015). Controllability of structural brain networks. Nature communications, 6(1), 1-10.*
>
> *[3] Khambhati, et al. (2019). Functional control of electrophysiological network architecture using direct neurostimulation in humans. Network Neuroscience, 3(3), 848-877.*

---

### Official Review · Reviewer_zMsx · 2021-11-04

**Correctness:** 3
**Technical Novelty And Significance:** 3
**Empirical Novelty And Significance:** 1
**Recommendation:** 3
**Confidence:** 2

**Main Review:**

What is the motivation of studying the controlability problem on SBMs?

page 3: It would be useful to explain Gramian and other background on dynamical systems, since typical ML reviewers
might not be completely familiar with the literature

Definition 1, 3rd line: should be v\in V'_k. Also in the line right after Definition 1

page 4: what does "c-nodes can cover the fine graph" mean? What is the notion of covering here?

The paper initially discusses partial observability, but it is not clear how that fits into the problem
formulation

The use of the term synchronization here seems odd, since this is usually considered in the context of
dynamical systems. Here it is about coverage of the coarse communities.

What is the justification for assumptions 2 and 3?

The problem statement in section 4 is not at all clear.

Remark 2: "This assumption is a very weaker condition than asking..." (needs to be rephrased)

The experiments section is quite limited. It would be useful to consider other networks as well.

The use of "learning" is not consistent with learning theory, and should be explained better

page 6: "Broadly, our analysis highlight the role of" (should be highlights)

**Summary Of The Paper:**

The paper studies the problem of controlling the dynamics of a networked dynamical system,
under partial observations. Such systems arise in a number of applications, and therefore the
control problem is important and well motivated.
The authors consider a reduced order system from coarse data,
and derive bounds on the convergence and approximation error for the original dynamical systems model.
These are evaluated empirically on a small synthetic dataset

**Summary Of The Review:**

The paper is written very poorly, and is hard to follow. Even the problem statement is not clear. The authors study the controlability problem for SBMs. It is not clear how well this is motivated from the applications they mention. The technical contribution is not very clear, and the experimental section is very weak. The fit for ICLR is not very clear

---

> ### Author Response · Authors · 2021-11-20
> **We elaborated the significance of studying controllability of SBMs. We provided clarification on the problem statement we studied. We explained the meaning of the notions of “coverage” by c-nodes and “synchronizability” in the context presented in the paper.**
>
> *[R1: 1] What is the motivation of studying the controllability problem on SBMs?*
>
> Many real world networked systems (dynamical systems evolving over networks) have built in community structure. A few such systems include power networks [1] and, most importantly, brain networks. The latter, at least empirically, are known to have community structure across spatial scales [2,3,4]. SBM and its variants, although a slightly restrictive class of generative models, provide a powerful modeling framework to facilitate fundamental understanding of network community organization. Moreover, the algorithms derived using SBM modeling assumption have proven to be generalizable and applicable to real-networks that are not necessarily realized from SBMs. Hence, we chose SBMs to characterize network controllability; specifically, to understand the impact of community structure on the scalar measures of controllability Gramian. To the best of our knowledge, our work is the first paper that theoretically shows explicit connection between community properties of the graph and the average controllability of its nodes (See Eq.17).  The use of generative random models (additive or multiplicative noise-driven dynamical systems with parameter uncertainties) to study performance measures such as controllability, observability, and stability has a long history since the pioneering works of Kalman and Bellman. In a similar vein, based on our rigorous theoretical results, we believe that SBMs can provide a starting point to model uncertainty in the network structure of networked dynamical systems. Importantly, “the uncertainty is in the structure of the network.”
>
> *[R1: 2] Unclear problem statement:*
>
> We appreciate the reviewer’s observation. The problem formulation revolves around coarse observability in two ways. Note that this observability is very different from the notion of dynamical systems’ observability.  First, our goal is to assess group average controllability -- the average controllability of groups of fine nodes that map to a c-node. Second, the only data available from which we can conduct an estimate is ~A, i.e. the coarse graph composed of partial observations.
> We will add additional clarifications to the problem statement section.
>
> *[R1: 3] page 4: what does "c-nodes can cover the fine graph" mean? What is the notion of covering here?*
>
> In our paper, we used covering and measuring interchangeably.  Recall that c-nodes are obtained by coarsening the fine graph. Thus, when we say “c-nodes can cover the graph sparsely”, we mean c-nodes can “measure” a small part of the fine graph. We understand this might be confusing as we have not explicitly defined the notion of coverage prior to this. We will include additional explanations in the final version.
>
> *[R1: 4] The use of the term synchronization here seems odd, since this is usually considered in the context of dynamical systems. Here it is about coverage of the coarse communities.*
>
> This is an interesting observation. Our usage is different from that in dynamical systems literature. Our choice of the term “synchronizability” was made to facilitate reader’s visualization of the scenario when each coarse measurement covers all the fine nodes that belong to a single community (c.f. Remark 1).
>
> *[1] N. Xue and A. Chakrabortty. "Control Inversion: A Clustering-Based Method for Distributed Wide-Area Control of Power Systems." IEEE*
>
> *[2] Pavlović, Dragana M., et al. "Multi-subject stochastic blockmodels for adaptive analysis of individual differences in human brain network cluster structure." NeuroImage 220 (2020): 116611.*
>
> *[3] Faskowitz, Joshua, et al. "Weighted stochastic block models of the human connectome across the life span." Scientific reports 8.1 (2018): 1-16.*
>
> *[4] Sporns, Olaf, and Richard F. Betzel. "Modular brain networks." Annual review of psychology 67 (2016): 613-640.*

---

> > ### Author Response · Authors · 2021-11-20
> > **We explained the feasibility of Assumptions 2 and 3. We elaborated on the concern around the experiments section as well as the “learning” term used in the paper. We discussed in detail the contribution of our work.**
> >
> > *[R1: 5] What is the justification for assumptions 2 and 3?*
> >
> > We thank the reviewer for the comment. Due to space constraints we removed the following justifications for these two assumptions in the submission. The following explanations will be added to the final draft.
> >
> > “*Assumption 2* requires at least one c-node per community to be perfectly synchronized with the community. This assumption will be held with high likelihood as in real-world networks. For instance, the formation of communities can be highly coherent with the actual geometrical (or topological) distance.  Nodes that are in close proximity to one another, are more likely to form communities. This way, since coarse measurements usually target topologically-close nodes too, it becomes very unlikely that all the measurements sit on the boundaries of communities, rather most measurements cover inside each community. This is true since in typical scenarios, there are many more measurements than communities.”
> >
> > “We should first note that community sizes of the fine graph scale with the fine graph size n (Assumption 1, which is in line with the literature). *Assumption 3* then ensures that all communities are covered by c-nodes almost uniformly, so that we will not end up with lots of measurements from one community but only a few from another community. This means the total measurement coverage per community (i.e. $1^T \Phi$) must scale with m, the number of c-nodes.”
> >
> > *[R1: 6] The experiments section is quite limited. It would be useful to consider other networks as well.*
> >
> > The goal of this submission is to develop a theoretical framework that allows for the estimation of controllability measures for networked systems, and to qualitatively characterize the effect of factors such network size, community structure, and the coarseness of measurements. This analysis specifically reveals the power of learning good representations of the underlying network and suggests novel algorithms. A richer set of experiments would complement our theoretical results, but at this stage is beyond the scope of our work. That said, we have an active collaboration with practicing neuroscientists to validate our theoretical results on real brain imaging datasets. These studies will eventually take the form of a separate, follow-on publication.
> >
> > *[R1: 7] The use of "learning" is not consistent with learning theory, and should be explained better.*
> >
> > Our algorithm relies on mixed membership community detection, which is a form of unsupervised learning. The adoption of such a learning-based technique for solving the problem at hand, i.e. estimating the average controllability from coarse data, is a first step towards future work of extending the proposed solution technique to more general models. For instance, one might consider more complex data-driven graph representation learning algorithms (say, based on graph neural networks) as a part of the control pipeline, which is a fascinating avenue for future research.
> >
> > *[R1: 8] The paper is written very poorly, and is hard to follow. *
> >
> > As the paper established connections between that of the traditional control theoretic MOR approach and those of a modern Mixed Membership based learning approach, we discuss both philosophies. This may be at the root of the reviewer’s sense that the paper is poorly presented or the main ideas are unclear. For the final version, we will go through the paper carefully and rearrange parts to improve the readability.
> >
> > *[R1: 9] The technical contribution is not very clear.*
> >
> > Due to the interdisciplinary nature of our work drawing on very disparate fields such as controls and community structure learning, our initial write-up might not have clearly reflected our contributions. In the revised version, we plan to clarify this in detail. For reviewer’s convenience, we summarize our key contributions. For the first time, we propose an estimation/learning framework for inferring controllability measures (which provides us with information about the degree of controllability) of fine networks using coarse networks. We provide tight upper bounds on the difference between estimated and true measure. We highlighted the trade-off between our proposed learning based error to the error associated with the MOR based approach. The reason for considering MOR is that it is a well-known method for understanding large scale dynamical systems behavior using lower order models. Since coarsening results in reduced order networks in some sense, we chose a MOR based approach for comparison.. Existing works that use the MOR approach, have provided some bounds though in other contexts [5]. Nevertheless, to the best of our knowledge, none of these works address the problem setup in our paper.
> >
> > *[5] X. Cheng and J. M. A. Scherpen. "Model Reduction Methods for Complex Network Systems." Annual Review of Control, Robotics, and Autonomous Systems 4 (2021): 425-453.*

---

> > > ### Author Response · Authors · 2021-11-20
> > > **We commented on the relevance and significance of our work to the ICLR/CS community.**
> > >
> > > *[R1: 10] The fit for ICLR is not very clear*
> > >
> > > SBMs, community structure and inference, as well as low-rank approximations (which are inherently closely related to the coarsening and spectral community recovery presented in our paper) are extremely popular among the computer science (CS) community. At the same time, and from a different angle, control theory has been revived into CS through growing interest in reinforcement learning. Therefore, we believe that as the ML tools in CS evolve, they provide greater opportunities to solve important problems in control systems. One example of such problems is finding the controllability of a network using coarse measurement, which we chose to investigate in our paper. In a real-world scenario, the graph can change over time and the coarse controllability is then computed as time passes.
> > >
> > > *[R1: 11] Add background on dynamical systems*
> > >
> > > Thank you for the comment. We will add a discussion of these objects in the final version of the manuscript.
> > >
> > > *[R1: 12] Minor corrections for Def. 1, Remark 2, and a sentence on page 6.*
> > >
> > > We thank the reviewer and will correct this in the final version.

---

> > ### Comment · Reviewer_zMsx · 2021-11-29
> > **Thanks for the response**
> >
> > SBMs are certainly good theoretical models to study in general, but they tend to have less heterogeneity in degrees. Is that an issue in the applications here. Power networks are quite different, I think. Ref [2] mentioned in the response uses Het-SBM.
> >
> > Regarding the other comments, I agree this is within the ICLR scope

---

> > > ### Author Response · Authors · 2021-12-02
> > > **We commented on the generality of the SBM framework used in the paper. We compared the work of Ref [2] and that of ours.**
> > >
> > > We thank the reviewer for the insightful comment. We agree that the degree-corrected SBM (DCSBM) has been found to better model realistic networks. We must emphasize that several of the techniques presented in our paper, especially the proposed algorithm, should extend to the heterogeneous-degree case with minimal modifications. Deriving the precise theoretical results is an interesting opportunity for future work.
> > >
> > > The second part of our answer is regarding the reviewer’s comment about Het-SBM in ref [2]. The heterogeneity in ref [2] is inherently different from degree-heterogeneity (degree-corrected) SBMs, despite using similar terminology which we agree at first glance could become confusing. From the definition of Het-SBM in Section 2.4. in [2], we observe the heterogeneity is in the domain of “subject-specific” community-related parameters, while the model itself (eq. 14 in [2]) is the exact modeling we used for the fine graph (eq. 3 in our paper). Ref [2] investigates variations of community-structured parameters “across subjects”, while one could say (from this perspective) our paper does per-subject analysis and rather focuses on the study of coarsening properties. The multi-subject vs coarsening studies are two distinguished realms. Hence, the model in our paper is usable for the Het-SBM application referred to in [2], except that subject-subject parameters would need to be connected such that hence a joint optimization problem would then need to be solved.
> > >
> > > Ref [2], along with many other papers in the network neuroscience literature, are examples that our methods are quite general.  We must emphasize that the graphs we model are both weighted and mixed membership (the general model that has only recently been studied). One important aspect of our work is to show that these graphs are very often (and sometimes even implicitly) the output of a coarsening process from an underlying fine graph that can be modeled by a classic SBM.

---

### Decision · Program_Chairs · 2022-01-20

**Decision:**

Reject

**Comment:**

The paper considers the problem of controlling the dynamics of a networked dynamical system, under partial observations, considering a reduced order system from coarse data, and providing approximation bounds and an empirical evaluation.  Reviewers agree it is a borderline paper.  Technical results are nontrivial, and it introduces new questions, but the main contribution is rather narrow and it could be better written.